# Design, Synthesis, and Characterization of Novel Thiazolidine-2,4-Dione-Acridine Hybrids as Antitumor Agents

**DOI:** 10.3390/molecules29143387

**Published:** 2024-07-18

**Authors:** Monika Garberová, Zuzana Kudličková, Radka Michalková, Monika Tvrdoňová, Danica Sabolová, Slávka Bekešová, Michal Gramblička, Ján Mojžiš, Mária Vilková

**Affiliations:** 1Institute of Chemistry, Faculty of Science, Pavol Jozef Šafárik University, Moyzesova 11, 040 01 Košice, Slovakia; monika.garberova@student.upjs.sk (M.G.); zuzana.kudlickova@upjs.sk (Z.K.); monika.tvrdonova@upjs.sk (M.T.); danica.sabolova@upjs.sk (D.S.); 2Department of Pharmacology, Faculty of Medicine, Pavol Jozef Šafárik University, Trieda SNP 1, 040 01 Košice, Slovakia; radka.michalkova@upjs.sk (R.M.); jan.mojzis@upjs.sk (J.M.); 3Thermo Fisher Scientific, Mlynské Nivy 5, 821 09 Bratislava, Slovakia; slavka.bekesova@thermofisher.com (S.B.); michal.gramblicka@thermofisher.com (M.G.)

**Keywords:** thiazolidine-2,4-dione, acetamido functionality, acridine, NMR, structure elucidation, antiproliferative activity, DNA and BSA binding activity

## Abstract

This study focuses on the synthesis and structural characterization of new compounds that integrate thiazolidine-2,4-dione, acridine moiety, and an acetamide linker, aiming to leverage the synergistic effects of these pharmacophores for enhanced therapeutic potential. The newly designed molecules were efficiently synthesized through a multi-step process and subsequently transformed into their hydrochloride salts. Comprehensive spectroscopic techniques, including nuclear magnetic resonance (NMR), high-resolution mass spectrometry (HRMS), infrared (IR) spectroscopy, and elemental analysis, were employed to determine the molecular structures of the synthesized compounds. Biological evaluations were conducted to assess the therapeutic potential of the new compounds. The influence of these derivatives on the metabolic activity of various cancer cell lines was assessed, with IC_50_ values determined via MTT assays. An in-depth analysis of the structure–activity relationship (SAR) revealed intriguing insights into their cytotoxic profiles. Compounds with electron-withdrawing groups generally exhibited lower IC_50_ values, indicating higher potency. The presence of the methoxy group at the linking phenyl ring modulated both the potency and selectivity of the compounds. The variation in the acridine core at the nitrogen atom of the thiazolidine-2,4-dione core significantly affects the activity against cancer cell lines, with the acridin-9-yl substituent enhancing the compounds’ antiproliferative activity. Furthermore, compounds in their hydrochloride salt forms demonstrated better activity against cancer cell lines compared to their free base forms. Compounds **12c**·**2HCl** (IC_50_ = 5.4 ± 2.4 μM), **13d** (IC_50_ = 4.9 ± 2.9 μM), and **12f**·**2HCl** (IC_50_ = 4.98 ± 2.9 μM) demonstrated excellent activity against the HCT116 cancer cell line, and compound **7d**·**2HCl** (IC_50_ = 4.55 ± 0.35 μM) demonstrated excellent activity against the HeLa cancer cell line. Notably, only a few tested compounds, including **7e**·**2HCl** (IC_50_ = 11.00 ± 2.2 μM), **7f** (IC_50_ = 11.54 ± 2.06 μM), and **7f**·**2HCl** (IC_50_ = 9.82 ± 1.92 μM), showed activity against pancreatic PATU cells. This type of cancer has a very high mortality due to asymptomatic early stages, the occurrence of metastases, and frequent resistance to chemotherapy. Four derivatives, namely, **7e**·**2HCl**, **12d**·**2HCl**, **13c**·**HCl**, and **13d**, were tested for their interaction properties with BSA using fluorescence spectroscopic studies. The values for the quenching constant (*K*_sv_) ranged from 9.59 × 10^4^ to 10.74 × 10^4^ M^−1^, indicating a good affinity to the BSA protein.

## 1. Introduction

Cancer remains a leading cause of mortality worldwide. Developing new therapeutic agents that combine improved efficacy and reduced side effects is a major challenge in medicinal chemistry [1]. Researchers are actively exploring the incorporation of various molecular moieties into drug candidates to modulate their pharmacological properties while minimizing toxicity to normal tissue, which is a significant source of adverse effects [2,3]. Among the various chemical entities investigated for their therapeutic potential, thiazolidine-2,4-dione and acridine derivatives have attracted significant attention due to their broad spectrum of biological activities.

Thiazolidine-2,4-diones (Figure 1), which act as agonists of PPARγ (peroxisome proliferator-activated receptor), are well known for their ability to reduce serum glucose levels in patients with diabetes [4,5,6]. However, their clinical use has been restricted due to significant side effects and toxicity, largely attributed to the full agonistic activity at PPARγ’s binding site [7,8,9]. In addition to their metabolic effects, these PPARγ agonists have demonstrated the ability to induce apoptosis, arrest the cell cycle, and promote differentiation in various cancer cell lines. Extensive research has highlighted the antitumor potential of diverse thiazolidinone-2,4-diones across a wide range of tumor types (Figure 2) [10,11,12]. Recent studies suggest that thiazolidine-2,4-diones with a benzylidene double bond can induce apoptosis and cell cycle arrest independently of PPARγ activation [13,14]. Additionally, these compounds can inhibit glucose transporters (GLUTs), which are often upregulated in cancer cells, providing a selective mechanism to eliminate tumor cells [9,10]. The critical structural features of all these antitumor agents include aryl acetamido functionality (Ar–NH–CO–CH_2_) and thiazolidine-2,4-dione with a benzylidene double bond [9,10,15].

In parallel, the integration of an acridine core into thiazolidine-2,4-dione frameworks represents a novel and promising direction in drug development [16,18]. Acridine derivatives exhibit a wide range of biological activities, including antitumor [19,20,21], antibacterial [22,23], antiviral [24,25], and antifungal properties [22], making them valuable scaffolds for drug design. These compounds exert their effects through multiple pathways, such as DNA intercalation [26], inhibition of topoisomerases I/II [27,28], and reduction in drug resistance [29], highlighting their potential as multifaceted antitumor agents [30].

Based on these findings, the design of new antiproliferative agents should include three essential components: an acridine skeleton, thiazolidine-2,4-dione with a benzylidene double bond, and an aryl acetamido functionality (Figure 2). This study focuses on the synthesis and structural characterization of new compounds that integrate thiazolidine-2,4-dione, acridine moieties, and an acetamide linker, aiming to leverage the synergistic effects of these pharmacophores for enhanced therapeutic potential. The diverse bioactivity of each moiety provides a robust foundation for developing multifunctional compounds with superior efficacy and selectivity.

Comprehensive spectroscopic techniques, including nuclear magnetic resonance (NMR), high-resolution mass spectrometry (HRMS), and infrared (IR) spectroscopy, were employed to accurately determine the molecular structures of synthesized compounds.

Biological evaluations were conducted to assess the therapeutic potential of the new compounds. By integrating synthetic chemistry, spectroscopic analysis, and biological evaluation, this research aims to advance drug discovery efforts. The ultimate goal is to develop new compounds with promising pharmacological properties that can lead to next-generation therapeutic agents, addressing unmet medical needs and benefiting patient care.

## 2. Results and Discussion

Considering the structural features [10,16,17] mentioned in the Introduction section and to further probe the structure–activity relationship (SAR), we designed the structures of some novel derivatives incorporated with acridine, an aryl acetamido-ether linker (Ar–NH–CO–CH_2_–O), and thiazolidine-2,4-dione with benzylidene double bond residues (Figure 2).

### 2.1. Synthesis

The synthetic pathways adopted to obtain the target compounds are illustrated in Figure 1 and Figure 2. The synthesis of the new derivatives **7** and **8** started with commercially available anilines **1a**–**g** (Figure 1). First, anilines **1a**–**g** were transformed into 2-chloro-*N*-phenylacetamides **2a**–**g** using a bifunctional reagent, i.e., chloroacetyl chloride [5,31,32]. However, subsequent substitution reactions between 2-chloro-*N*-phenylacetamides **2a**–**g** and 4-hydroxybenzaldehyde were unsuccessful; therefore, 2-bromo-*N*-phenylacetamides **3a**–**g**, containing better leaving groups, were synthesized as described by Ang and coworkers [33]. The substitution reactions, which resulted in 2-(4-formylphenoxy)-*N*-phenylacetamides **4a**–**g** and 2-(4-formyl-2-methoxyphenoxy)-*N*-phenylacetamides **5a**–**g** in 54% to 87% yields, were performed in refluxing acetone in the presence of anhydrous potassium carbonate and a catalytic amount of potassium iodide [7]. The next step included Knoevenagel condensation of derivatives **4a**–**g** and **5a**–**g** with thiazolidine-2,4-dione (**9**) in toluene under the catalysis of piperidine and glacial acetic acid at 110 °C [5]. Although products **6a**–**d** were obtained with high purity and excellent yields (78–87%), the reactions of **4e**–**g** and **5a**–**g** with thiazolidine-2,4-dione (**9**) were ineffective. Then, the introduction of acridine moiety via *N*-substitution on the thiazolidine-2,4-dione was accomplished. The reaction of **6a**–**d** with 9-(bromomethyl)acridine was carried out in refluxing acetone in the presence of anhydrous sodium carbonate and a catalytic amount of potassium iodide to give 2-(4-{[(5*Z*)-3-[(acridin-9-yl)methyl]-2,4-dioxo-1,3-thiazolidin-5-ylidene]methyl}phenoxy)-*N*-phenylacetamides **7a**–**d** with yields ranging from 54% to 56% (Figure 1). However, when the derivatives **6a**–**d** were similarly allowed to react with 4-(bromomethyl)acridine, the derivative **6a** failed to react under these conditions, while the other derivatives, i.e., **6b**–**d**, afforded the corresponding expected products **8b**–**d** but in very low yields (16–38%, Figure 1).

Due to the issues associated with the final step of the linear synthesis, a convergent strategy was adopted, as outlined in Figure 2. The first step involved the reaction of thiazolidine-2,4-dione (**9**) with KOH in ethanol at room temperature for 2 h. Subsequently, the potassium salt of thiazolidine-2,4-dione reacted with 9-(bromomethyl)acridine or 4-(bromomethyl)acridine, producing derivatives **10** and **11**, respectively. Then, derivatives **10** and **11** were subjected to Knoevenagel condensation with 2-(4-formylphenoxy)-*N*-phenylacetamides **4a**–**g** and 2-(4-formyl-2-methoxyphenoxy)-*N*-phenylacetamides **5a**–**g** to form the final derivatives **7a**–**g**, **8a**–**g**, **12a**–**g**, and **13a**–**g** with yields higher than 60%, except derivative **12d** (38%). Eventually, the final derivatives **7a**–**g**, **8a**–**g**, **12a**–**g**, and **13a**–**g** were transformed into hydrochlorides by bubbling HCl gas into their ethanolic suspensions (Figure 2). 

### 2.2. NMR Spectroscopy

The compounds **7**, **8**, **12**, and **13**, along with their hydrochlorides **7**·**2HCl**, **8**·**HCl**, **12**·**2HCl**, and **13**·**HCl**, were studied using 1D and 2D NMR experiments to determine their structure and spectral assignments. In this discussion, we will focus on the spectrum of compound **8b**. The ^1^H NMR spectrum of compound **8b** contains six signals: two two-proton singlets at 4.80 ppm and 5.60 ppm from the methylene groups CH_2_-3 and CH_2_-10, three one-proton singlets at 10.10, 7.97, and 9.17 ppm from protons H-1, H-4, and H-9‴, and one six-proton singlet at 3.71 ppm of two equivalent methoxy groups. The proton bound to the nitrogen atom N-1 was distinguished based on the ^1^H,^13^C-HSQC spectra since there was no correlation with carbon in the spectra. The singlets of two methylene groups, i.e., CH_2_-3 and CH_2_-10, were distinguished based on HMBC spectra (Figure 3). The 4.80 ppm (CH_2_-3) singlet had an HMBC correlation with one carbonyl carbon at 166.1 ppm (C-2), and the 5.60 ppm (CH_2_-10) singlet had HMBC correlations with two carbonyl carbons at 165.9 and 167.5 ppm. To distinguish between the chemical shifts of the carbonyl carbons C-6 and C-8, as well as to differentiate between two singlets at 7.97 and 9.17 ppm from protons H-4 and H-9‴, HMBC correlation analysis was used. The HMBC correlation was observed between the proton H-4 (7.97 ppm) and the carbonyl carbon C-6 (165.9 ppm). Therefore, the singlet belonging to proton H-9‴ had a chemical shift of 9.17 ppm.

The ^1^H,^1^H-COSY experiment, homonuclear coupling constants (from ^1^H NMR spectra), and characteristic splitting patterns were used to distinguish two spin systems of the acridin-4-yl fragment and two phenyl nuclei. Heteronuclear ^1^H,^13^C-HSQC spectra were used to determine the chemical shifts of protonated carbon atoms. The chemical shifts of the protons of the acridin-4-yl fragment were assigned based on HMBC correlations between the proton H-9‴ (9.17 ppm) and the carbons C-1‴ (128.1 ppm) and C-8‴ (128.5 ppm). The sequence of the protons of the two spin systems of the acridin-4-yl fragment was subsequently determined. Non-protonated carbon atoms of the acridin-4-yl fragment were assigned chemical shifts based on HMBC correlations (Figure 3). Crucial HMBC correlations for assigning chemical shifts to non-protonated carbons C-4‴ (132.1 ppm) and C-4‴a (146.1 ppm) were their correlations with the methylene protons CH_2_-10 (5.60 ppm). Carbon C-10‴a (147.7 ppm) was correlated with protons H-9‴ (9.17 ppm) and H-6‴ (7.89 ppm), carbon C-9‴a (126.0 ppm) was correlated with proton H-2‴ (7.55 ppm), and carbon C-8‴a (126.2 ppm) was correlated with protons H-5‴ (8.17 ppm) and H-7‴ (7.66 ppm) (Figure 4).

The chemical shifts to protons H-2′,6′ (7.63 ppm) and H-4′ (6.25 ppm), as well as carbons C-2′,6′ (97.9 ppm) and C-4′ (95.7 ppm), were determined by analyzing the characteristic shape, homonuclear coupling constants, and integral values of multiplets of the 1,3,5-trisubstituted phenyl. The non-protonated carbons C-3′,5′ were determined by the only HMBC correlation between methoxy group protons (3.71 ppm) and carbons with a chemical shift of 160.5 ppm. Similarly, the correlation between protons H-2′,6′ (7.63 ppm) and carbon with a chemical shift of 140.0 ppm was used to determine carbon C-1′ (Figure 4).

In addition, HMBC correlations between proton H-4 (7.97 ppm) and carbon with a chemical shift of 132.2 ppm (C-2″,6″) helped distinguish the multiplets belonging to protons H-2″,6″ and protons H-3″,5″ of the last spin system AA′BB′ of the 1,4-disubstituted phenyl nucleus. The chemical shift of protons H-2″,6″ is 7.66 ppm, and that of protons H-3″,5″ is 7.18 ppm. Crucial HMBC correlations were observed between the methylene group protons CH_2_-3 (4.80 ppm) and carbon C-4″ (159.7 ppm), and protons H-3″,5″ (7.18 ppm) and carbon C-1″ (126.1 ppm) to assign chemical shifts to non-protonated carbons C-1″ and C-4″ (Figure 4).

The configuration of the double bond C4=C5 of derivatives **7**, **8**, **12**, and **13** was determined based on heteronuclear coupling constants ^n^*J*_CH_ obtained from the ^1^H,^13^C-HMBC experiment measured without the suppression of the one-bond coupling constant ^1^*J*_CH_ and the EXSIDE experiment (Figure 5) [34]. The values of the coupling constants ^1^*J*_C4H4_ as well as ^3^*J*_C6H4_ confirm the *Z*-configuration of the C4=C5 bond [35,36,37,38].

By comparing selected ^1^H, ^13^C, and ^15^N chemical shifts of derivative **7** to derivative **8** and derivative **12** to derivative **13**, more significant differences were observed only in the chemical shifts of protons H-4 and H-10 and carbons C-4, C-5, C-6, C-8, and C-10 (Table 1 and Table 2). We assume that the shift of the ^1^H NMR resonance lines of proton H-10 of derivative **7** to higher ppm values compared to derivative **8** is caused by the anisotropy of all three acridine aromatic rings. The proton H-10 of derivative **8** is influenced only by the anisotropy of the side and middle acridine rings. The chemical shift values of carbon C-10 of derivatives **7** and **12** are mainly influenced by the electron-acceptor effect of the nitrogen atom N-10‴ of acridine. Compared to the chemical shift of carbon C-10 of derivatives **8** and **13**, it is more shielded, and it shifted to lower ppm values. The influence of nitrogen N-10‴ in derivatives **7** and **12** is also manifested by the higher polarization of the C4=C5 bond. For comparison, the difference in the chemical shifts of carbon atoms C4 and C5 for derivative **7a** is 16.2, and for derivative **8a**, it is only 14.4 (Table 1 and Table 2).

The structure of the hydrochloride derivatives **7**, **8**, **12**, and **13** was confirmed by CHN analysis and NMR spectroscopy. CHN analysis revealed that derivatives **7** and **12** are bound to two molecules of HCl, while derivatives **8** and **13** are bound to only one molecule of HCl.

The assignment of chemical shifts to individual atoms of the hydrochloride derivatives **7**, **8**, **12**, and **13** was conducted similarly as described above. Significant differences in chemical shifts were observed only in derivatives **7**·**2HCl** and **12**·**2HCl** compared to derivatives **7** and **12**. The largest changes were evident for the hydrogen and carbon atoms of the acridin-9-yl fragment (Figure 6).

In the ^1^H NMR spectra of hydrochlorides **7**·**2HCl** and **12**·**2HCl**, slight broadening of the resonance lines was observed. In the ^13^C NMR spectra, resonance lines of the C-3‴,6‴, C-4‴,5‴, C-4‴a,10‴a, and C-9‴ nuclei were not present (see NMR spectra in SI). However, in some cases, the chemical shifts of these atoms could be determined from 2D ^1^H,^13^C-HSQC and ^1^H,^13^C-HMBC spectra. Binding two molecules of HCl to molecule **7** resulted in the deshielding of the hydrogen and carbon nuclei and a shift of resonances to higher ppm values (Appendix A).

### 2.3. IR Spectroscopy

FTIR analysis was used to identify the characteristic functional groups of synthesized derivatives **7**, **8**, **12**, and **13** and their hydrochlorides. Infrared spectroscopic characterization revealed the presence of all the expected signals related to the functional groups. Some of the primary vibration modes of **8a**–**g** are shown in Figure 7, with the rest available in the ESI. The weak N–H stretching vibration originating from the amido group occurs in the range of 3420 to 3272 cm^−1^. Bands of very low intensity attributed to the C–H stretching vibrations are observed around 3070–2917 cm^−1^. The presence of the methoxy group of **b** and **c** series derivatives was confirmed by the appearance of weak absorption bands in the 2849–2824 cm^−1^ region, corresponding to the stretching OCH_3_ vibrations along with C–O stretching, which is sometimes observed as the splitting of the C–O vibration of the non-substituted derivatives (see IR spectrum of derivative **8c** in Supporting Information). Two typical absorption bands are noticed in the spectral region of 1745–1673 cm^−1^ associated with C=O frequencies, with compound **8a** showing a weak thiolactone band at 1742 cm^−1^ and a strong lactam band at 1673 cm^−1^. In lower spectral regions ranging from 1600 to 1400 cm^−1^, multiple mixed bands of aromatic C=C and C=N stretching vibrations appear with strong-to-medium intensity, together with C–H bending vibrations of acridine and aromatic moieties. For all derivatives with CF_3_ groups, strong bands are noticed at the frequencies of 1385–1294 cm^−1^, corresponding to C–CF_3_, C–C, and C–F stretching, sometimes overlapping with the C=C band. Further characteristic absorption bands appear around 1253–1243 cm^−1^, assigned as the C–O stretching vibrations, and in the lower region, C–N stretching vibrations at 1137–1041 cm^−1^ can be found. In the FTIR absorption spectra of nitro derivatives, two characteristic bands appear; the one at 1345–1320 cm^−1^ is assigned to the symmetric valence vibration of the C–NO_2_, and the antisymmetric one can be found around 1503 cm^−1^. A typical aromatic intense band appears around 755–738 cm^−1^, corresponding to the out-of-plane bending vibrations of aromatic hydrogens. The weak-to-medium peak in the range of 717 to 688 cm^−1^ indicated a C–S stretching vibration in the thiazolidinone ring. 

### 2.4. Biological Activity

#### 2.4.1. In Vitro Antitumor Activity

The assessment of cytotoxicity is fundamental in the development of novel anticancer agents. Herein, we present a comprehensive analysis of the cytotoxic properties of a series of compounds bearing diverse substituents (R^1^) on the phenyl ring, evaluated against a panel of cancer cell lines: lung adenocarcinoma (A549), hepatocellular carcinoma (Hep G2), ovarian adenocarcinoma (A2780), ovarian adenocarcinoma cisplatin-resistant (A2780cis), cervical adenocarcinoma (HeLa), triple-negative breast adenocarcinoma (MDA-MB-231), colorectal adenocarcinoma (HCT116), pancreas adenocarcinoma (PaTu8902), human melanoma (A2058), glioblastoma (U87), and acute T-lymphoblastic leukemia (Jurkat). The therapeutic safety of the newly synthesized molecules was assessed by evaluating their cytotoxicity on the two non-cancerous cell lines: epithelial breast cells (MCF-10A) and dermal fibroblasts (BJ-5ta) to display the selective cytotoxicity toward normal cells and cancer cells. IC_50_ values serve as a crucial metric for quantifying a compound’s ability to inhibit cell growth, with lower values indicating enhanced potency. Compounds with IC_50_ values exceeding 50 µM were considered ineffectual.

The results indicate that none of the studied compounds exhibited activity against the cancer cell lines A2780cis, MBA-MB-231, U87, and Jurkat within the observed concentration range. Therefore, the data were not included in Table 3 and Table 4. 

In various cell lines, compounds containing an unsubstituted phenyl ring showed varying efficacy. Compound **12a**·**2HCl** (IC_50_ = 6.55 ± 1.65 µM) demonstrated excellent efficacy against HCT116 cells. On the other hand, compound **7a**·**2HCl** (IC_50_ = 13.10 ± 2.30 µM) showed moderate efficacy against HCT116 cells, and compound **12a**·**2HCl** (IC_50_ = 14.95 ± 1.35 µM) exhibited moderate efficacy against A2780 cells. Both **7a**·**2HCl** and **12a**·**2HCl** also demonstrated activity against the non-cancerous cell lines MCF-10A and Bj-5ta.

Compounds containing the 3,5-dimethoxyphenyl group demonstrated significant efficacy. Compounds **12b** (IC_50_ = 7.30 ± 3.40 µM) and **7b**·**2HCl** (IC_50_ = 6.11 ± 1.20 µM) showed notable activity against HeLa cells. Additionally, moderate activity against the HeLa cell line was observed for derivatives **7b** (IC_50_ = 11.35 ± 2.75 µM), **8b** (IC_50_ = 12.03 ± 2.80 µM), and **12b**·**2HCl** (IC_50_ = 11.92 ± 7.40 µM). It is also worth mentioning the activity of **12b** (IC_50_ = 10.20 ± 2.10 µM) against HCT116 and of **12b**·**2HCl** (IC_50_ = 15.00 ± 1.40 µM) against A2780. Unfortunately, derivatives containing a 3,5-dimethoxyphenyl ring showed low selectivity, displaying relatively high activity against non-cancerous cell lines MCF-10A and Bj-5ta. 

The 3,4,5-trimethoxyphenyl substituent generally resulted in compounds with modest efficacy. For instance, compound **7c** (IC_50_ = 6.80 ± 2.40 µM) showed efficacy against HeLa cells, while **12c**·**2HCl** (IC_50_ = 5.40 ± 2.40 µM) exhibited high efficacy against HCT116 cells (Figure 8). Moderate efficacy was observed for compound **13c**·**HCl** (IC_50_ = 14.60 ± 3.30 and 14.30 ± 6.43 µM) against A549 and HeLa cell lines.

Compounds with the 4-nitrophenyl substituent demonstrated noteworthy efficacy. Compound **12d** (IC_50_ = 9.40 ± 0.30 µM) and **7d**·**2HCl** (IC_50_ = 4.55 ± 0.35 µM) showed notable activity against the HeLa cell line, and **13d** (IC_50_ = 4.90 ± 2.90 µM) and **7d**·**2HCl** (IC_50_ = 8.60 ± 2.90 µM) displayed notable activity against the HCT116 cell line (Figure 8). 

Compounds containing the 2-trifluoromethylphenyl substituent exhibited notable efficacy for **13e** (IC_50_ = 9.90 ± 1.70 µM) and **7e**·**2HCl** (IC_50_ = 8.90 ± 3.10 µM) against HCT116 cells, while **12e**·**2HCl** (IC_50_ = 13.16 ± 4.90 µM) and **13e**·**HCl** (IC_50_ = 13.70 ± 4.50 µM) showed moderate efficacy against HeLa cells (Figure 8). Compounds **12e** (IC_50_ = 14.30 ± 1.80 µM) and **13e**·**HCl** (IC_50_ = 14.50 ± 3.40 µM) displayed activity against HCT116 (Figure 8) and **7e**·**2HCl** (IC_50_ = 11.00 ± 2.20 µM) against the PATU cell line.

Compounds bearing 3-trifluoromethylphenyl, namely, **7f** (IC_50_ = 6.20 ± 1.30 µM) and **7f**·**2HCl** (IC_50_ = 6.90 ± 0.20 µM) displayed efficacy against HeLa cells, while compounds **7f** (IC_50_ = 8.80 ± 3.70 µM) and **12f**·**2HCl** (IC_50_ = 4.98 ± 2.90 µM) showed efficacy against HCT116, and compound **7f**·**2HCl** (IC_50_ = 9.82 ± 1.92 µM) showed efficacy against the PATU cell line. Moderate activity was observed for compounds **12f** (IC_50_ = 11.50 ± 3.30 µM) and **13f**·**HCl** (IC_50_ = 12.30 ± 4.90 µM) against the Hela cancer cell line. Next, compound **13f**·**HCl** (IC_50_ = 14.30 ± 3.20 µM) showed activity against HCT116, and **7f** (IC_50_ = 11.54 ± 2.06 µM) showed activity against PATU. These compounds showed very bad selectivity because they inhibited cell growth of the normal cell lines MCF-10A and Bj-5ta.

Among the compounds with the 3,5-ditrifluoromethylphenyl substituent, only compound **12g** (IC_50_ = 6.70 ± 2.00 µM) showed activity against the HCT116 cell line. Additionally, there was very low selectivity. 

It is important to note that only a few tested compounds, namely, **7e**·**2HCl**, **7f**, and **7f**·**2HCl**, showed activity against pancreatic PATU cells. This type of cancer has a very high mortality due to asymptomatic early stages, the occurrence of metastases, and frequent resistance to chemotherapy. 

These comparisons highlight the differential efficacy of compounds based on the substituent R^1^, underscoring the importance of structural modifications in modulating the potency of anticancer agents against specific cell lines.

#### 2.4.2. Structure–Activity Relationship (SAR) Analysis

Structure–activity relationship (SAR) analysis involves examining how variations in chemical structure impact the activity of compounds. In the dataset provided, the SAR can be deduced by comparing the IC_50_ values of compounds with different substituents (R^1^) against various cell lines.

An in-depth analysis of the structure–activity relationship (SAR) for the compounds evaluated against a spectrum of cancer cell lines revealed intriguing insights into their cytotoxic profiles. Compounds with 3-trifluoromethylphenyl, 2-trifluoromethylphenyl, or 3,5-dimethoxyphenyl tend to have lower IC_50_ values across multiple cell lines compared to compounds with unsubstituted phenyl. Conversely, compounds featuring the 4-nitrophenyl substituent generally exhibit diminished activity. However, compounds with 3,5-dimethoxyphenyl and 3-trifluoromethylphenyl showed low selectivity, while compounds with 3,4,5-trimethoxyphenyl, 4-nitrophenyl, and 2-trifluoromethylphenyl displayed high selectivity. This suggests that these groups enhance the antiproliferative activity of the compounds, possibly by increasing their interaction with cellular targets or enhancing their cellular uptake. Moreover, specific substitutions on the aromatic ring may contribute to the selectivity of compounds toward cancer cells, potentially by targeting pathways or receptors overexpressed in cancer cells (Figure 9).

Compounds with larger substituents or a higher electron density on R^1^ generally exhibit lower IC_50_ values, suggesting increased potency. For example, compounds with 3,4,5-trimethoxyphenyl or 3,5-bistrifluoromethylphenyl tend to have lower IC_50_ values. This implies that bulkier or more electron-rich substituents may enhance the interaction of the compounds with their molecular targets, leading to improved antiproliferative activity.

Substitution at position 3″ on the second phenyl ring by a methoxy group has a significant impact on the activity of these molecules. The presence of the methoxy group at this position appears to modulate both the potency and selectivity of the compounds, likely by influencing the electronic properties and steric interactions within the cellular environment (Figure 9).

The variation in the substituent, either acridin-9-yl or acridine-4-yl, at the nitrogen atom of the thiazolidine-2,4-dione core significantly affects the activity against cancer cell lines. The acridin-9-yl substituent typically enhances the compounds’ antiproliferative activity, potentially due to its planar structure which may facilitate better intercalation with DNA or interaction with other cellular targets (Figure 9).

Furthermore, compounds in their hydrochloride salt forms demonstrated better activity against cancer cell lines compared to their free base forms. This increased activity could be attributed to several factors, including improved solubility in aqueous environments, which enhances bioavailability and cellular uptake.

Overall, SAR analysis revealed the importance of substituent effects on the antiproliferative activity and selectivity of the compounds. Understanding these relationships can guide the design and optimization of future compounds with improved efficacy and reduced toxicity for cancer therapy.


molecules-29-03387-t003_Table 3Table 3IC_50_ values [μM] ± SD of the tested compounds **7**, **8**, **12**, and **13** on the cell lines after 48 h of incubation.Cmpd.R^1^

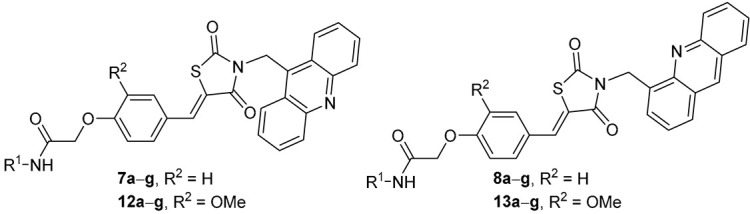

IC_50_ [µM]MCF-10ABj-5taA549Hep G2A2780HeLaHCT116PATUA2058
**7a**
Ph>50>50>50>50>5024.27 ± 3.837.6 ± 4.2>50>50
**7b**
3,5-diMeOPh28.8 ± 2.3>5034.6 ± 5.4>50>5011.35 ± 2.7534.5 ± 2.98>50>50
**7c**
3,4,5-triMeOPh14.5 ± 2.5>50>50>50>506.8 ± 2.4>50>50>50
**7d**
4-NO_2_Ph>50>50>50>50>5020.05 ± 2.95>50>50>50
**7e**
2-CF_3_Ph>50>50>50>50>50>5022.3 ± 4.6>50>50
**7f**
3-CF_3_Ph9.8 ± 1.932.6 ± 7.416.4 ± 1.8528.1 ± 5.5>506.2 ± 1.38.8 ± 3.711.54 ± 2.06>50
**7g**
3,5-diCF_3_Ph>50>5018.95 ± 3.85>50>5020.9 ± 2.818.0 ± 1.2>50>50
**12a**
Ph>50>50>50>50>5018.65 ± 0.45>50>50>50
**12b**
3,5-diMeOPh15.6 ± 2.6>5034.54 ± 5.4>50>507.3 ± 3.410.2 ± 2.1>50>50
**12c**
3,4,5-triMeOPh>50>50>50>50>5019.9 ± 2.0515.1 ± 2.0>50>50
**12d**
4-NO_2_Ph>50>5031.2 ± 1.1>50>509.4 ± 0.314.6 ± 1.7>50>50
**12e**
2-CF_3_Ph>50>5031.1 ± 4.2>50>5028.2 ± 3.714.3 ± 1.8>50>50
**12f**
3-CF_3_Ph>50>50>50>50>5011.5 ± 3,3>50>50>50
**12g**
3,5-diCF_3_Ph>50>50>50>50>5020.7 ± 5.36.7 ± 2.0>50>50
**8a**
Ph>50>50>50>50>5021.06 ± 7.226.9 ± 1.85>50>50
**8b**
3,5-diMeOPh>50>50>50>50>5012.03 ± 2.8>50>50>50
**8c**
3,4,5-triMeOPh>50>50>50>50>50>50>50>50>50
**8d**
4-NO_2_Ph>50>50>50>50>5044.9 ± 4.247.6 ± 3.8>50>50
**8e**
2-CF_3_Ph>50>50>50>50>5043.3 ± 5.224.3 ± 3.8>50>50
**8f**
3-CF_3_Ph>50>50>50>50>5027.65 ± 4.625.1± 7.8>50>50
**8g**
3,5-diCF_3_Ph>50>50>50>50>50>50>50>50>50
**13a**
Ph>50>50>50>50>5041.9 ± 5.7>50>5049.15 ± 0.85
**13b**
3,5-diMeOPh43.6 ± 3.341.7 ± 0.838.9 ± 2.1>50>5025.7 ± 3.4544.7 ± 4.1>5034.35 ± 2.45
**13c**
3,4,5-triMeOPh>50>50>50>50>5044.2 ± 4.123.1 ± 5.7>5042.5 ± 0.25
**13d**
4-NO_2_Ph>5038.45 ± 4.546.9 ± 3.1>50>5036.2 ± 5.234.9 ± 2.9>5038.05 ± 1.65
**13e**
2-CF_3_Ph14.9 ± 0.16>50>50>50>5026.65 ± 3.89.9 ± 1.7>5035.2 ± 2.87
**13f**
3-CF_3_Ph28.9 ± 5.6>50>50>50>5021.8 ± 6.436.3 ± 4.6>5031.9 ± 4.1
**13g**
3,5-diCF_3_Ph>50>50>50>50>5017.7 ± 2.924.9 ± 3.5>50>50**Cisplatin** [39]25.9 ± 2.131.0 ± 0.717.3 ± 2.214.0 ± 2.8NT30.4 ± 1.414.5 ± 2.520.7 ± 3.118.8 ± 5.5
molecules-29-03387-t004_Table 4Table 4IC_50_ values [μM] ± SD of the tested compounds **7·2HCl**, **8·HCl**, **12·2HCl**, and **13·HCl** on the cell lines after 48 h of incubation.Cmpd. R 

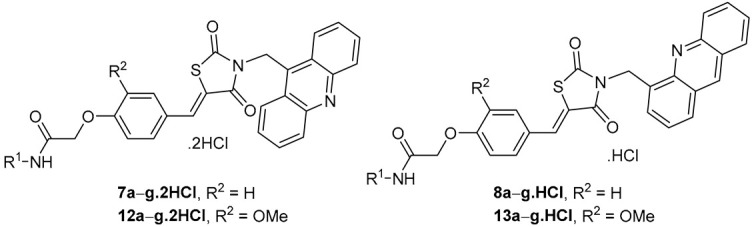

IC_50_ [µM]MCF-10ABj-5taA549Hep G2A2780HeLaHCT116PATUA2058
**7a·2HCl**
Ph17.6 ± 4.329.9 ± 7.216.96 ± 1.35>50>5017.7 ± 4.0413.1 ± 2.3>50>50
**7b·2HCl**
3,5-diMeOPh11.35 ± 2.0534.5 ± 5.521.95 ± 3.35>50>506.11 ± 1.217.4 ± 1.74>50>50
**7c·2HCl**
3,4,5-triMeOPh>50>50>50>50>5021.44 ± 5.816.95 ± 1.75>50>50
**7d·2HCl**
4-NO_2_Ph>50>50>50>50>504.55 ± 0.358.6 ± 2.9>50>50
**7e·2HCl**
2-CF_3_Ph>50>5034.3 ± 5.7>50>5020.5 ± 3.88.9 ± 3.111.0 ± 2.2>50
**7f·2HCl**
3-CF_3_Ph7.7 ± 1.539.5 ± 6.533.95 ± 6.0527.4 ± 5.9>506.9 ± 0.215.2 ± 3.49.82 ± 1.92>50
**7g·2HCl**
3,5-diCF_3_Ph13.23 ± 3.2>5032.4 ± 7.6>50>5019.3 ± 4.6>50>50>50
**12a·2HCl**
Ph19.3 ± 1.135.5 ± 4.5>50>5014.95 ± 1.3518.2 ± 5.16.55 ± 1.65>50>50
**12b·2HCl**
3,5-diMeOPh12.4 ± 1.929.6 ± 4.4>50>5015.0 ± 1.411.92 ± 7.427.3 ± 3.75>50>50
**12c·2HCl**
3,4,5-triMeOPh>50>50>50>50>5020.24 ± 5.45.4 ± 2.4>50>50
**12d·2HCl**
4-NO_2_PhNT>50>50>50NT19.32 ± 5.517.6 ± 4.530.8 ± 6.2>50
**12e·2HCl**
2-CF_3_Ph>50>50>50>50>5013.16 ± 4.929.25 ± 3.7>50>50
**12f·2HCl**
3-CF_3_Ph>50>50>50>50>5019.36 ± 6.484.98 ± 2.9>50>50
**12g·2HCl**
3,5-diCF_3_Ph>50>50>50>50>5025.14 ± 6.426.65 ± 9.01>50>50
**8a·HCl**
PhNT>50>50>50NT42.6 ± 0.75>50>50>50
**8b·HCl**
3,5-diMeOPh>50>50>50>50>5047.9 ± 1.432.4 ± 6.3>50>50
**8c·HCl**
3,4,5-triMeOPhNT>50>50>50NT>50>50>50>50
**8d·HCl**
4-NO_2_PhNT>50>50>50NT>50>50>50>50
**8e·HCl**
2-CF_3_PhNT>50>50>50NT>50>50>50>50
**8f·HCl**
3-CF_3_PhNT>50>50>50NT>50>50>50>50
**8g·HCl**
3,5-diCF_3_Ph45.0 ± 2.8>50>50>50>5045.2 ± 3.526.3 ± 3.4>50>50
**13a·HCl**
PhNT>50>50>50NT>50>50>50>50
**13b·HCl**
3,5-diMeOPhNT>50>50>50NT45.5 ± 2.9>50>50>50
**13c·HCl**
3,4,5-triMeOPhNT>5014.6 ± 3.323.4 ± 3.4NT14.3 ± 6.4>50>50>50
**13d·HCl**
4-NO_2_PhNT>50>50>50NT>50>50>50>50
**13e·HCl**
2-CF_3_PhNT>5042.8 ± 5.4>50NT13.7 ± 4.514.5 ± 3.439.8 ± 7.1>50
**13f·HCl**
3-CF_3_PhNT>5038.25 ± 7.35>50NT12.3 ± 4.914.3 ± 3.243.7 ± 5.8>50
**13g·HCl**
3,5-diCF_3_PhNT>50>50>50NT34.7 ± 6.4>50>50>50**Cisplatin** [39]25.9 ± 2.131.0 ± 0.717.3 ± 2.214.0 ± 2.8NT30.4 ± 1.414.5 ± 2.520.7 ± 3.118.8 ± 5.5


#### 2.4.3. Fluorescence Quenching Studies

Human serum albumin (HSA) and bovine serum albumin (BSA) play vital roles in drug transport and are frequently investigated as model proteins of blood plasma [40,41]. Fluorescence spectroscopy effectively provides detailed information on the binding mode, mechanism, and binding constants of various ligands or small molecules to proteins. 

BSA contains 582 amino acid residues, including tyrosine (Tyr), phenylalanine (Phe), and tryptophan (Trp). The tryptophan residues located at positions 134 and 212 being predominantly responsible for its intrinsic fluorescence due to their higher quantum yield than the other amino acids [41,42,43]. When excitation occurs at 280 nm, both Trp and Tyr contribute to the fluorescence of the protein [44]. In this study, we investigated the interaction of newly synthesized derivatives (**7e**·**2HCl**, **12d**·**2HCl**, **13c**·**HCl**, and **13d**), which have promising anticancer properties, with BSA. BSA (1 μM) was titrated with increasing concentration of acridine derivatives, and fluorescence spectra were recorded in the range of 300–500 nm (Figure 10). The fluorescence quenching for the quencher (Q) and protein interaction was analyzed using the Stern–Volmer equation [45,46]:$$F_{0}/F=1+K_{sv}\;[Q]=1+K_{q}\;\tau_{0}[Q]$$
where F_0_ and F denote the fluorescence intensities in the absence and presence of the quencher Q, respectively. *K_sv_* is the Stern–Volmer constant, *k_q_* is the bimolecular quenching rate constant (*k_q_ = K_sv_*/τ_0_), and τ_0_ is the lifetime of BSA in the absence of quencher, which is equal to 10^−8^ s for the tryptophan fluorescence of proteins [46].

The results (Figure 10) show that the intrinsic fluorescence of BSA was successfully suppressed by the addition of different concentrations of all studied acridine derivatives.

BSA exhibits a strong emission band of about 350 nm upon excitation at 280 nm. The gradual addition of acridine derivative to BSA resulted in decreased fluorescence emission with increasing concentration of the examined derivatives. Table 5 presents the calculated *K_sv_* and *k_q_* values for the interaction of acridine derivatives with BSA.

The acridine derivative **7e**·**2HCl** had the highest *K_sv_* value, while the lowest *K_sv_* value was determined for derivative **13d**. The studied compounds exhibited a tenfold higher *K_sv_* constant than acridine-thiosemicarbazone derivatives investigated by da Silva Filho et al. [47]. Values for the quenching constant (*K_sv_*) ranged from 9.59 × 10^4^ to 10.74 × 10^4^ M^−1^, indicating an appropriate affinity to the BSA protein.

## 3. Experimental

### 3.1. General

All the reagents were used as supplied without further purification. The progress of the reaction was monitored by analytical thin-layer chromatography using TLC sheets ALUGRAM-SIL G/UV254 (Macherey Nagel, Düren, Germany). Purification by flash chromatography was performed using silica gel (60 Å and 230–400 mesh, Merck, Darmstadt, Germany) with the indicated eluent. Melting points were determined on a Stuart^TM^ melting point apparatus SMP10 (Bibby Scientific Ltd. Staffordshire, United Kingdom) and were uncorrected. However, during the melting point determination, it was observed that some compounds underwent decomposition before reaching their true melting point. As a result, the definitive melting points could not be obtained for these derivatives, and only decomposition points (d) were observed.

### 3.2. General Synthetic Procedure for Compounds **2a**–**d**

To the solution of the corresponding amine (500 mg) in CHCl_3_ (10 mL), triethylamine (1.05 equiv.) was added. Subsequently, the mixture was cooled to 0 ⁰C, and the solution of chloroacetyl chloride (1.05 equiv.) in CHCl_3_ (2 mL) was added dropwise. The reaction was controlled using TLC for about 2 h. Then, the solvent was evaporated. The product was extracted into EtOAc and washed with an aqueous solution of NaHCO_3_. The combined organic phases were dried over anhydrous Na_2_SO_4_ and then evaporated under a vacuum. 

*2-Chloro-N-phenylacetamide* (**2a**). Beige solid. Yield: 880 mg (97%). Mp. 133–135 °C (134–137 °C [48]). ^1^H NMR (400 MHz, CDCl_3_): δ 8.24 (br s, 1H, NH), 7.55 (dd, *J* = 8.6, 1.1 Hz, 2H, H-2,6), 7.37 (t, *J* = 7.2 Hz, 2H, H-3,5), 7.18 (t, *J* = 7.2 Hz, 1H, H-4), and 4.20 (s, 2H, CH_2_) ppm.

*2-Chloro-N-(3,5-dimethoxyphenyl)acetamide* (**2b**). Light brown solid. Yield: 710 mg (95%). Mp. 98–100 °C (97–98 °C [48]). ^1^H NMR (400 MHz, CDCl_3_): δ 8.18 (br s, 1H, NH), 6.78 (d, *J* = 2.2 Hz, 2H, H-2,6), 6.29 (t, *J* = 2.2 Hz, 1H, H-4), 4.17 (s, 2H, CH_2_), and 3.79 (s, 6H, OCH_3_) ppm.

*2-Chloro-N-(3,4,5-trimethoxyphenyl)acetamide* (**2c**). Light brown solid. Yield: 695 mg (98%). Mp. 118–120 °C (115–117 °C [49]). ^1^H NMR (400 MHz, CDCl_3_): δ 8.16 (br s, 1H, NH), 6.84 (s, 2H, H-2,6), 4.19 (s, 2H, CH_2_), 3.87 (s, 6H, OCH_3_), and 3.83 (s, 3H, OCH_3_) ppm.

*2-Chloro-N-(4-nitrophenyl)acetamide (***2d***).* Dark orange solid. Yield: 746 mg (96%). Mp. 188–189 °C (188–190 °C [50]). ^1^H NMR (400 MHz, CDCl_3_): δ 8.53 (br s, 1H, NH), 8.26 (d, *J* = 9.2 Hz, 2H, H-3,5), 7.78 (d, *J* = 9.2 Hz, 2H, H-2,6), and 4.25 (s, 2H, CH_2_) ppm.

### 3.3. General Synthetic Procedure for Compounds **3a**–**g**

**Procedure 1** (for compounds **3a**–**d**): To the solution of the corresponding amine (500 mg) in DCM (10 mL), the anhydrous K_2_CO_3_ (1.2 equiv.) was added. Subsequently, the mixture was cooled to 0 ⁰C, and the solution of bromoacetyl bromide (1.2 equiv.) in DCM (3 mL) was added dropwise. The reaction was controlled using TLC for about 1 h. Then, the phases were separated, and the aqueous phase was extracted with DCM (2 × 15 mL). The combined organic phases were dried over anhydrous Na_2_SO_4_ and then evaporated under a *vacuum*. 

**Procedure 2** (for compounds **3e**–**g**): To the solution of the corresponding amine (500 mg) in DCM (10 mL), the anhydrous K_2_CO_3_ (1.2 equiv.) was added. Subsequently, the mixture was cooled to 0 ⁰C, and the solution of bromoacetyl bromide (1.2 equiv.) in DCM (3 mL) was added dropwise. The reaction was controlled using TLC for about 1 h. After the reaction was complete, the reaction mixture was filtered, the filtrate was evaporated, and the product was crystallized using a DCM/*n*Hex system.

*2-Bromo-N-phenylacetamide* (**3a**). White solid. Yield: 1103 mg (96%). Mp. 130–132 (CH_2_Cl_2_, 127–128 °C [51], 123–124 °C [52]). ^1^H NMR (400 MHz, CDCl_3_): δ 8.17 (br s, 1H, NH), 7.53 (d, *J* = 7.5 Hz, 2H, H-2,6), 7.36 (t, *J* = 7.4 Hz, 2H, H-3,5), 7.17 (t, *J* = 7.4 Hz, 1H, H-4), and 4.02 (s, 2H, CH_2_) ppm.

*2-Bromo-N-(3,5-dimethoxyphenyl)acetamide* (**3b**). White solid. Yield: 858 mg (96%). Mp. 99–100 (CH_2_Cl_2_, 90–93 °C [53]). ^1^H NMR (400 MHz, CDCl_3_): δ 8.08 (br s, 1H, NH), 6.76 (d, *J* = 2.2 Hz, 2H, H-2,6), 6.29 (t, *J* = 2.2 Hz, 1H, H-4), 4.01 (s, 2H, CH_2_), and 3.79 (s, 6H, OCH_3_) ppm.

*2-Bromo-N-(3,4,5-trimethoxyphenyl)acetamide* (**3c**). White solid. Yield: 813 mg (98%). Mp. 125–127 (CH_2_Cl_2_, 125–126 °C [52]). ^1^H NMR (400 MHz, CDCl_3_): δ 8.07 (br s, 1H, NH), 6.82 (s, 2H, H-2,6), 4.02 (s, 2H, CH_2_), 3.86 (s, 6H, OCH_3_), and 3.83 (s, 3H, OCH_3_) ppm.

*2-Bromo-N-(4-nitrophenyl)acetamide* (**3d**). Yellow solid. Yield: 919 mg (98%). Mp. 173–175 (CH_2_Cl_2_, 165–166 °C [52]). ^1^H NMR (400 MHz, CDCl_3_): δ 8.39 (br s, 1H, NH), 8.26 (d, *J* = 9.2 Hz, 2H, H-3,5), 7.76 (d, *J* = 9.2 Hz, 2H, H-2,6), and 4.07 (s, 2H, CH_2_) ppm.

*2-Bromo-N-(2-(trifluoromethyl)phenyl)acetamide* (**3e**). White solid. Yield: 743 mg (85%). Mp. 105–107 (CH_2_Cl_2_/*n*Hex, 104–106 °C [54]). ^1^H NMR (400 MHz, DMSO-d_6_): δ 10.01 (s, 1H, NH), 7.76 (d, *J* = 7.9 Hz, 1H, H-3), 7.71 (t, *J* = 7.7 Hz, 1H, H-5), 7.50 (m, 2H, H-4, H-6), and 4.10 (s, 2H, CH_2_) ppm.

*2-Bromo-N-(3-(trifluoromethyl)phenyl)acetamide* (**3f**). White solid. Yield: 700 mg (80%). Mp. 78–80 (CH_2_Cl_2_/*n*Hex, 83–84 °C [54]). ^1^H NMR (400 MHz, DMSO-d_6_): δ 10.75 (s, 1H, NH), 8.07 (s, 1H, H-2), 7.76 (d, *J* = 7.7 Hz, 1H, H-6), 7.58 (t, *J* = 7.7 Hz, 1H, H-5), 7.45 (d, *J* = 7.7 Hz, 1H, H-4), and 4.07 (s, 2H, CH_2_) ppm.

*N-(3,5-Bis(trifluoromethyl)phenyl)-2-bromoacetamide* (**3g**). White solid. Yield: 626 mg (82%). Mp. 104–105 (CH_2_Cl_2_/*n*Hex, 95–96 °C [52]). ^1^H NMR (400 MHz, DMSO-d_6_): δ 10.06 (s, 1H, NH), 8.24 (s, 2H, H-2,6), 7.83 (br s, 1H, H-4), and 4.10 (s, 2H, CH_2_) ppm.

### 3.4. General Synthetic Procedure for Compounds **4a**–**g** and **5a**–**g**

To a solution of 4-hydroxybenzaldehyde (200 mg, 1.64 mmol) or vaniline (200 mg, 1.31 mmol) in acetone (5 mL), anhydrous K_2_CO_3_ was added (1.2 equiv.), and the mixture was stirred for 20 min. Then, the corresponding 2-bromo-*N*-phenylacetamide **3a**–**g** (1.0 equiv.) and KI (0.2 equiv.) were added. The reaction mixture was stirred under reflux at 65–70 °C. The course of the reaction was monitored by TLC (*n*Hex:EtOAc, *v*/*v* 1:2). After the completion of the reaction, the formed precipitate was filtered and dried. 

*2-(4-Formylphenoxy)-N-phenylacetamide* (**4a**). White solid. Yield: 280 mg (67%). Mp. 129–130 °C (*n*Hex/EtOAc, 118–120 °C [1]). ^1^H NMR (600 MHz, DMSO-d_6_): δ 10.16 (s, 1H, H-1), 9.88 (s, 1H, H-4), 7.89 (d, *J* = 8.9 Hz, 2H, H-2″,6″), 7.62 (d, *J* = 8.6 Hz, 2H, H-2′,6′), 7.33 (t, *J* = 8.5 Hz, 2H, H-3′,5′), 7.19 (d, *J* = 8.9 Hz, 2H, H-3″,5″), 7.08 (t, *J* = 7.3 Hz, 1H, H-4′), and 4.85 (s, 2H, H-3) ppm. 

*N-(3,5-Dimethoxyphenyl)-2-(4-formylphenoxy)acetamide* (**4b**). White solid. Yield: 336 mg (65%). Mp. 131–133 °C (*n*Hex/EtOAc). IR ν_max_ 3396, 1681, 1595, 1426, 1247, 1160, 1055, 821, 685, 617 cm^−1^. ^1^H NMR (600 MHz, DMSO-d_6_): δ 10.10 (s, 1H, H-1), 9.88 (s, 1H, H-4), 7.89 (d, *J* = 8.9 Hz, 2H, H-3″,5″), 7.18 (d, *J* = 8.9 Hz, 2H, H-2″,6″), 6.89 (d, *J* = 2.3 Hz, 2H, H-2′,6′), 6.25 (t, *J* = 2.3 Hz, 1H, H-4′), 4.83 (s, 2H, H-3), and 3.71 (s, 6H, 2 × OCH_3_) ppm. ^13^C NMR (151 MHz, DMSO-d_6_): δ 191.3 (C-4), 165.9 (C-2), 162.7 (C-4″), 160.5 (3′,5′), 140.0 (C-1′), 131.7 (C-3″,5″), 130.1 (C-1″), 115.2 (C-2″,6″), 97.9 (C-2′,6′), 95.7 (C-4′), 67.0 (C-3), and 55.1 (OCH_3_) ppm. HRMS: *m*/*z* [M + H]^+^ for C_17_H_17_NO_5_ calc. 316.11795; exp. 316.11832.

*2-(4-Formylphenoxy)-N-(3,4,5-trimethoxyphenyl)acetamide* (**4c**). White solid. Yield: 393 mg (69%). Mp. 143–144 °C (*n*Hex/EtOAc). IR ν_max_ 3406, 1681, 1596, 1504, 1229, 1127, 1006, 849, 640 cm^−1^. ^1^H NMR (600 MHz, DMSO-d_6_): δ 10.08 (s, 1H, H-1), 9.88 (s, 1H, H-4), 7.90 (d, *J* = 8.8 Hz, 2H, H-3″,5″), 7.18 (d, *J* = 8.7 Hz, 2H, H-2″,6″), 7.03 (s, 2H, H-2′,6′), 4.82 (s, 2H, H-3), 3.74 (s, 6H, 2 × OCH_3_), and 3.62 (s, 3H, OCH_3_) ppm. ^13^C NMR (151 MHz, DMSO-d_6_): δ 191.4 (C-4), 165.7 (C-2), 162.8 (C-4″), 152.7 (C-3′,5′), 134.5 (C-1′), 133.8 (C-4′), 131.8 (C-3″,5″), 130.1 (C-1″), 115.2 (C-2″,6″), 97.4 (C-2′,6′), 67.1 (C-3), 60.1 (OCH_3_), and 55.8 (2 × OCH_3_) ppm. HRMS: *m*/*z* [M + H]^+^ for C_18_H_19_NO_6_ calc. 346.12851; exp. 346.1288.

*2-(4-Formylphenoxy)-N-(4-nitrophenyl)acetamide* (**4d**). Yellow solid. Yield: 477 mg (97%). Mp. 189–191 °C (*n*Hex/CHCl_3_). ^1^H NMR (600 MHz, DMSO-d_6_): δ 10.79 (s, 1H, H-1), 9.88 (s, 1H, H-4), 8.25 (d, *J* = 9.3 Hz, 2H, H-3′,5′), 7.89 (m, 4H, H-2′,6′, H-2″,6″), 7.19 (d, *J* = 8.8 Hz, 2H, H-3″,5″), and 4.94 (s, 2H, H-3) ppm. ^13^C NMR (151 MHz, DMSO-d_6_): δ 191.3 (C-4), 166.9 (C-2), 162.6 (C-4″), 144.5 (C-4′), 142.6 (C-1′), 131.7 (C-2″,6″), 130.2 (C-1″), 125.0 (C-3′,5′), 119.3 (C-2′,6′), 115.2 (C-3″,5″), and 67.0 (C-3) ppm. 

*2-(4-Formylphenoxy)-N-[2-(trifluoromethyl)phenyl]acetamide* (**4e**). White needles. Yield: 284 mg (54%). Mp. 90–91 °C (*n*Hex/CH_2_Cl_2_). IR ν_max_ 3420, 1707, 1592, 1292, 1098, 828, 761, 643, 505 cm^−1^. ^1^H NMR (600 MHz, DMSO-d_6_): δ 9.90 (s, 1H, H-4), 9.82 (s, 1H, H-1), 7.91 (d, *J* = 8.7 Hz, 2H, H-3″,5″), 7.77 (d, *J* = 8.0 Hz, 1H, H-3′), 7.71 (t, *J* = 7.7 Hz, 1H, H-5′), 7.63 (d, *J* = 8.0 Hz, 1H, H-6′), 7.49 (t, *J* = 7.6 Hz, 1H, H-4′), 7.19 (d, *J* = 8.7 Hz, 2H, H-2″,6″), and 4.90 (s, 2H, H-3) ppm. ^13^C NMR (151 MHz, DMSO-d_6_): δ 191.4 (C-4), 167.0 (C-2), 162.4 (C-1″), 134.6 (C-1′), 133.2 (C-5′), 131.7 (C-3″,5″), 130.3 (C-4″), 129.4 (C-6′), 127.0 (C-4′), 126.4 (C-3′), 123.6 (q, *J* = 273.4 Hz, CF_3_), 115.2 (C-2″,6″), and 66.9 (C-3) ppm. HRMS: *m*/*z* [M + H]^+^ for C_16_H_12_F_3_NO_3_ calc. 324.0842; exp. 324.08447.

*2-(4-Formylphenoxy)-N-[3-(trifluoromethyl)phenyl]acetamide*(**4f**). White solid. Yield: 415 mg (78%). Mp. 120–121 °C (*n*Hex/EtOAc). IR ν_max_ 3270, 3114, 1721, 1678, 1602, 1564, 1446, 1254, 1193, 1106, 827, 793, 697, 509 cm^−1^. ^1^H NMR (600 MHz, DMSO-d_6_): δ 10.49 (s, 1H, H-1), 9.89 (s, 1H, H-4), 8.11 (br s, 1H, H-2′), 7.90 (d, *J* = 8.8 Hz, 2H, H-3″,5″), 7.86 (d, *J* = 8.3 Hz, 1H, H-6′), 7.58 (t, *J* = 8.0 Hz, 1H, H-5′), 7.45 (d, *J* = 7.8 Hz, 1H, H-4′), 7.20 (d, *J* = 8.8 Hz, 2H, H-2″,6″), and 4.90 (s, 2H, H-3) ppm. ^13^C NMR (151 MHz, DMSO-d_6_): δ 191.4 (C-4), 166.6 (C-2), 162.6 (C-1″), 139.1 (C-1′), 131.7 (C-3″,5″), 130.2 (C-5′), 130.1 (C-4″), 129.5 (q, *J* = 31.6 Hz, C-3′), 124.1 (q, *J* = 272.3 Hz, CF_3_), 123.2 (C-6′), 120.1 (q, *J* = 3.9 Hz, C-4′), 115.7 (q, *J* = 4.1 Hz, C-2′), 115.2 (C-2″,6″), and 67.0 (C-3) ppm. HRMS: *m*/*z* [M + H]^+^ for C_16_H_12_F_3_NO_3_ calc. 324.0842; exp. 324.08441.

*N-[3,5-Bis(trifluoromethyl)phenyl]-2-(4-formylphenoxy)acetamide* (**4g**). White needles. Yield: 456 mg (71%). Mp. 139–141 °C (*n*Hex/EtOAc). IR ν_max_ 3403, 3046, 1690, 1604, 1378, 1278, 1239, 1161, 1125, 1109, 886, 702 cm^−1^. ^1^H NMR (600 MHz, DMSO-d_6_): δ 10.78 (s, 1H, H-1), 9.89 (s, 1H, H-4), 8.35 (br s, 2H, H-2′,6′), 7.90 (d, *J* = 8.7 Hz, 2H, H-3″,5″), 7.82 (br s, 1H, H-4′), 7.22 (d, *J* = 8.7 Hz, 2H, H-2″,6″), and 4.93 (s, 2H, H-3) ppm. ^13^C NMR (151 MHz, DMSO-d_6_): δ 191.4 (C-4), 167.2 (C-2), 162.5 (C-1″), 140.2 (C-1′), 131.8 (C-3″,5″), 130.8 (q, *J* = 32.9 Hz, C-3′,5′), 130.3 (C-4″), 123.2 (q, *J* = 272.7 Hz, CF_3_), 119.5 (C-2′,6′), 116.7 (C-4′), 115.3 (C-2″,6″), and 66.9 (C-3) ppm. HRMS: *m*/*z* [M + H]^+^ for C_17_H_11_F_6_NO_3_ calc. 392.07159; exp. 392.0719.

*2-(4-Formyl-2-methoxyphenoxy)-N-phenylacetamide* (**5a**). White solid. Yield: 221 mg (59%). Mp. 152–154 °C (CH_2_Cl_2_/MeOH, 148–149 °C [55]). ^1^H NMR (400 MHz, DMSO-d_6_): δ 10.20 (s, 1H, H-1), 9.85 (s, 1H, H-4), 7.60 (dd, *J* = 8.6, 1.3 Hz, 2H, H-2′,6′), 7.54 (dd, *J* = 8.3, 1.9 Hz, 1H, H-6″), 7.45 (d, *J* = 1.9 Hz, 1H, H-2″), 7.32 (t, *J* = 8.5 Hz, 2H, H-3′,5′), 7.12 (d, *J* = 8.3 Hz, 1H, H-5″), 7.08 (t, *J* = 7.4 Hz, 1H, H-4′), 4.85 (s, 2H, H-3), and 3.88 (s, 3H, OCH_3_) ppm.

*N-(3,5-Dimethoxyphenyl)-2-(4-formyl-2-methoxyphenoxy)acetamide* (**5b**). White needles. Yield: 390 mg (86%). Mp. 169 °C (*n*Hex/CHCl_3_). IR ν_max_ 3394, 3073, 2911, 1677, 1611, 1449, 1417, 1286, 1152, 1131, 1033, 835, 803, 646, 574, 509 cm^−1^. ^1^H NMR (600 MHz, DMSO-d_6_): δ 10.13 (s, 1H, H-1), 9.85 (s, 1H, H-4), 7.54 (dd, *J* = 8.2, 1.9 Hz, 1H, H-5″), 7.45 (d, *J* = 1.9 Hz, 1H, H-3″), 7.11 (d, *J* = 8.2 Hz, 1H, H-6″), 6.86 (d, *J* = 2.3 Hz, 2H, H-2′,6′), 6.24 (t, *J* = 2.3 Hz, 1H, H-4′), 4.83 (s, 2H, H-3), 3.88 (s, 3H, OCH_3_), and 3.71 (s, 6H, 2 × OCH_3_) ppm. ^13^C NMR (151 MHz, DMSO-d_6_): δ 191.4 (C-4), 165.9 (C-2), 160.5 (C-3′,5′), 152.8 (C-1″), 149.3 (C-2″), 140.0 (C-1′), 130.3 (C-4″), 125.6 (C-5″), 112.8 (C-6″), 110.1 (C-3″), 97.7 (C-2′,6′), 95.7 (C-4′), 67.6 (C-3), 55.6 (OCH_3_), and 55.1 (OCH_3_) ppm. HRMS: *m*/*z* [M + H]^+^ for C_18_H_19_NO_6_ calc. 346.12851; exp. 346.1292.

*2-(4-Formyl-2-methoxyphenoxy)-N-(3,4,5-trimethoxyphenyl)acetamide* (**5c**). White solid. Yield: 331 mg (67%). Mp. 158–159 °C (CH_2_Cl_2_/MeOH). IR ν_max_ 3256, 2934, 2837, 1678, 1589, 1509, 1226, 1129, 1031, 971, 813, 782 cm^−1^. ^1^H NMR (600 MHz, DMSO-d_6_): δ 10.12 (s, 1H, H-1), 9.85 (s, 1H, H-4), 7.54 (dd, *J* = 8.3, 1.9 Hz, 1H, H-5″), 7.45 (d, *J* = 1.9 Hz, 1H, H-3″), 7.12 (d, *J* = 8.3 Hz, 1H, H-6″), 7.00 (s, 2H, H-2′,6′), 4.83 (s, 2H, H-3), 3.88 (s, 3H, OCH_3_), 3.73 (s, 6H, 2 × OCH_3_), and 3.62 (s, 3H, OCH_3_) ppm. ^13^C NMR (151 MHz, DMSO-d_6_): δ 191.4 (C-4), 165.7 (C-2), 152.8 (C-1″), 152.7 (C-3′,5′), 149.3 (C-2″), 134.5 (C-1′), 133.7 (C-4′), 130.3 (C-4″), 125.6 (C-5″), 112.8 (C-6″), 110.1 (C-3″), 97.1 (C-2′,6′), 67.6 (C-3), 60.1 (OCH_3_), 55.7 (OCH_3_), and 55.6 (OCH_3_) ppm. HRMS: *m*/*z* [M + H]^+^ for C_19_H_21_NO_7_ calc. 376.13908; exp. 376.1391.

*2-(4-Formyl-2-methoxyphenoxy)-N-(4-nitrophenyl)acetamide* (**5d**). Light yellow solid. Yield: 364 mg (84%). Mp. 161–164 °C (CHCl_3_/*n*Hex). ^1^H NMR (400 MHz, DMSO-d_6_): δ 9.85 (s, 1H, H-4), 8.24 (d, *J* = 9.3 Hz, 2H, H-3′,5′), 7.85 (d, *J* = 9.3 Hz, 2H, H-2′,6′), 7.53 (dd, *J* = 8.3, 1.9 Hz, 1H, H-6″), 7.45 (d, *J* = 1.9 Hz, 1H, H-2″), 7.13 (d, *J* = 8.3 Hz, 1H, H-5″), 4.94 (s, 2H, H-3), and 3.88 (s, 3H, OCH_3_) ppm.

*2-(4-Formyl-2-methoxyphenoxy)-N-(2-(trifluoromethyl)phenyl)acetamide* (**5e**). White solid. Yield: 384 mg (83%). Mp. 152–153 °C (EtOAc/*n*Hex, 155–156 °C [55]). ^1^H NMR (400 MHz, DMSO-d_6_): δ 9.87 (s, 1H, H-4), 9.63 (s, 1H, H-1), 7.81 (d, *J* = 8.2 Hz, 1H, H-6′), 7.77 (d, *J* = 7.4 Hz, 1H, H-3′), 7.71 (t, *J* = 7.7 Hz, 1H, H-5′), 7.57 (dd, *J* = 8.3, 1.9 Hz, 1H, H-6″), 7.46 (m, 2H, H-4′, H-2″), 7.17 (d, *J* = 8.3 Hz, 1H, H-5″), 4.91 (s, 2H, H-3), and 3.88 (s, 3H, OCH_3_) ppm.

*2-(4-Formyl-2-methoxyphenoxy)-N-(3-(trifluoromethyl)phenyl)acetamide* (**5f**). White solid. Yield: 381 mg (82%). Mp. 123–124 °C (EtOAc/*n*Hex, 120–121 °C [55]). ^1^H NMR (400 MHz, DMSO-d_6_): δ 10.57 (s, 1H, H-1), 9.85 (s, 1H, H-4), 8.10 (s, 1H, H-2′), 7.81 (dd, *J* = 8.1, 2.0 Hz, 1H, H-6′), 7.58 (t, *J* = 8.0 Hz, 1H, H-5′), 7.54 (dd, *J* = 8.3, 1.9 Hz, 1H, H-6″), 7.44 (m, 2H, H-4′, H-2″), 7.13 (d, *J* = 8.3 Hz, 1H, H-5″), 4.90 (s, 2H, H-3), and 3.88 (s, 3H, OCH_3_) ppm.

*N-[3,5-Bis(trifluoromethyl)phenyl]-2-(4-formyl-2-methoxyphenoxy)acetamide* (**5g**). White solid. Yield: 380 mg (69%). Mp. 164–165 °C (MeOH/Et_2_O). IR ν_max_ 3390, 3047, 1702, 1682, 1590, 1543, 1510, 1377, 1268, 1122, 887, 812, 680 cm^−1^. ^1^H NMR (600 MHz, DMSO-d_6_): δ 10.86 (s, 1H, H-1), 9.86 (s, 1H, H-4), 8.30 (br s, 2H, H-2′,6′), 7.81 (br s, 1H, H-4′), 7.53 (dd, *J* = 8.3, 1.9 Hz, 1H, H-5″), 7.46 (d, *J* = 1.9 Hz, 1H, H-3″), 7.16 (d, *J* = 8.3 Hz, 1H, H-6″), 4.93 (s, 2H, H-3), and 3.88 (s, 3H, OCH_3_) ppm. ^13^C NMR (151 MHz, DMSO-d_6_): δ 191.4 (C-4), 167.3 (C-2), 152.6 (C-1″), 149.4 (C-2″), 140.3 (C-1′), 130.8 (q, *J* = 32.8 Hz, C-3′,5′), 130.5 (C-4″), 125.5 (C-5″), 123.2 (q, *J* = 272.7 Hz, CF_3_), 119.2 (br s, C-2′,6′), 116.6 (C-4′), 113.1 (C-6″), 110.2 (C-3″), 67.5 (C-3), and 55.7 (OCH_3_) ppm. HRMS: *m*/*z* [M + H]^+^ for C_18_H_13_F_6_NO_4_ calc. 422.08215; exp. 422.0821.

### 3.5. General Synthetic Procedure for Compounds **6a**–**d**

To a suspension of **4a**–**d** (100 mg) in dry toluene (3 mL), thiazolidine-2,4-dione (**9**, 1.0 equiv.) and 3 drops of glacial acetic acid and piperidine were added. The reaction mixture was stirred at 110–120 °C for 6–7 h. The course of the reaction was monitored by TLC (DCM:MeOH, *v*/*v* 9:1). After the completion of the reaction, the reaction mixture was cooled to room temperature, and the formed precipitate was filtered, washed with a small amount of toluene, and dried.

*2-(4-{[(5Z)-2,4-Dioxo-1,3-thiazolidin-5-ylidene]methyl}phenoxy)-N-phenylacetamide* (**6a**). Yellow solid. Yield: 112 mg (81%). Mp. 257–259 °C (toluene), 218–221 °C [5]. ^1^H NMR (600 MHz, DMSO-d_6_): δ 12.52 (s, 1H, H-7), 10.14 (s, 1H, H-1), 7.75 (s, 1H, H-4), 7.62 (d, *J* = 7.3 Hz, 2H, H-2′,6′), 7.58 (d, *J* = 8.9 Hz, 2H, H-2″,6″), 7.32 (t, *J* = 8.5 Hz, 2H, H-3′,5′), 7.15 (d, *J* = 8.9 Hz, 2H, H-3″,5″), 7.08 (t, *J* = 7.4 Hz, 1H, H-4′), and 4.80 (s, 2H, H-3) ppm.

*N-(3,5-Dimethoxyphenyl)-2-(4-{[(5Z)-2,4-dioxo-1,3-thiazolidin-5-ylidene]methyl}phenoxy)acetamide* (**6b**). Pale yellow solid. Yield: 154 mg (87%). Mp. 254–256 °C (toluene). IR ν_max_ 3396, 3105, 2937, 2840, 1682, 1595, 1549, 1426, 1311, 1260, 1160, 1012, 854 cm^−1^. ^1^H NMR (600 MHz, DMSO-d_6_): δ 10.09 (s, 1H, H-1), 7.74 (s, 1H, H-4), 7.58 (d, *J* = 8.9 Hz, 2H, H-2″,6″), 7.14 (d, *J* = 8.9 Hz, 2H, H-3″,5″), 6.89 (d, *J* = 2.3 Hz, 2H, H-2′,6′), 6.24 (t, *J* = 2.3 Hz, 1H, H-4′), 4.78 (s, 2H, H-3), and 3.71 (s, 6H, 2 × OCH_3_) ppm. ^13^C NMR (151 MHz, DMSO-d_6_): δ 168.1 (C-8), 167.7 (C-6), 166.2 (C-2), 160.5 (C-3′,5′), 159.5 (C-4″), 140.0 (C-1′), 132.0 (C-2″,6″), 131.6 (C-4), 126.2 (C-1″), 121.0 (C-5), 115.6 (C-3″,5″), 98.0 (C-2′,6′), 95.8 (C-4′), 67.1 (C-3), and 55.2 (OCH_3_) ppm.

*2-(4-{[(5Z)-2,4-Dioxo-1,3-thiazolidin-5-ylidene]methyl}phenoxy)-N-(3,4,5-trimethoxyphenyl)acetamide* (**6c**). Yellow solid. Yield: 145 mg (76%). Mp. 202–204 °C (toluene). IR ν_max_ 3340, 2997, 2837, 1737, 1697, 1667, 1597, 1504, 1412, 1230, 1125, 1014, 999, 822, 689 cm^−1^. ^1^H NMR (600 MHz, DMSO-d_6_): δ 10.06 (s, 1H, H-1), 7.73 (s, 1H, H-4), 7.58 (d, *J* = 8.9 Hz, 2H, H-2″,6″), 7.15 (d, *J* = 8.9 Hz, 2H, H-3″,5″), 7.03 (s, 2H, H-2′,6′), 4.77 (s, 2H, H-3), 3.73 (s, 6H, OCH_3_), and 3.61 (s, 3H, OCH_3_) ppm. ^13^C NMR (151 MHz, DMSO-d_6_): δ 168.4 (C-6), 168.4 (C-8), 166.0 (C-2), 159.4 (C-4″), 152.8 (C-3′,5′), 134.5 (C-1′), 133.8 (C-4′), 132.0 (C-2″,6″), 131.2 (C-4), 126.3 (C-1″), 121.5 (C-5), 115.6 (C-3″,5″), 97.5 (C-2′,6′), 67.1 (C-3), 60.2 (OCH_3_), and 55.8 (OCH_3_) ppm. HRMS: *m*/*z* [M + H]^+^ for C_21_H_20_N_2_O_7_S calc. 445.1064; calc. 445.1066.

*2-(4-{[(5Z)-2,4-Dioxo-1,3-thiazolidin-5-ylidene]methyl}phenoxy)-N-(4-nitrophenyl)acetamide* (**6d**). Pale yellow solid. Yield: 148 mg (87%). Mp. 292–294 °C (toluene). IR ν_max_ 3372, 3041, 1737, 1692, 1591, 1508, 1342, 1256, 1180, 1150, 852, 689 cm^−1^. ^1^H NMR (600 MHz, DMSO-d_6_): δ 12.52 (br s, 1H, H-7), 10.76 (s, 1H, H-1), 8.24 (d, *J* = 9.2 Hz, 2H, H-3′,5′), 7.89 (d, *J* = 9.2 Hz, 2H, H-2′,6′), 7.73 (s, 1H, H-4), 7.58 (d, *J* = 8.9 Hz, 2H, H-2″,6″), 7.16 (d, *J* = 8.9 Hz, 2H, H-3″,5″), and 4.89 (s, 2H, H-3) ppm. ^13^C NMR (151 MHz, DMSO-d_6_): δ 168.3 (C-6), 168.1 (C-8), 167.1 (C-2), 159.3 (C-4″), 144.5 (C-1′), 142.6 (C-4′), 132.0 (C-2″,6″), 131.2 (C-4), 126.3 (C-1″), 125.0 (C-3′,5′), 121.4 (C-5), 119.3 (C-2′,6′), 115.6 (C-3″,5″), and 67.0 (C-3) ppm. HRMS: *m*/*z* [M + H]^+^ for C_18_H_13_N_3_O_6_S calc. 400.05978; exp. 400.0596.

### 3.6. General Synthetic Procedure for Compounds **10** and **11**

To a suspension of the potassium salt of thiazolidine-2,4-dione (300 mg, 1.10 mmol) in *N*,*N*-dimethylformamide (5 mL), 9-(bromomethyl)acridine (171 mg, 1.10 mmol) or 4-(bromomethyl)acridine (171 mg, 1.10 mmol) was added. The reaction mixture was stirred at 100 °C for 1 h. The course of the reaction was monitored by TLC (*n*Hex:EtOAc, *v*/*v* 2:1). After the completion of the reaction, the reaction mixture was poured into crushed ice, and the formed precipitate was filtered, washed with water, and dried.

*3-(Acridin-9-ylmethyl)thiazolidine-2,4-dione* (**10**). Yellow solid. Yield: 566 mg (95%). Mp. 196–198 °C (chloroform, 196–197 °C [16]). ^1^H NMR (400 MHz, DMSO-d_6_): δ 8.42 (d, *J* = 8.3 Hz, 2H, H-1′,8′), 8.17 (d, *J* = 8.1 Hz, 2H, H-4′,5′), 7.85 (ddd, *J* = 8.7, 6.5, 1.2 Hz, 2H, H-3′,6′), 7.67 (ddd, *J* = 8.9, 6.5, 1.3 Hz, 2H, H-2′,7′), 5.75 (s, 2H, H-6), and 4.22 (s, 2H, H-5) ppm. 

*3-(Acridin-4-ylmethyl)thiazolidine-2,4-dione (**11**).* Pale orange solid. Yield: 572 mg (96%). Mp. 150–151 °C (chloroform). IR ν_max_ 2990, 2921, 2359, 2341, 1741, 1661, 1374, 1306, 1154, 896 cm^−1^. ^1^H NMR (600 MHz, DMSO-d_6_): δ 9.16 (s, 1H, H-9′), 8.20 (d, *J* = 8.1 Hz, 1H, H-8′), 8.18 (d, *J* = 8.4 Hz, 1H, H-5′), 8.12 (dd, *J* = 8.5, 1.4 Hz, 1H, H-1′), 7.90 (ddd, *J* = 8.4, 6.5, 1.4 Hz, 1H, H-6′), 7.66 (ddd, *J* = 8.1, 6.6, 1.1 Hz, 1H, H-7′), 7.56 (dd, *J* = 8.4, 6.8 Hz, 1H, H-2′), 7.50 (dd, *J* = 6.8, 1.4 Hz, 1H, H-3′), 5.46 (s, 2H, H-6), and 4.38 (s, 2H, H-5) ppm. ^13^C NMR (151 MHz, DMSO-d_6_): δ 172.4 (C-4), 172.2 (C-2), 147.6 (C-10′a), 146.0 (C-4′a), 136.7 (C-9′), 132.2 (C-4′), 130.8 (C-6′), 129.2 (C-5′), 128.5 (C-8′), 128.0 (C-1′), 126.5 (C-3′), 126.1 (C-7′), 126.1 (C-8′a), 125.9 (C-9′a), 125.2 (C-2′), 41.8 (C-6), and 34.2 (C-5) ppm.

### 3.7. General Synthetic Procedure for Compounds **7**, **8**, **12**, and **13**

**Procedure 1** (for compounds **7a**–**d** and **8a**–**d**, Figure 1): To a suspension of acetamide **6a**–**d** (50 mg) in dry acetone (3 mL), anhydrous K_2_CO_3_ (1.2 equiv.), KI (0.2 equiv.), and 9-(bromomethyl)acridine (1.0 equiv.) or 4-(bromomethyl)acridine (1.0 equiv.) were added. The reaction mixture was stirred at 65–70 °C for the appropriate time (**7a**: 3 h, **7b**,**c**: 6 h, **7d**: 7 h, **8a**: 7 h, and **8b**–**d**: 6 h). The course of the reaction was monitored by TLC (DCM). After the completion of the reaction, the reaction mixture was evaporated under reduced pressure, and the crude product was suspended in water, filtered, and dried. Products **7a**–**d** and **8a**–**d** were crystallized using DMSO/MeOH.

**Procedure 2** (for compounds **7a**–**g**, **8a**–**g**, **12a**–**g**, **13a**–**g**, Figure 2): To a suspension of acetamide **4a**–**g** or **5a**–**g** (50 mg) in dry ethanol (3 mL), derivative **10** or **11** (1.0 equiv.) and 3 drops of piperidine and glacial acetic acid were added. The reaction mixture was stirred and refluxed for the appropriate time (**7a**: 1 h, **7b**,**e**–**g**: 2 h, **7c**: 3 h, **7d**: 4 h; **8a**: 2 h, **8b**–**g**: 3 h, **12a**–**e**: 5 h, **12f**: 4 h, **12g**: 3 h, **13a**,**d**–**f**: 4 h, and **13b**,**c**,**g**: 5 h). The course of the reaction was monitored by TLC (*n*Hex:EtOAc, *v*/*v* 2:1). After the reaction was completed, the reaction mixture was cooled, and the formed precipitate was filtered, washed with absolute ethanol, and dried.

*2-(4-{[(5Z)-3-[(Acridin-9-yl)methyl]-2,4-dioxo-1,3-thiazolidin-5-ylidene]methyl}phenoxy)-N-phenylacetamide* (**7a**). Pale yellow solid. Yield: 42 mg (55%, procedure 1), 84 mg (79%, procedure 2). Mp. 249–250 °C (EtOH). IR ν_max_ 3273, 3062, 1741, 1673, 1596, 1538, 1505, 1252, 1179,1058, 756, 717 cm^−1^. ^1^H NMR (600 MHz, DMSO-d_6_): δ 10.13 (s, 1H, H-1), 8.46 (d, *J* = 9.0 Hz, 2H, H-1‴,8‴), 8.18 (d, *J* = 8.7 Hz, 2H, H-4‴,5‴), 7.89 (s, 1H, H-4), 7.86 (ddd, *J* = 8.8, 6.5, 1.2 Hz, 2H, H-3‴,6‴), 7.68 (ddd, *J* = 8.8, 6.5, 1.2 Hz, 2H, H-2‴,7‴), 7.60 (d, *J* = 7.3 Hz, 2H, H-2′,6′), 7.56 (d, *J* = 9.0 Hz, 2H, H-2″,6″), 7.31 (dd, *J* = 8.6, 7.3 Hz, 2H, H-3′,5′), 7.13 (d, *J* = 9.0 Hz, 2H, H-3″,5″), 7.07 (tt, *J* = 7.3, 1.2 Hz, 1H, H-4′), 5.92 (s, 2H, H-10), and 4.79 (s, 2H, H-3) ppm. ^13^C NMR (151 MHz, DMSO-d_6_): *δ* 167.5 (C-8), 166.0 (C-2), 165.7 (C-6), 159.9 (C-4″), 148.1 (C-4‴a,10‴a), 138.3 (C-1′), 137.7 (C-9‴), 133.8 (C-4), 132.4 (C-2″,6″), 130.1 (C-3‴,6‴), 129.9 (C-4‴,5‴), 128.8 (C-3′,5′), 126.5 (C-2‴,7‴), 125.9 (C-1″), 125.2 (C-8‴a,9‴a), 124.7 (C-1‴,8‴), 123.8 (C-4′), 119.7 (C-2′,6′), 117.6 (C-5), 115.7 (C-3″,5″), 67.0 (C-3), and 38.3 (C-10) ppm. ^15^N NMR (61 MHz, DMSO-d_6_): δ −250.8 (N-1) and −211.8 (N-7) ppm. HRMS: *m*/*z* [M + H]^+^ for C_32_H_23_N_3_O_4_S calc. 546.1482; exp. 546.1489.

*2-(4-{[(5Z)-3-[(Acridin-9-yl)methyl]-2,4-dioxo-1,3-thiazolidin-5-ylidene]methyl}phenoxy)-N-(3,5-dimethoxyphenyl)acetamide* (**7b**). Pale yellow solid. Yield: 40 mg (54%, procedure 1), 73 mg (75%, procedure 2). Mp. 280–282 °C (EtOH). IR ν_max_ 3408, 2838, 1736, 1682, 1593, 1542, 1510, 1254, 1186, 1153, 1058, 748, 716 cm^−1^. ^1^H NMR (600 MHz, DMSO-d_6_): δ 10.09 (s, 1H, H-1), 8.46 (d, *J* = 9.0 Hz, 2H, H-1‴,8‴), 8.18 (d, *J* = 8.7 Hz, 2H, H-4‴,5‴), 7.90 (s, 1H, H-4), 7.86 (ddd, *J* = 8.7, 6.5, 1.2 Hz, 2H, H-3‴,6‴), 7.69 (ddd, *J* = 8.9, 6.5, 1.2 Hz, 2H, H-2‴,7‴), 7.57 (d, *J* = 9.0 Hz, 2H, H-2″,6″), 7.12 (d, *J* = 9.0 Hz, 2H, H-3″,5″), 6.87 (d, *J* = 2.3 Hz, 2H, H-2′,6′), 6.24 (t, *J* = 2.3 Hz, 1H, H-4′), 5.92 (s, 2H, H-10), 4.77 (s, 2H, H-3), and 3.70 (s, 6H, 2 × OCH_3_) ppm. ^13^C NMR (151 MHz, DMSO-d_6_): δ 167.5 (C-8), 166.1 (C-2), 165.7 (C-6), 160.5 (C-3′,5′), 159.9 (C-4″), 148.1 (C-4‴a,10‴a), 140.0 (C-1′), 137.7 (C-9‴), 133.8 (C-4), 132.4 (C-2″,6″), 130.1 (C-3‴,6‴), 129.9 (C-4‴,5‴), 126.5 (C-2‴,7‴), 125.8 (C-1″), 125.2 (C-8‴a,9‴a), 124.7 (C-1‴,8‴), 117.5 (C-5), 115.7 (C-3″,5″), 97.9 (C-2′,6′), 95.7 (C-4′), 67.0 (C-3), 55.1 (OCH_3_), and 38.3 (C-10) ppm. ^15^N (61 MHz, DMSO-d_6_): δ −250.1 (N-1) and −211.7 (N-7) ppm. HRMS: *m*/*z* [M + H]^+^ for C_34_H_27_N_3_O_6_S calc. 606.16933; exp. 606.1702.

*2-(4-{[(5Z)-3-[(Acridin-9-yl)methyl]-2,4-dioxo-1,3-thiazolidin-5-ylidene]methyl}phenoxy)-N-(3,4,5-trimethoxyphenyl)acetamide* (**7c**). Pale yellow solid. Yield: 40 mg (56%, procedure 1), 74 mg (83%, procedure 2). Mp. 248–249 °C (EtOH). IR ν_max_ 3406, 2942, 2825, 1737, 1681, 1589, 1542, 1510, 1234, 1186, 1183, 1126, 1055, 750, 716 cm^−1^. ^1^H NMR (600 MHz, DMSO-d_6_): δ 10.08 (s, 1H, H-1), 8.46 (d, *J* = 9.0 Hz, 2H, H-1‴,8‴), 8.18 (d, *J* = 8.6 Hz, 2H, H-4‴,5‴), 7.90 (s, 1H, H-4), 7.86 (ddd, *J* = 8.7, 6.5, 1.2 Hz, 2H, H-3‴,6‴), 7.68 (ddd, *J* = 8.9, 6.5, 1.2 Hz, 2H, H-2‴,7‴), 7.57 (d, *J* = 9.0 Hz, 2H, H-2″,6″), 7.13 (d, *J* = 9.0 Hz, 2H, H-3″,5″), 7.02 (s, 2H, H-2′,6′), 5.92 (s, 2H, H-10), 4.77 (s, 2H, H-3), 3.72 (s, 6H, 2 × OCH_3_), and 3.61 (s, 3H, OCH_3_) ppm. ^13^C NMR (151 MHz, DMSO-d_6_): δ 167.5 (C-8), 165.8 (C-2), 165.7 (C-6), 159.9 (C-4″), 152.7 (C-3′,5′), 148.1 (C-4‴a,10‴a), 137.7 (C-9‴), 134.5 (C-1′), 133.8 (C-4), 133.7 (C-4′), 132.4 (C-2″,6″), 130.1 (C-3‴,6‴), 129.9 (C-4‴,5‴), 126.4 (C-2‴,7‴), 125.9 (C-1″), 125.2 (C-8‴a,9‴a), 124.7 (C-1‴,8‴), 117.5 (C-5), 115.7 (C-3″,5″), 97.3 (C-2′,6′), 67.0 (C-3), 60.1 (OCH_3_), 55.7 (OCH_3_), and 38.3 (C-10) ppm. ^15^N (61 MHz, DMSO-d_6_): δ −250.9 (N-1), −211.7 (N-7), and −70.9 (N-10‴) ppm. HRMS: *m*/*z* [M + H]^+^ for C_35_H_29_N_3_O_7_S calc. 636.1799; exp. 636.1809.

*2-(4-{[(5Z)-3-[(Acridin-9-yl)methyl]-2,4-dioxo-1,3-thiazolidin-5-ylidene]methyl}phenoxy)-N-(4-nitrophenyl)acetamide* (**7d**). Pale yellow solid. Yield: 21 mg (28%, procedure 1), 70 mg (70%, procedure 2). Mp. 293–294 °C (EtOH). IR ν_max_ 3068, 1737, 1687, 1592, 1544, 1503, 1331, 1243, 1181, 1042, 750, 714 cm^−1^. ^1^H NMR (600 MHz, DMSO-d_6_): δ 10.77 (s, 1H, H-1), 8.46 (d, *J* = 9.0 Hz, 2H, H-1‴,8‴), 8.24 (d, *J* = 9.2 Hz, 2H, H-3′,5′), 8.18 (d, *J* = 8.7 Hz, 2H, H-4‴,5‴), 7.90 (s, 1H, H-4), 7.87 (d, *J* = 9.2 Hz, 2H, H-2′,6′), 7.86 (ddd, *J* = 8.7, 6.5, 1.2 Hz, 2H, H-3‴,6‴), 7.68 (ddd, *J* = 8.8, 6.5, 1.2 Hz, 2H, H-2‴,7‴), 7.57 (d, *J* = 9.0 Hz, 2H, H-2″,6″), 7.14 (d, *J* = 9.0 Hz, 2H, H-3″,5″), 5.92 (s, 2H, H-10), and 4.88 (s, 2H, H-3) ppm. ^13^C NMR (151 MHz, DMSO-d_6_): δ 167.5 (C-8), 167.1 (C-2), 165.7 (C-6), 159.7 (C-4″), 148.1 (C-4‴a,10‴a), 144.5 (C-1′), 142.6 (C-4′), 137.7 (C-9‴), 133.8 (C-4), 132.4 (C-2″,6″), 130.1 (C-3‴,6‴), 129.9 (C-4‴,5‴), 126.4 (C-2‴,7‴), 125.9 (C-1″), 125.2 (C-8‴a,9‴a), 125.0 (C-3′,5′), 124.7 (C-1‴,8‴), 119.3 (C-2′,6′), 117.6 (C-5), 115.7 (C-3″,5″), 66.9 (C-3), and 38.3 (C-10) ppm. ^15^N (61 MHz, DMSO-d_6_): δ −248.6 (N-1), −211.7 (N-7), and −70.9 (N-10‴) ppm. HRMS: *m*/*z* [M + H]^+^ for C_32_H_22_N_4_O_6_S calc. 591.13328; exp. 591.1348.

*2-(4-{[(5Z)-3-[(Acridin-9-yl)methyl]-2,4-dioxo-1,3-thiazolidin-5-ylidene]methyl}phenoxy)-N-[2-(trifluoromethyl)phenyl]acetamide* (**7e**). Pale yellow solid. Yield: 67 mg (73%). Mp. 256–257 °C (d, EtOH). IR ν_max_ 3419, 3038, 2917, 1736, 1681, 1590, 1538, 1500, 1321, 1253, 1178, 1058, 765, 750, 716 cm^−1^. ^1^H NMR (600 MHz, DMSO-d_6_): δ 9.78 (s, 1H, H-1), 8.47 (d, *J* = 8.9 Hz, 2H, H-1‴,8‴), 8.19 (d, *J* = 8.7 Hz, 2H, H-4‴,5‴), 7.91 (s, 1H, H-4), 7.86 (t, *J* = 7.6 Hz, 2H, H-3‴,6‴), 7.75 (d, *J* = 7.9 Hz, 1H, H-3′), 7.69 (m, 3H, H-5′,2‴,7‴), 7.62 (d, *J* = 8.1 Hz, 1H, H-6′), 7.58 (d, *J* = 8.5 Hz, 2H, H-2″,6″), 7.47 (t, *J* = 7.7 Hz, 1H, H-4′), 7.14 (d, *J* = 8.4 Hz, 2H, H-3″,5″), 5.93 (s, 2H, H-10), and 4.84 (s, 2H, H-3) ppm. ^13^C NMR (151 MHz, DMSO-d_6_): δ 167.4 (C-8), 167.1 (C-2), 165.7 (C-6), 159.5 (C-4″), 148.1 (C-4‴a,10‴a), 137.6 (C-9‴), 134.6 (C-1′), 133.8 (C-4), 133.2 (C-5′), 132.3 (C-2″,6″), 130.0 (C-3‴,6‴), 129.9 (C-4‴,5‴), 129.3 (C-6′), 126.9 (C-4′), 126.4 (C-2‴,7‴), 126.4 (C-3′), 126.0 (C-1″), 125.1 (C-8‴a,9‴a), 124.7 (C-1‴,8‴), 124.5 (C-2′), 117.7 (C-5), 115.7 (C-3″,5″), 66.8 (C-3), and 38.3 (C-10) ppm. ^15^N (61 MHz, DMSO-d_6_): δ −259.5 (N-1) and −211.7 (N-7) ppm.

*2-(4-{[(5Z)-3-[(Acridin-9-yl)methyl]-2,4-dioxo-1,3-thiazolidin-5-ylidene]methyl}phenoxy)-N-[3-(trifluoromethyl)phenyl]acetamide* (**7f**). Pale yellow solid. Yield: 70 mg (76%). Mp. 250–251 °C (d, EtOH). IR ν_max_ 3585, 3401, 3068, 2919, 1740, 1686, 1595, 1549, 1509, 1338, 1257, 1184, 1049, 765, 750, 715 cm^−1^. ^1^H NMR (600 MHz, DMSO-d_6_): δ 10.47 (s, 1H, H-1), 8.46 (d, *J* = 8.8 Hz, 2H, H-1‴,8‴), 8.18 (d, *J* = 8.6 Hz, 2H, H-4‴,5‴), 8.10 (s, 1H, H-2′), 7.90 (s, 1H, H-4), 7.85 (m, 3H, H-6′, H-3‴,6‴), 7.68 (ddd, *J* = 8.9, 6.5, 1.2 Hz, 2H, H-2‴,7‴), 7.57 (d, *J* = 8.5 Hz, 2H, H-2″,6″), 7.56 (m, 1H, H-5′), 7.44 (d, *J* = 7.8 Hz, 1H, H-4′), 7.15 (d, *J* = 9.0 Hz, 2H, H-3″,5″), 5.92 (s, 2H, H-10), and 4.84 (s, 2H, H-3) ppm. ^13^C NMR (151 MHz, DMSO-d_6_): δ 167.4 (C-8), 166.7 (C-2), 165.7 (C-6), 159.7 (C-4″), 148.1 (C-4‴a,10‴a), 139.1 (C-1′), 137.6 (C-9‴), 133.8 (C-4), 132.3 (C-2″,6″), 130.1 (C-3‴,6‴), 129.9 (C-4‴,5‴), 129.9 (C-5′), 129.4 (q, *J* = 32.0 Hz, C-3′), 126.8 (C-2‴,7‴), 125.9 (C-1″), 125.1 (C-8‴a,9‴a), 124.7 (C-1‴,8‴), 124.1 (q, *J* = 272.0 Hz, CF_3_), 123.2 (C-6′), 120.1 (br s, C-4′), 117.6 (C-5), 115.7 (C-3″,5″), 115.7 (br s, C-2′), 66.9 (C-3), and 38.3 (C-10) ppm. ^15^N (61 MHz, DMSO-d_6_): δ −251.4 (N-1) and −211.7 (N-7) ppm.

*2-(4-{[(5Z)-3-[(Acridin-9-yl)methyl]-2,4-dioxo-1,3-thiazolidin-5-ylidene]methyl}phenoxy)-N-[3,5-bis(trifluoromethyl)phenyl]acetamide* (**7g**). Pale yellow solid. Yield: 67 mg (76%). Mp. 227–228 °C (d, EtOH). IR ν_max_ 3401, 3284, 1730, 1715, 1682, 1596, 1544, 1509, 1385, 1372, 1281, 1181, 1060, 756, 714 cm^−1^. ^1^H NMR (600 MHz, DMSO-d_6_): δ 10.75 (s, 1H, H-1), 8.46 (d, *J* = 8.7 Hz, 2H, H-1‴,8‴), 8.33 (s, 2H, H-2′,6′), 8.14 (d, *J* = 8.8 Hz, 2H, H-4‴,5‴), 7.90 (s, 1H, H-4), 7.86 (ddd, *J* = 8.7, 6.6, 1.2 Hz, 2H, H-3‴,6‴), 7.81 (s, 1H, H-4′), 7.68 (ddd, *J* = 8.9, 6.5, 1.2 Hz, 2H, H-2‴,7‴), 7.58 (d, *J* = 9.0 Hz, 2H, H-2″,6″), 7.17 (d, *J* = 9.0 Hz, 2H, H-3″,5″), 5.92 (s, 2H, H-10), and 4.87 (s, 2H, H-3) ppm. ^13^C NMR (151 MHz, DMSO-d_6_): δ 167.4 (C-8), 167.3 (C-2), 165.7 (C-6), 159.6 (C-4″), 148.1 (C-4‴a,10‴a), 140.2 (C-1′), 137.6 (C-9‴), 133.7 (C-4), 132.3 (C-2″,6″), 130.8 (q, *J* = 32.0 Hz, C-3′,5′), 130.0 (C-3‴,6‴), 129.9 (C-4‴,5‴), 126.4 (C-2‴,7‴), 126.0 (C-1″), 125.1 (C-8‴a,9‴a), 124.7 (C-1‴,8‴), 123.2 (q, *J* = 272.0 Hz, CF_3_), 119.5 (br s, C-2′,6′), 117.7 (C-5), 116.7 (br s, C-4′), 115.7 (C-3″,5″), 66.9 (C-3), and 38.3 (C-10) ppm. ^15^N (61 MHz, DMSO-d_6_): δ −251.5 (N-1) and −211.7 (N-7) ppm.

*2-(4-{[(5Z)-3-[(Acridin-4-yl)methyl]-2,4-dioxo-1,3-thiazolidin-5-ylidene]methyl}phenoxy)-N-phenylacetamide* (**8a**). Pale yellow solid. Yield: 74 mg (69%). Mp. 258–259 °C (d, EtOH). IR ν_max_ 3326, 3037, 2906, 1742, 1673, 1595, 1528, 1499, 1245, 1178, 1139, 1057, 747, 738, 716 cm^−1^. ^1^H NMR (600 MHz, DMSO-d_6_): δ 10.16 (s, 1H, H-1), 9.17 (s, 1H, H-9‴), 8.20 (d, *J* = 8.4 Hz, 1H, H-8‴), 8.17 (dd, *J* = 8.8, 1.0 Hz, 1H, H-5‴), 8.13 (dd, *J* = 7.8, 1.8 Hz, 1H, H-1‴), 7.97 (s, 1H, H-4), 7.89 (ddd, *J* = 8.7, 6.6, 1.5 Hz, 1H, H-6‴), 7.66 (m, 3H, H-2″,6″, H-7‴), 7.63 (d, *J* = 7.4 Hz, 2H, H-2′,6′), 7.55 (m, 2H, H-2‴, H-3‴), 7.33 (dd, *J* = 8.5, 7.4 Hz, 2H, H-3′,5′), 7.19 (d, *J* = 8.8 Hz, 2H, H-3″,5″), 7.09 (tt, *J* = 7.4, 1.2 Hz, 1H, H-4′), 5.60 (s, 2H, H-10), and 4.83 (s, 2H, H-3) ppm. ^13^C NMR (151 MHz, DMSO-d_6_): δ 167.5 (C-8), 166.0 (C-2), 165.9 (C-6), 159.8 (C-4″), 147.7 (C-10‴a), 146.1 (C-4‴a), 138.3 (C-1′), 136.8 (C-9‴), 133.0 (C-4), 132.2 (C-2″,6″), 132.1 (C-4‴), 130.9 (C-6‴), 129.1 (C-5‴), 128.8 (C-3′,5′), 128.5 (C-8‴), 128.1 (C-1‴), 127.0 (C-3‴), 126.2 (C-7‴), 126.2 (C-8‴a), 126.1 (C-1″), 126.0 (C-9‴a), 125.3 (C-2‴), 123.7 (C-4′), 119.7 (C-2′,6′), 118.6 (C-5), 115.7 (C-3″,5″), 67.0 (C-3), and 42.3 (C-10) ppm. ^15^N (61 MHz, DMSO-d_6_): δ −250.8 (N-1) and −215.0 (N-7) ppm. HRMS: *m*/*z* [M + H]^+^ for C_32_H_23_N_3_O_4_S calc. 546.1482; exp. 546.1484.

*2-(4-{[(5Z)-3-[(Acridin-4-yl)methyl]-2,4-dioxo-1,3-thiazolidin-5-ylidene]methyl}phenoxy)-N-(3,5-dimethoxyphenyl)acetamide* (**8b**). Pale yellow solid. Yield: 22 mg (30%, procedure 1), 60 mg (62%, procedure 2). Mp. 168–169 °C (d, EtOH). IR ν_max_ 3340, 2917, 2848, 1737, 1677, 1598, 1547, 1507, 1247, 1180, 1146, 1058, 739 cm^−1^. ^1^H NMR (600 MHz, DMSO-d_6_): δ 10.10 (s, 1H, H-1), 9.17 (s, 1H, H-9‴), 8.20 (d, *J* = 8.4 Hz, 1H, H-8‴), 8.17 (dd, *J* = 8.8, 1.0 Hz, 1H, H-5‴), 8.13 (dd, *J* = 7.8, 1.8 Hz, 1H, H-1‴), 7.97 (s, 1H, H-4), 7.89 (ddd, *J* = 8.7, 6.6, 1.5 Hz, 1H, H-6‴), 7.66 (m, 3H, H-2″,6″, H-7‴), 7.63 (d, *J* = 2.3 Hz, 2H, H-2′,6′), 7.55 (m, 2H, H-2‴,3‴), 7.18 (d, *J* = 8.8 Hz, 2H, H-3″,5″), 6.25 (t, *J* = 2.3 Hz, 1H, H-4′), 5.60 (s, 2H, H-10), 4.80 (s, 2H, H-3), and 3.71 (s, 6H, 2 × OCH_3_) ppm. ^13^C NMR (151 MHz, DMSO-d_6_): δ 167.5 (C-8), 166.1 (C-2), 165.9 (C-6), 160.5 (C-3′,5′), 159.7 (C-4″), 147.7 (C-10‴a), 146.1 (C-4‴a), 140.0 (C-1′), 136.8 (C-9‴), 133.0 (C-4), 132.2 (C-2″,6″), 132.1 (C-4‴), 130.9 (C-6‴), 129.1 (C-5‴), 128.5 (C-8‴), 128.1 (C-1‴), 127.0 (C-3‴), 126.2 (C-7‴), 126.2 (C-8‴a), 126.1 (C-1″), 126.0 (C-9‴a), 125.3 (C-2‴), 118.6 (C-5), 115.7 (C-3″,5″), 97.9 (C-2′,6′), 95.7 (C-4′), 67.0 (C-3), 55.1 (OCH_3_), and 42.3 (C-10) ppm. ^15^N (61 MHz, DMSO-d_6_): δ −250.1 (N-1) and −215.0 (N-7) ppm. HRMS: *m*/*z* [M + H]^+^ for C_34_H_27_N_3_O_6_S calc. 606.16933; exp. 606.16950.

*2-(4-{[(5Z)-3-[(Acridin-4-yl)methyl]-2,4-dioxo-1,3-thiazolidin-5-ylidene]methyl}phenoxy)-N-(3,4,5-trimethoxyphenyl)acetamide* (**8c**). Pale yellow solid. Yield: 12 mg (16%, procedure 1), 67 mg (75%, procedure 2). Mp. 193–194 °C (d, EtOH). IR ν_max_ 3278, 2936, 2839, 1742, 1677, 1598, 1526, 1506, 1251, 1231, 1180, 1133, 1052, 740 cm^−1^. ^1^H NMR (600 MHz, DMSO-d_6_): δ 10.08 (s, 1H, H-1), 9.17 (s, 1H, H-9‴), 8.20 (dd, *J* = 8.4, 1.0 Hz, 1H, H-8‴), 8.17 (dd, *J* = 8.8, 1.0 Hz, 1H, H-5‴), 8.13 (dd, *J* = 7.9, 2.0 Hz, 1H, H-1‴), 7.97 (s, 1H, H-4), 7.89 (ddd, *J* = 8.7, 6.6, 1.5 Hz, 1H, H-6‴), 7.66 (m, 3H, H-2″,6″, H-7‴), 7.55 (m, 2H, H-2‴, H-3‴), 7.19 (d, *J* = 8.8 Hz, 2H, H-3″,5″), 7.04 (s, 2H, H-2′,6′), 5.60 (s, 2H, H-10), 4.80 (s, 2H, H-3), 3.74 (s, 6H, 2 × OCH_3_), and 3.62 (s, 3H, OCH_3_) ppm. ^13^C NMR (151 MHz, DMSO-d_6_): δ 167.5 (C-8), 165.9 (C-2), 165.9 (C-6), 159.7 (C-4″), 152.7 (C-3′,5′), 147.7 (C-10‴a), 146.1 (C-4‴a), 136.8 (C-9‴), 134.5 (C-1′), 133.7 (C-4′), 133.0 (C-4), 132.2 (C-2″,6″), 132.1 (C-4‴), 130.9 (C-6‴), 129.1 (C-5‴), 128.5 (C-8‴), 128.1 (C-1‴), 127.0 (C-3‴), 126.2 (C-7‴), 126.2 (C-8‴a), 126.1 (C-1″), 126.0 (C-9‴a), 125.4 (C-2‴), 118.6 (C-5), 115.7 (C-3″,5″), 97.4 (C-2′,6′), 67.0 (C-3), 60.1 (OCH_3_), 55.7 (OCH_3_), and 42.4 (C-10) ppm. HRMS: *m*/*z* [M + H]^+^ for C_35_H_29_N_3_O_7_S calc. 636.1799; exp. 636.1806.

*2-(4-{[(5Z)-3-[(Acridin-4-yl)methyl]-2,4-dioxo-1,3-thiazolidin-5-ylidene]methyl}phenoxy)-N-(4-nitrophenyl)acetamide* (**8d**). Pale yellow solid. Yield: 28 mg (38%, procedure 1), 63 mg (63%, procedure 2). Mp. > 300 °C (d, EtOH). IR ν_max_ 3325, 3028, 1745, 1674, 1597, 1535, 1502, 1345, 1246, 1180, 1058, 743, 712 cm^−1^. ^1^H NMR (600 MHz, DMSO-d_6_): δ 10.78 (s, 1H, H-1), 9.17 (s, 1H, H-9‴), 8.25 (d, *J* = 9.3 Hz, 2H, H-3′,5′), 8.21 (d, *J* = 8.4 Hz, 1H, H-8‴), 8.17 (d, *J* = 8.6 Hz, 1H, H-5‴), 8.13 (d, *J* = 7.9 Hz, 1H, H-1‴), 7.97 (s, 1H, H-4), 7.89 (d, *J* = 9.3 Hz, 2H, H-2′,6′), 7.89 (m, 1H, H-6‴), 7.67 (m, 3H, H-2″,6″, H-7‴), 7.56 (m, 2H, H-2‴, H-3‴), 7.20 (d, *J* = 8.8 Hz, 2H, H-3″,5″), 5.60 (s, 2H, H-10), and 4.80 (s, 2H, H-3) ppm. ^13^C NMR (151 MHz, DMSO-d_6_): δ 167.5 (C-8), 167.1 (C-2), 165.9 (C-6), 159.6 (C-4″), 147.7 (C-10‴a), 146.1 (C-4‴a), 144.5 (C-1′), 142.6 (C-4′), 136.8 (C-9‴), 133.0 (C-4), 132.2 (C-2″,6″), 132.1 (C-4‴), 130.9 (C-6‴), 129.1 (C-5‴), 128.5 (C-8‴), 128.1 (C-1‴), 127.0 (C-3‴), 126.2 (C-7‴), 126.2 (C-1″), 126.0 (C-8‴a), 126.0 (C-9‴a), 125.3 (C-2‴), 125.0 (C-3′,5′), 119.3 (C-2′,6′), 118.7 (C-5), 115.7 (C-3″,5″), 67.0 (C-3), and 42.3 (C-10) ppm. HRMS: *m*/*z* [M + H]^+^ for C_32_H_22_N_4_O_6_S calc. 591.13328; exp. 591.1336.

*2-(4-{[(5Z)-3-[(Acridin-4-yl)methyl]-2,4-dioxo-1,3-thiazolidin-5-ylidene]methyl}phenoxy)-N-[2-(trifluoromethyl)phenyl]acetamide* (**8e**). Pale yellow solid. Yield: 67 mg (73%). Mp. 205–206 °C (d, EtOH). IR ν_max_ 3263, 3039, 1741, 1674, 1599, 1519, 1504, 1323, 1237, 1172, 1059, 756, 739, 716 cm^−1^. ^1^H NMR (600 MHz, DMSO-d_6_): δ 9.80 (s, 1H, H-1), 9.17 (s, 1H, H-9‴), 8.21 (d, *J* = 8.4 Hz, 1H, H-8‴), 8.17 (d, *J* = 8.8 Hz, 1H, H-5‴), 8.13 (dd, *J* = 7.4, 2.5 Hz, 1H, H-1‴), 7.99 (s, 1H, H-4), 7.77 (d, *J* = 7.8 Hz, 1H, H-3′), 7.89 (ddd, *J* = 8.6, 6.6, 1.5 Hz, 1H, H-6‴), 7.72 (t, *J* = 7.5 Hz, 1H, H-5′), 7.68 (d, *J* = 8.8 Hz, 2H, H-2″,6″), 7.65 (m, 4H, H-6′, H-3‴ and H-2‴,7‴), 7.49 (t, *J* = 7.6 Hz, 1H, H-4′), 7.19 (d, J = 8.8 Hz, 2H, H-3″,5″), 5.61 (s, 2H, H-10), and 4.87 (s, 2H, H-3) ppm. ^13^C NMR (151 MHz, DMSO-d_6_): δ 167.5 (C-8), 167.1 (C-2), 165.9 (C-6), 159.4 (C-4″), 147.7 (C-10‴a), 146.1 (C-4‴a), 136.8 (C-9‴), 134.6 (C-1′), 133.2 (C-5′), 133.0 (C-4), 132.2 (C-4‴), 132.2 (C-2″,6″), 130.9 (C-6‴), 129.3 (C-6′), 129.1 (C-5‴), 128.5 (C-8‴), 128.1 (C-1‴), 127.0 (C-3‴), 126.9 (C-4′), 126.4 (C-3′), 126.4 (C-1″), 126.2 (C-7‴), 126.3 (C-8‴a), 126.0 (C-9‴a), 125.4 (C-2‴), 124.2 (q, *J* = 30.0 Hz, C-2′), 123.6 (q, *J* = 273.4 Hz, CF_3_), 118.8 (C-5), 115.7 (C-3″,5″), 66.8 (C-3), and 42.3 (C-10) ppm. ^15^N (61 MHz, DMSO-d_6_): δ −259.5 (N-1) and −215.0 (N-7) ppm. HRMS: *m*/*z* [M + H]^+^ for C_33_H_22_F_3_N_3_O_4_S calc. 614.13559; exp. 614.13590.

*2-(4-{[(5Z)-3-[(Acridin-4-yl)methyl]-2,4-dioxo-1,3-thiazolidin-5-ylidene]methyl}phenoxy)-N-[3-(trifluoromethyl)phenyl]acetamide* (**8f**). Pale yellow solid. Yield: 76 mg (83%). Mp. 219–220 °C (d, EtOH). IR ν_max_ 3310, 3028, 1742, 1673, 1597, 1534, 1507, 1335, 1247, 1164, 1058, 741, 716 cm^−1^. ^1^H NMR (600 MHz, DMSO-d_6_): δ 10.50 (s, 1H, H-1), 9.17 (s, 1H, H-9‴), 8.20 (d, *J* = 8.4 Hz, 1H, H-8‴), 8.17 (dd, *J* = 8.8, 1.0 Hz, 1H, H-5‴), 8.13 (m, 2H, H-2′, H-1‴), 7.97 (s, 1H, H-4), 7.89 (ddd, *J* = 8.7, 6.6, 1.5 Hz, 1H, H-6‴), 7.86 (d, *J* = 2.1 Hz, 1H, H-6′), 7.67 (d, *J* = 8.9 Hz, 2H, H-2″,6″), 7.59 (t, *J* = 8.0 Hz, 1H, H-5′), 7.55 (m, 2H, H-2‴, H-3‴), 7.45 (d, *J* = 7.9 Hz, 1H, H-4′), 7.20 (d, *J* = 8.9 Hz, 2H, H-3″,5″), 5.60 (s, 2H, H-10), and 4.87 (s, 2H, H-3) ppm. ^13^C NMR (151 MHz, DMSO-d_6_): δ 167.5 (C-8), 166.7 (C-2), 165.9 (C-6), 159.6 (C-4″), 147.7 (C-10‴a), 146.1 (C-4‴a), 139.1 (C-1′), 136.8 (C-9‴), 133.0 (C-4), 132.3 (C-2″,6″), 132.2 (C-4‴), 130.9 (C-6‴), 130.1 (C-5′), 129.5 (q, *J* = 31.4 Hz, C-3′), 129.1 (C-5‴), 128.5 (C-8‴), 128.1 (C-1‴), 127.0 (C-3‴), 126.2 (C-7‴), 126.2 (C-8‴a), 126.2 (C-1″), 126.0 (C-9‴a), 125.4 (C-2‴), 124.1 (q, *J* = 272.5 Hz, CF_3_), 123.3 (C-6′), 120.1 (C-4′), 118.7 (C-5), 115.7 (C-3″,5″), 115.7 (C-2′), 67.0 (C-3), and 42.3 (C-10) ppm. ^15^N (61 MHz, DMSO-d_6_): δ −251.4 (N-1) ppm. HRMS: *m*/*z* [M + H]^+^ for C_33_H_22_F_3_N_3_O_4_S calc. 614.13559; exp. 614.13560.

*2-(4-{[(5Z)-3-[(Acridin-4-yl)methyl]-2,4-dioxo-1,3-thiazolidin-5-ylidene]methyl}phenoxy)-N-[3-(trifluoromethyl)phenyl]acetamide* (**8g**). Pale yellow solid. Yield: 59 mg (67%). Mp. 246–247 °C (d, EtOH). IR ν_max_ 3281, 3058, 1737, 1673, 1597, 1550, 1505, 1374, 1275, 1166, 1064, 751, 713 cm^−1^. ^1^H NMR (600 MHz, DMSO-d_6_): δ 10.78 (s, 1H, H-1), 9.17 (s, 1H, H-9‴), 8.36 (s, 2H, H-2′,6′), 8.20 (d, *J* = 8.5 Hz, 1H, H-8‴), 8.17 (dd, *J* = 8.8, 1.1 Hz, 1H, H-5‴), 8.13 (dd, *J* = 7.8, 2. Hz, 1H, H-1‴), 7.98 (s, 1H, H-4), 7.89 (ddd, *J* = 8.5, 6.6, 1.4 Hz, 1H, H-6‴), 7.82 (s, 1H, H-4′), 7.68 (d, *J* = 8.8 Hz, 2H, H-2″,6″), 7.65 (ddd, *J* = 8.5, 6.6, 1.2 Hz, 1H, H-7‴), 7.55 (m, 2H, H-2‴, H-3‴), 7.22 (d, *J* = 8.9 Hz, 2H, H-3″,5″), 5.60 (s, 2H, H-10), and 4.90 (s, 2H, H-3) ppm. ^13^C NMR (151 MHz, DMSO-d_6_): δ 167.4 (C-8), 167.4 (C-2), 165.9 (C-6), 159.5 (C-4″), 147.6 (C-10‴a), 146.1 (C-4‴a), 140.2 (C-1′), 136.8 (C-9‴), 132.9 (C-4), 132.2 (C-2″,6″), 132.2 (C-4‴), 130.9 (C-6‴), 130.8 (q, *J* = 32.9 Hz, C-3′,5′), 129.1 (C-5‴), 128.5 (C-8‴), 128.1 (C-1‴), 127.0 (C-3‴), 126.3 (C-1″), 126.2 (C-7‴), 126.2 (C-8‴a), 126.0 (C-9‴a), 125.3 (C-2‴), 123.2 (q, *J* = 272.9 Hz, CF_3_), 119.5 (br s, C-2′,6′), 118.8 (C-5), 116.7 (br s, C-4′), 115.8 (C-3″,5″), 66.9 (C-3), and 42.3 (C-10) ppm. ^15^N (61 MHz, DMSO-d_6_): δ −251.4 (N-1) ppm. HRMS: *m*/*z* [M + H]^+^ for C_34_H_21_F_6_N_3_O_4_S calc. 682.12297; exp. 682.1234.

*2-(4-{[(5Z)-3-[(Acridin-9-yl)methyl]-2,4-dioxo-1,3-thiazolidin-5-ylidene]methyl}-2-methoxyphenoxy)-N-phenylacetamide* (**12a**). Pale yellow solid. Yield: 65 mg (65%). Mp. 260–262 °C (EtOH). IR ν_max_ 3391, 3007, 1745, 1693, 1593, 1543, 1519, 1270, 1176, 1149, 1056 753, 703 cm^−1^. ^1^H NMR (600 MHz, DMSO-d_6_): δ 10.13 (s, 1H, H-1), 8.46 (d, *J* = 8.7 Hz, 2H, H-1‴,8‴), 8.19 (d, *J* = 8.7 Hz, 2H, H-4‴,5‴), 7.90 (s, 1H, H-4), 7.86 (ddd, *J* = 8.6, 6.5, 1.2 Hz, 2H, H-3‴,6‴), 7.68 (ddd, *J* = 8.9, 6.6, 1.2 Hz, 2H, H-2‴,7‴), 7.59 (d, *J* = 8.7 Hz, 2H, H-2′,6′), 7.31 (t, *J* = 7.8 Hz, 2H, H-3′,6′), 7.22 (d, *J* = 2.1 Hz, 1H, H-6″), 7.15 (dd, *J* = 8.5, 2.1 Hz, 1H, H-2″), 7.07 (m, 3H, H-4′, H-3″,5″), 5.92 (s, 2H, H-10), 4.79 (s, 2H, H-3), and 3.83 (s, 3H, OCH_3_) ppm. ^13^C NMR (151 MHz, DMSO-d_6_): δ 167.4 (C-8), 166.0 (C-2), 165.6 (C-6), 149.8 (C-4″), 149.1 (C-5″), 148.1 (C-4‴a,10‴a), 138.4 (C-1′), 137.6 (C-9‴), 134.1 (C-4), 130.1 (C-3‴,6‴), 129.9 (C-4‴,5‴), 128.8 (C-3′,5′), 126.4 (C-2‴,7‴), 126.3 (C-1″), 125.1 (C-8‴a,9‴a), 124.7 (C-1‴,8‴), 123.7 (C-2″), 123.7 (C-4′), 119.4 (C-2′,6′), 117.8 (C-5), 113.9 (C-6″), 113.8 (C-3″), 67.7 (C-3), 55.7 (OCH_3_), and 38.3 (C-10) ppm. HRMS: *m*/*z* [M + H]^+^ for C_33_H_25_N_3_O_5_S calc. 576.15877; exp. 576.1599.

*2-(4-{[(5Z)-3-[(Acridin-9-yl)methyl]-2,4-dioxo-1,3-thiazolidin-5-ylidene]methyl}-2-methoxyphenoxy)-N-(3,5-dimethoxyphenyl)acetamide* (**12b**). Pale yellow solid. Yield: 64 mg (72%). Mp. 264–265 °C (EtOH). IR ν_max_ 3402, 3009, 2935, 2837, 1732, 1681, 1604, 1593, 1514, 1270, 1199, 1150, 1128, 1058, 750 cm^−1^. ^1^H NMR (600 MHz, DMSO-d_6_): δ 10.09 (s, 1H, H-1), 8.46 (d, *J* = 8.9 Hz, 2H, H-1‴,8‴), 8.19 (d, *J* = 8.8 Hz, 2H, H-4‴,5‴), 7.90 (s, 1H, H-4), 7.86 (ddd, *J* = 8.7, 6.5, 1.2 Hz, 2H, H-3‴,6‴), 7.68 (ddd, *J* = 8.9, 6.6, 1.2 Hz, 2H, H-2‴,7‴), 7.22 (d, *J* = 2.2 Hz, 1H, H-6″), 7.15 (dd, *J* = 8.6, 2.2 Hz, 1H, H-2″), 7.04 (d, *J* = 8.6 Hz, 1H, H-3″), 6.84 (d, *J* = 2.2 Hz, 2H, H-2′,6′), 6.23 (t, *J* = 2.2 Hz, 1H, H-4′), 5.92 (s, 2H, H-10), 4.77 (s, 2H, H-3), 3.83 (s, 3H, OCH_3_), and 3.70 (s, 6H, 2 × OCH_3_) ppm. ^13^C NMR (151 MHz, DMSO-d_6_): δ 167.4 (C-8), 166.0 (C-2), 165.6 (C-6), 160.5 (C-3′,5′), 149.7 (C-4″), 149.1 (C-5″), 148.1 (C-4‴a,10‴a), 140.0 (C-1′), 137.6 (C-9‴), 134.1 (C-4), 130.0 (C-3‴,6‴), 129.9 (C-4‴,5‴), 126.4 (C-2‴,7‴), 126.3 (C-1″), 125.1 (C-8‴a,9‴a), 124.7 (C-1‴,8‴), 123.7 (C-2″), 117.9 (C-5), 113.9 (C-6″), 113.7 (C-3″), 97.7 (C-2′,6′), 95.1 (C-4′), 67.6 (C-3), 55.7 (OCH_3_), 55.1 (2 × OCH_3_), and 38.3 (C-10) ppm. ^15^N (61 MHz, DMSO-d_6_): δ −250.0 (N-1) and −211.7 (N-7) ppm. HRMS: *m*/*z* [M + H]^+^ for C_35_H_29_N_3_O_7_S calc. 636.1799; exp. 636.1802.

*2-(4-{[(5Z)-3-[(Acridin-9-yl)methyl]-2,4-dioxo-1,3-thiazolidin-5-ylidene]methyl}-2-methoxyphenoxy)-N-(3,4,5-trimethoxyphenyl)acetamide* (**12c**). Pale yellow solid. Yield: 54 mg (62%). Mp. 268–269 °C (EtOH). IR ν_max_ 3220, 3065, 2938, 2838, 1732, 1682, 1592, 1556, 1509, 1231, 1180, 1133, 753 cm^−1^. ^1^H NMR (600 MHz, DMSO-d_6_): δ 10.07 (s, 1H, H-1), 8.46 (d, *J* = 8.9 Hz, 2H, H-1‴,8‴), 8.19 (d, *J* = 8.7 Hz, 2H, H-4‴,5‴), 7.90 (s, 1H, H-4), 7.86 (ddd, *J* = 8.7, 6.5, 1.2 Hz, 2H, H-3‴,6‴), 7.68 (ddd, *J* = 8.9, 6.6, 1.2 Hz, 2H, H-2‴,7‴), 7.22 (d, *J* = 2.2 Hz, 1H, H-6″), 7.15 (dd, *J* = 8.6, 2.2 Hz, 1H, H-2″), 7.05 (d, *J* = 8.6 Hz, 1H, H-3″), 6.99 (s, 2H, H-2′,6′), 5.92 (s, 2H, H-10), 4.76 (s, 2H, H-3), 3.83 (s, 3H, OCH_3_), 3.72 (s, 6H, 2 × OCH_3_), and 3.61 (s, 3H, OCH_3_) ppm. ^13^C NMR (151 MHz, DMSO-d_6_): δ 167.4 (C-8), 165.8 (C-2), 165.6 (C-6), 152.7 (C-3′,5′), 149.7 (C-4″), 149.2 (C-5″), 148.1 (C-4‴a,10‴a), 137.6 (C-9‴), 134.5 (C-1′), 134.1 (C-4), 133.7 (C-4′), 130.1 (C-3‴,6‴), 129.9 (C-4‴,5‴), 126.4 (C-2‴,7‴), 126.3 (C-1″), 125.1 (C-8‴a,9‴a), 124.7 (C-1‴,8‴), 123.7 (C-2″), 117.9 (C-5), 113.9 (C-6″), 113.8 (C-3″), 97.1 (C-2′,6′), 67.6 (C-3), 60.1 (OCH_3_), 55.7 (3 × OCH_3_), and 38.3 (C-10) ppm. ^15^N (61 MHz, DMSO-d_6_): δ −250.8 (N-1) and −211.7 (N-7) ppm. HRMS: *m*/*z* [M + H]^+^ for C_36_H_31_N_3_O_8_S calc. 666.19046; exp. 666.19140.

*2-(4-{[(5Z)-3-[(Acridin-9-yl)methyl]-2,4-dioxo-1,3-thiazolidin-5-ylidene]methyl}-2-methoxyphenoxy)-N-(4-nitrophenyl)acetamide* (**12d**). Pale yellow solid. Yield: 35 mg (38%). Mp. 288–290 °C (EtOH). IR ν_max_ 3371, 1729, 1677, 1596, 1545, 1509, 1346, 1278, 1145, 1054, 753 cm^−1^. ^1^H NMR (600 MHz, DMSO-d_6_): δ 10.78 (s, 1H, H-1), 8.48 (d, *J* = 8.9 Hz, 2H, H-1‴,8‴), 8.23 (d, *J* = 9.2 Hz, 2H, H-3′,5′), 8.19 (d, *J* = 8.6 Hz, 2H, H-4‴,5‴), 7.90 (s, 1H, H-4), 7.86 (ddd, *J* = 8.7, 6.5, 1.2 Hz, 2H, H-3‴,6‴), 7.68 (ddd, *J* = 8.9, 6.6, 1.2 Hz, 2H, H-2‴,7‴), 7.23 (d, *J* = 2.2 Hz, 1H, H-6″), 7.15 (dd, *J* = 8.6, 2.2 Hz, 1H, H-2″), 7.06 (d, *J* = 8.6 Hz, 1H, H-3″), 6.84 (d, *J* = 9.2 Hz, 2H, H-2′,6′), 5.92 (s, 2H, H-10), 4.87 (s, 2H, H-3), and 3.83 (s, 3H, OCH_3_) ppm. ^13^C NMR (151 MHz, DMSO-d_6_): δ 167.4 (C-8), 167.1 (C-2), 165.6 (C-6), 149.6 (C-4″), 149.1 (C-5″), 148.1 (C-4‴a,10‴a), 144.6 (C-1′), 142.5 (C-4′), 137.6 (C-9‴), 134.1 (C-4), 130.1 (C-3‴,6‴), 129.9 (C-4‴,5‴), 126.4 (C-2‴,7‴), 126.4 (C-1″), 125.1 (C-8‴a,9‴a), 125.0 (C-3′,5′), 124.7 (C-1‴,8‴), 123.7 (C-2″), 119.1 (C-2′,6′), 117.9 (C-5), 114.0 (C-6″), 113.9 (C-3″), 67.5 (C-3), 55.7 (OCH_3_), and 38.3 (C-10) ppm. ^15^N (61 MHz, DMSO-d_6_): δ −248.3 (N-1) and −211.7 (N-7) ppm. HRMS: *m*/*z* [M + H]^+^ for C_33_H_24_N_4_O_7_S calc. 621.14385; exp. 621.14400.

*2-(4-{[(5Z)-3-[(Acridin-9-yl)methyl]-2,4-dioxo-1,3-thiazolidin-5-ylidene]methyl}-2-methoxyphenoxy)-N-[2-(trifluoromethyl)phenyl]acetamide* (**12e**). Pale yellow solid. Yield: 60 mg (67%). Mp. 255–256 °C (EtOH). IR ν_max_ 3398, 3036, 1732, 1682, 1589, 1549, 1519, 1321 1281, 1176, 1145, 1062, 761, 748 cm^−1^. ^1^H NMR (600 MHz, DMSO-d_6_): δ 9.56 (s, 1H, H-1), 8.46 (d, *J* = 8.7 Hz, 2H, H-1‴,8‴), 8.19 (d, *J* = 8.6 Hz, 2H, H-4‴,5‴), 7.92 (s, 4H, H-4), 7.86 (ddd, *J* = 8.7, 6.5, 1.2 Hz, 2H, H-3‴,6‴), 7.81 (d, *J* = 8.1 Hz, 1H, H-6′), 7.75 (dd, *J* = 8.0, 1.5 Hz, 1H, H-3′), 7.69 (m, 3H, H-5′, H-2‴,7‴), 7.44 (t, *J* = 7.7 Hz, 1H, H-4′), 7.24 (d, *J* = 2.2 Hz, 1H, H-6″), 7.17 (dd, *J* = 8.6, 2.2 Hz, 1H, H-2″), 7.11 (d, *J* = 8.5 Hz, 1H, H-3″), 5.93 (s, 2H, H-10), 4.84 (s, 2H, H-3), and 3.83 (s, 3H, OCH_3_) ppm. ^13^C NMR (151 MHz, DMSO-d_6_): δ 167.4 (C-8), 166.9 (C-2), 165.6 (C-6), 149.2 (C-5″), 149.1 (C-4″), 148.1 (C-4‴a,10‴a), 137.6 (C-9‴), 134.5 (C-1′), 134.1 (C-4), 133.3 (C-5′), 130.1 (C-3‴,6‴), 129.9 (C-4‴,5‴), 127.6 (br s, C-6′), 126.6 (C-1″), 126.4 (C-2‴,7‴), 126.3 (C-3′), 126.3 (C-4′), 125.1 (C-8‴a,9‴a), 124.7 (C-1‴,8‴), 123.7 (C-2″), 122.7 (C-2′), 118.1 (C-5), 113.9 (C-3″), 113.8 (C-6″), 67.3 (C-3), 55.7 (OCH_3_), and 38.3 (C-10) ppm. HRMS: *m*/*z* [M + H]^+^ for C_34_H_24_F_3_N_3_O_5_S calc. 644.14615; exp. 644.14600.

*2-(4-{[(5Z)-3-[(Acridin-9-yl)methyl]-2,4-dioxo-1,3-thiazolidin-5-ylidene]methyl}-2-methoxyphenoxy)-N-[3-(trifluoromethyl)phenyl]acetamide* (**12f**). Pale yellow solid. Yield: 67 mg (74%). Mp. 269–270 °C (EtOH). IR ν_max_ 3395, 2916, 1737, 1681, 1594, 1548, 1515, 1330, 1280, 1180, 1144, 1069, 752 cm^−1^. ^1^H NMR (600 MHz, DMSO-d_6_): δ 10.50 (s, 1H, H-1), 8.46 (d, *J* = 8.8 Hz, 2H, H-1‴,8‴), 8.19 (d, *J* = 8.9 Hz, 2H, H-4‴,5‴), 8.09 (s, 1H, H-2′), 7.91 (s, 1H, H-4), 7.86 (ddd, *J* = 8.7, 6.7, 1.2 Hz, 2H, H-3‴,6‴), 7.80 (d, *J* = 8.2 Hz, 1H, H-6′), 7.68 (ddd, *J* = 8.8, 6.6, 1.2 Hz, 2H, H-2‴,7‴), 7.56 (t, *J* = 8.0 Hz, 1H, H-5′), 7.43 (d, *J* = 7.3 Hz, 1H, H-4′), 7.23 (d, *J* = 2.2 Hz, 1H, H-6″), 7.15 (dd, *J* = 8.6, 2.2 Hz, 1H, H-2″), 7.07 (d, *J* = 8.5 Hz, 1H, H-3″), 5.92 (s, 2H, H-10), 4.83 (s, 2H, H-3), and 3.83 (s, 3H, OCH_3_) ppm. ^13^C NMR (151 MHz, DMSO-d_6_): δ 167.4 (C-8), 166.7 (C-2), 165.6 (C-6), 149.7 (C-4″), 149.2 (C-5″), 148.1 (C-4‴a,10‴a), 139.1 (C-1′), 137.6 (C-9‴), 134.1 (C-4), 130.1 (C-5′), 130.0 (C-3‴,6‴), 129.9 (C-4‴,5‴), 129.6 (q, *J* = 30.7 Hz, C-3′), 126.4 (C-2‴,7‴), 126.4 (C-1″), 125.1 (C-8‴a,9‴a), 124.7 (C-1‴,8‴), 124.0 (q, *J* = 272.0 Hz, CF_3_), 123.7 (C-2″), 123.0 (C-6′), 120.0 (C-4′), 117.9 (C-5), 115.5 (br s, C-2′), 114.0 (C-6″), 113.9 (C-3″), 67.5 (C-3), 55.7 (OCH_3_), and 38.3 (C-10) ppm. ^15^N (61 MHz, DMSO-d_6_): δ −251.1 (N-1) and −211.7 (N-7) ppm. HRMS: *m*/*z* [M + H]^+^ for C_34_H_24_F_3_N_3_O_5_S calc. 644.14615; exp. 644.14610.

*2-(4-{[(5Z)-3-[(Acridin-9-yl)methyl]-2,4-dioxo-1,3-thiazolidin-5-ylidene]methyl}-2-methoxyphenoxy)-N-[3,5-bis(trifluoromethyl)phenyl]acetamide* (**12g**). Pale yellow solid. Yield: 53 mg (62%). Mp. 287–288 °C (EtOH). IR ν_max_ 3380, 2958, 1731, 1678, 1596, 1549, 1510, 1381, 1276, 1173, 1133, 1054, 753 cm^−1^. ^1^H NMR (600 MHz, DMSO-d_6_): δ 10.82 (s, 1H, H-1), 8.46 (d, *J* = 8.9 Hz, 2H, H-1‴,8‴), 8.29 (s, 2H, H-2′,6′), 8.19 (d, *J* = 8.7 Hz, 2H, H-4‴,5‴), 7.91 (s, 1H, H-4), 7.86 (ddd, *J* = 8.4, 6.7, 1.2 Hz, 2H, H-3‴,6‴), 7.80 (s, 1H, H-4′), 7.68 (ddd, *J* = 8.8, 6.6, 1.2 Hz, 2H, H-2‴,7‴), 7.24 (d, *J* = 2.2 Hz, 1H, H-6″), 7.14 (dd, *J* = 8.6, 2.2 Hz, 1H, H-2″), 7.09 (d, *J* = 8.5 Hz, 1H, H-3″), 5.93 (s, 2H, H-10), 4.86 (s, 2H, H-3), and 3.83 (s, 3H, OCH_3_) ppm. ^13^C NMR (151 MHz, DMSO-d_6_): δ 167.4 (C-2), 167.4 (C-8), 165.6 (C-6), 149.7 (C-4″), 149.2 (C-5″), 148.1 (C-4‴a,10‴a), 140.3 (C-1′), 137.6 (C-9‴), 134.1 (C-4), 130.7 (q, *J* = 32.0 Hz, C-3′,5′), 130.0 (C-3‴,6‴), 129.9 (C-4‴,5‴), 126.5 (C-1″), 126.4 (C-2‴,7‴), 125.1 (C-8‴a,9‴a), 124.7 (C-1‴,8‴), 123.7 (C-2″), 123.2 (q, *J* = 272.5 Hz, CF_3_), 119.3 (br s, C-2′,6′), 118.0 (C-5), 116.6 (C-4′), 114.1 (C-6″), 114.1 (C-3″), 67.6 (C-3), 55.7 (OCH_3_), and 38.3 (C-10) ppm. HRMS: *m*/*z* [M + H]^+^ for C_35_H_23_F_6_N_3_O_5_S calc. 712.13354; exp. 712.13370.

*2-(4-{[(5Z)-3-[(Acridin-4-yl)methyl]-2,4-dioxo-1,3-thiazolidin-5-ylidene]methyl}-2-methoxyphenoxy)-N-phenylacetamide* (**13a**). Pale yellow solid. Yield: 73 mg (70%). Mp. 213–214 °C (EtOH). IR ν_max_ 3389, 3055, 1731, 1682, 1601, 1542, 1512, 1268, 1143, 741 cm^−1^. ^1^H NMR (600 MHz, DMSO-d_6_): δ 10.15 (s, 1H, H-1), 9.17 (s, 1H, H-9‴), 8.20 (dd, *J* = 8.4, 0.8 Hz, 1H, H-8‴), 8.17 (dd, *J* = 8.8, 1.0 Hz, 1H, H-5‴), 8.13 (dd, *J* = 8.0, 1.7 Hz, 1H, H-1‴), 7.97 (s, 1H, H-4), 7.89 (ddd, *J* = 8.8, 6.6, 1.4 Hz, 1H, H-6‴), 7.66 (ddd, *J* = 8.0, 6.6, 1.1 Hz, 1H, H-7‴), 7.62 (d, *J* = 7.4 Hz, 2H, H-2′,6′), 7.55 (m, 2H, H-2‴, H-3‴), 7.33 (m, 2H, H-3′,5′), 7.33 (d, *J* = 2.1 Hz, 1H, H-6″), 7.25 (dd, *J* = 8.7, 2.1 Hz, 1H, H-2″), 7.12 (d, *J* = 8.5 Hz, 1H, H-3″), 7.08 (t, *J* = 7.4 Hz, 1H, H-4′), 5.60 (s, 2H, H-10), 4.82 (s, 2H, H-3), and 3.88 (s, 3H, OCH_3_) ppm. ^13^C NMR (151 MHz, DMSO-d_6_): δ 167.5 (C-8), 166.0 (C-2), 165.8 (C-6), 149.6 (C-4″), 149.2 (C-5″), 147.7 (C-10‴a), 146.1 (C-4‴a), 138.4 (C-1′), 136.8 (C-9‴), 133.3 (C-4), 132.2 (C-4‴), 130.9 (C-6‴), 129.1 (C-5‴), 128.8 (C-3′,5′), 128.5 (C-8‴), 128.1 (C-1‴), 127.0 (C-1″), 126.6 (C-3‴), 126.2 (C-7‴), 126.2 (C-8‴a), 126.0 (C-9‴a), 125.3 (C-2‴), 123.7 (C-4′), 123.5 (C-2″), 119.4 (C-2′,6′), 118.9 (C-5), 114.0 (C-6″), 113.8 (C-3″), 67.7 (C-3), 55.7 (OCH_3_), and 42.3 (C-10) ppm. ^15^N (61 MHz, DMSO-d_6_): δ −250.6 (N-1) and −215.0 (N-7) ppm. HRMS: *m*/*z* [M + H]^+^ for C_33_H_25_N_3_O_5_S calc. 576.15880; exp. 576.15880.

*2-(4-{[(5Z)-3-[(Acridin-4-yl)methyl]-2,4-dioxo-1,3-thiazolidin-5-ylidene]methyl}-2-methoxyphenoxy)-N-(3,5-dimethoxyphenyl)acetamide* (**13b**). Pale yellow solid. Yield: 63 mg (71%). Mp. 222–224 °C (EtOH). IR ν_max_ 3350, 2935, 1728, 1665, 1603, 1556, 1510, 1270, 1148, 1061, 742 cm^−1^. ^1^H NMR (600 MHz, DMSO-d_6_): δ 10.12 (s, 1H, H-1), 9.17 (s, 1H, H-9‴), 8.20 (d, *J* = 8.4 Hz, 1H, H-8‴), 8.17 (d, *J* = 8.8 Hz, 1H, H-5‴), 8.13 (dd, *J* = 8.3, 1.8 Hz, 1H, H-1‴), 7.97 (s, 1H, H-4), 7.89 (ddd, *J* = 8.3, 6.6, 1.4 Hz, 1H, H-6‴), 7.66 (ddd, *J* = 8.0, 6.6, 1.1 Hz, 1H, H-7‴), 7.56 (m, 2H, H-2‴, H-3‴), 7.33 (d, *J* = 2.1 Hz, 1H, H-6″), 7.25 (dd, *J* = 8.5, 2.1 Hz, 1H, H-2″), 7.10 (d, *J* = 8.5 Hz, 1H, H-3″), 6.87 (d, *J* = 2.2 Hz, 2H, H-2′,6′), 6.25 (t, *J* = 2.2 Hz, 1H, H-4′), 5.60 (s, 2H, H-10), 4.80 (s, 2H, H-3), 3.88 (s, 3H, OCH_3_), and 3.71 (s, 6H, 2 × OCH_3_) ppm. ^13^C NMR (151 MHz, DMSO-d_6_): δ 167.5 (C-8), 166.1 (C-2), 165.8 (C-6), 160.5 (C-3′,5′), 149.6 (C-4″), 149.2 (C-5″), 147.7 (C-10‴a), 146.1 (C-4‴a), 140.0 (C-1′), 136.8 (C-9‴), 133.3 (C-4), 132.2 (C-4‴), 130.9 (C-6‴), 129.1 (C-5‴), 128.5 (C-8‴), 128.1 (C-1‴), 127.0 (C-3‴), 126.6 (C-1″), 126.2 (C-7‴), 126.2 (C-8‴a), 126.0 (C-9‴a), 125.3 (C-2‴), 123.5 (C-2″), 118.9 (C-5), 114.0 (C-6″), 113.8 (C-3″), 97.7 (C-2′,6′), 95.7 (C-4′), 67.7 (C-3), 55.7 (OCH_3_), 55.1 (OCH_3_), and 42.3 (C-10) ppm. ^15^N (61 MHz, DMSO-d_6_): δ −250.0 (N-1) and −215.0 (N-7) ppm. HRMS: *m*/*z* [M + H]^+^ for C_35_H_29_N_3_O_7_S calc. 636.17990; exp. 636.18010.

*2-(4-{[(5Z)-3-[(Acridin-4-yl)methyl]-2,4-dioxo-1,3-thiazolidin-5-ylidene]methyl}-2-methoxyphenoxy)-N-(3,4,5-trimethoxyphenyl)acetamide* (**13c**). Pale yellow solid. Yield: 59 mg (68%). Mp. 228–230 °C (d, EtOH). IR ν_max_ 3361, 2929, 1736, 1677, 1606, 1538, 1504, 1279, 1232, 1148, 1129, 1048, 739 cm^−1^. ^1^H NMR (600 MHz, DMSO-d_6_): δ 10.10 (s, 1H, H-1), 9.17 (s, 1H, H-9‴), 8.20 (d, *J* = 8.4 Hz, 1H, H-8‴), 8.17 (dd, *J* = 8.8, 1.1 Hz, 1H, H-5‴), 8.13 (dd, *J* = 8.1, 1.8 Hz, 1H, H-1‴), 7.97 (s, 1H, H-4), 7.89 (ddd, *J* = 8.4, 6.6, 1.4 Hz, 1H, H-6‴), 7.66 (ddd, *J* = 8.0, 6.6, 1.1 Hz, 1H, H-7‴), 7.55 (m, 2H, H-2‴, H-3‴), 7.33 (d, *J* = 2.1 Hz, 1H, H-6″), 7.25 (dd, *J* = 8.5, 2.1 Hz, 1H, H-2″), 7.11 (d, *J* = 8.5 Hz, 1H, H-3″), 7.02 (s, 2H, H-2′,6′), 5.60 (s, 2H, H-10), 4.80 (s, 2H, H-3), 3.88 (s, 3H, OCH_3_), 3.74 (s, 6H, 2 × OCH_3_), and 3.62 (s, 3H, OCH_3_) ppm. ^13^C NMR (151 MHz, DMSO-d_6_): δ 167.5 (C-8), 165.9 (C-2), 165.8 (C-6), 152.8 (C-3′,5′), 149.6 (C-4″), 149.2 (C-5″), 147.7 (C-10‴a), 146.1 (C-4‴a), 136.8 (C-9‴), 134.5 (C-1′), 133.7 (C-4′), 133.3 (C-4), 132.2 (C-4‴), 130.9 (C-6‴), 129.1 (C-5‴), 128.5 (C-8‴), 128.1 (C-1‴), 127.0 (C-3‴), 126.6 (C-1″), 126.2 (C-7‴), 126.2 (C-8‴a), 126.0 (C-9‴a), 125.3 (C-2‴), 123.5 (C-2″), 118.9 (C-5), 114.0 (C-6″), 113.9 (C-3″), 97.1 (C-2′,6′), 60.1 (OCH_3_), 55.7 (OCH_3_), 67.7 (C-3), and 42.3 (C-10) ppm. ^15^N (61 MHz, DMSO-d_6_): δ −250.8 (N-1) and −215.0 (N-7) ppm. HRMS: *m*/*z* [M + H]^+^ for C_36_H_31_N_3_O_8_S calc. 666.19050; exp. 666.19070.

*2-(4-{[(5Z)-3-[(Acridin-4-yl)methyl]-2,4-dioxo-1,3-thiazolidin-5-ylidene]methyl}-2-methoxyphenoxy)-N-(4-nitrophenyl)acetamide* (**13d**). Pale yellow solid. Yield: 65 mg (70%). Mp. 254–257 °C (d, EtOH). IR ν_max_ 3367, 2936, 1738, 1678, 1610, 1544, 1508, 1338, 1265, 1144, 1059, 741 cm^−1^. ^1^H NMR (600 MHz, DMSO-d_6_): δ 10.81 (s, 1H, H-1), 9.16 (s, 1H, H-9‴), 8.24 (d, *J* = 9.3 Hz, 2H, H-3′,5′), 8.20 (d, *J* = 8.8 Hz, 1H, H-8‴), 8.17 (d, *J* = 8.8 Hz, 1H, H-5″), 8.13 (dd, *J* = 8.1, 1.8 Hz, 1H, H-1″), 7.97 (s, 1H, H-4), 7.89 (ddd, *J* = 8.4, 6.6, 1.4 Hz, 1H, H-6‴), 7.87 (d, *J* = 9.2 Hz, 2H, H-2′,6′), 7.66 (ddd, *J* = 8.0, 6.6, 1.1 Hz, 1H, H-7‴), 7.55 (m, 2H, H-2‴, H-3‴), 7.33 (d, *J* = 2.1 Hz, 1H, H-6″), 7.24 (dd, *J* = 8.5, 2.1 Hz, 1H, H-2″), 7.12 (d, *J* = 8.5 Hz, 1H, H-3″), 5.60 (s, 2H, H-10), 4.90 (s, 2H, H-3), and 3.88 (s, 3H, OCH_3_) ppm. ^13^C NMR (151 MHz, DMSO-d_6_): δ 167.5 (C-8), 167.2 (C-2), 165.8 (C-6), 149.5 (C-4″), 149.2 (C-5″), 147.7 (C-10‴a), 146.1 (C-4‴a), 144.6 (C-1′), 142.5 (C-4′), 136.8 (C-9‴), 133.3 (C-4), 132.2 (C-4‴), 130.9 (C-6‴), 129.1 (C-5‴), 128.5 (C-8‴), 128.2 (C-1‴), 127.0 (C-3‴), 126.7 (C-1″), 126.2 (C-7‴), 126.2 (C-8‴a), 126.0 (C-9‴a), 125.4 (C-2‴), 125.1 (C-3′,5′), 123.5 (C-2″), 119.2 (C-2′,6′), 119.0 (C-5), 114.1 (C-6″), 113.9 (C-3″), 67.6 (C-3), 55.8 (OCH_3_), and 42.4 (C-10) ppm. ^15^N (61 MHz, DMSO-d_6_): δ −248.3 (N-1) and −215.0 (N-7) ppm. HRMS: *m*/*z* [M + H]^+^ for C_33_H_24_N_4_O_7_S calc. 621.14390; exp. 621.14420.

*2-(4-{[(5Z)-3-[(Acridin-4-yl)methyl]-2,4-dioxo-1,3-thiazolidin-5-ylidene]methyl}-2-methoxyphenoxy)-N-[2-(trifluoromethyl)phenyl]acetamide* (**13e**). Pale yellow solid. Yield: 66 mg (73%). Mp. 217–218 °C (EtOH). IR ν_max_ 3393, 3025, 1741, 1676, 1607, 1562, 1510, 1334, 1269, 1152, 1072, 758, 743 cm^−1^. ^1^H NMR (600 MHz, DMSO-d_6_): δ 9.58 (s, 1H, H-1), 9.17 (s, 1H, H-9‴), 8.21 (d, *J* = 8.4 Hz, 1H, H-8‴), 8.17 (d, *J* = 8.8 Hz, 1H, H-5‴), 8.13 (dd, *J* = 8.0, 2.3 Hz, 1H, H-1‴), 7.99 (s, 1H, H-4), 7.89 (ddd, *J* = 8.5, 6.6, 1.5 Hz, 1H, H-6‴), 7.84 (d, *J* = 8.1 Hz, 1H, H-6′), 7.77 (d, *J* = 7.3 Hz, 1H, H-3′), 7.72 (t, *J* = 7.5 Hz, 1H, H-5′), 7.66 (ddd, *J* = 8.1, 6.6, 1.1 Hz, 1H, H-7‴), 7.56 (m, 2H, H-2‴, H-3‴), 7.45 (t, *J* = 7.7 Hz, 1H, H-4′), 7.34 (d, *J* = 2.1 Hz, 1H, H-6″), 7.27 (dd, *J* = 8.5, 2.1 Hz, 1H, H-2″), 7.17 (d, *J* = 8.5 Hz, 1H, H-3″), 5.61 (s, 2H, H-10), 4.87 (s, 2H, H-3), and 3.88 (s, 3H, OCH_3_) ppm. ^13^C NMR (151 MHz, DMSO-d_6_): δ 167.4 (C-8), 166.9 (C-2), 165.8 (C-6), 149.2 (C-5″), 148.9 (C-4″), 147.7 (C-10‴a), 146.1 (C-4‴a), 136.8 (C-9‴), 134.5 (C-1′), 133.3 (C-4), 133.3 (C-5′), 132.1 (C-4‴), 130.9 (C-6‴), 129.1 (C-5‴), 128.5 (C-8‴), 128.1 (C-1‴), 127.6 (C-6′), 127.0 (C-3‴), 126.9 (C-1″), 126.4 (C-3′), 126.4 (C-4′), 126.3 (C-7‴), 126.2 (C-8‴a), 126.0 (C-9‴a), 125.5 (q, *J* = 272.8 Hz, CF_3_), 125.3 (C-2‴), 123.4 (C-2″), 122.6 (q, *J* = 32.4 Hz, C-2′), 119.2 (C-5), 114.0 (C-3″), 113.9 (C-6″), 67.4 (C-3), 55.8 (OCH_3_), and 42.3 (C-10) ppm. ^15^N (61 MHz, DMSO-d_6_): δ −260.3 (N-1) and −215.0 (N-7) ppm. HRMS: *m*/*z* [M + H]^+^ for C_34_H_24_F_3_N_3_O_5_S calc. 644.14620; exp. 644.14610.

*2-(4-{[(5Z)-3-[(Acridin-4-yl)methyl]-2,4-dioxo-1,3-thiazolidin-5-ylidene]methyl}-2-methoxyphenoxy)-N-[3-(trifluoromethyl)phenyl]acetamide* (**13f**). Pale yellow solid. Yield: 68 mg (75%). Mp. 203–205 °C (EtOH). IR ν_max_ 3347, 2917, 1727, 1664, 1607, 1562, 1510, 1334, 1269, 1152, 1072, 758, 743 cm^−1^. ^1^H NMR (600 MHz, DMSO-d_6_): δ 10.53 (s, 1H, H-1), 9.17 (s, 1H, H-9‴), 8.20 (d, *J* = 8.4 Hz, 1H, H-8‴), 8.17 (dd, *J* = 8.8, 1.0 Hz, 1H, H-5‴), 8.13 (dd, *J* = 8.2, 1.8 Hz, 1H, H-1‴), 8.11 (br s, 1H, H-2′), 7.97 (s, 1H, H-4), 7.89 (ddd, *J* = 8.5, 6.6, 1.4 Hz, 1H, H-6‴), 7.83 (d, *J* = 8.6 Hz, 1H, H-6′), 7.66 (ddd, *J* = 8.2, 6.7, 1.1 Hz, 1H, H-7‴), 7.56 (m, 3H, H-5′, H-2‴, H-3‴), 7.45 (d, *J* = 7.8 Hz, 1H, H-4′), 7.34 (d, *J* = 2.1 Hz, 1H, H-6″), 7.25 (dd, *J* = 8.5, 2.1 Hz, 1H, H-2″), 7.13 (d, *J* = 8.5 Hz, 1H, H-3″), 5.60 (s, 2H, H-10), 4.86 (s, 2H, H-3), and 3.88 (s, 3H, OCH_3_) ppm. ^13^C NMR (151 MHz, DMSO-d_6_): δ 167.5 (C-8), 166.8 (C-2), 165.8 (C-6), 149.5 (C-4″), 149.2 (C-5″), 147.7 (C-10‴a), 146.1 (C-4‴a), 139.2 (C-1′), 136.8 (C-9‴), 133.3 (C-4), 132.2 (C-4‴), 130.9 (C-6‴), 130.1 (C-5′), 129.5 (q, *J* = 31.6 Hz, C-3′), 129.1 (C-5‴), 128.5 (C-8‴), 128.1 (C-1‴), 127.0 (C-3‴), 126.7 (C-1″), 126.2 (C-7‴), 126.2 (C-8‴a), 126.0 (C-9‴a), 125.3 (C-2‴), 124.1 (q, *J* = 272.3 Hz, CF_3_), 123.5 (C-2″), 123.0 (C-6′), 120.0 (br s, C-4′), 119.1 (C-5), 115.5 (br s, C-2′), 114.0 (C-6″), 114.0 (C-3″), 67.6 (C-3), 55.7 (OCH_3_), and 42.3 (C-10) ppm. ^15^N (61 MHz, DMSO-d_6_): δ −251.1 (N-1) and −215.0 (N-7) ppm. HRMS: *m*/*z* [M + H]^+^ for C_34_H_24_F_3_N_3_O_5_S calc. 644.14620; exp. 644.14630.

*2-(4-{[(5Z)-3-[(Acridin-4-yl)methyl]-2,4-dioxo-1,3-thiazolidin-5-ylidene]methyl}-2-methoxyphenoxy)-N-[3,5-bis(trifluoromethyl)phenyl]acetamide* (**13g**). Pale yellow solid. Yield: 59 mg (69%). Mp. 249–250 °C (EtOH). IR ν_max_ 3348, 2914, 1731, 1674, 1607, 1544, 1505, 1379, 1276, 1171, 1056, 744 cm^−1^. ^1^H NMR (600 MHz, DMSO-d_6_): δ 10.85 (s, 1H, H-1), 9.17 (s, 1H, H-9‴), 8.32 (s, 2H, H-2′,6′), 8.20 (d, *J* = 8.4 Hz, 1H, H-8‴), 8.17 (dd, *J* = 8.8, 1.0 Hz, 1H, H-5‴), 8.13 (dd, *J* = 8.0, 1.9 Hz, 1H, H-1‴), 7.97 (s, 1H, H-4), 7.89 (ddd, *J* = 8.5, 6.6, 1.4 Hz, 1H, H-6‴), 7.81 (s, 1H, H-4′), 7.66 (ddd, *J* = 8.1, 6.6, 1.2 Hz, 1H, H-7‴), 7.55 (m, 2H, H-2‴, H-3‴), 7.34 (d, *J* = 2.1 Hz, 1H, H-6″), 7.24 (dd, *J* = 8.5, 2.1 Hz, 1H, H-2″), 7.15 (d, *J* = 8.5 Hz, 1H, H-3″), 5.60 (s, 2H, H-10), 4.90 (s, 2H, H-3), and 3.88 (s, 3H, OCH_3_) ppm. ^13^C NMR (151 MHz, DMSO-d_6_): δ 167.5 (C-8), 167.5 (C-2), 165.8 (C-6), 149.4 (C-4″), 149.3 (C-5″), 147.7 (C-10‴a), 146.1 (C-4‴a), 140.3 (C-1′), 136.8 (C-9‴), 133.3 (C-4), 132.2 (C-4‴), 130.9 (C-6‴), 130.8 (q, *J* = 32.0 Hz, C-3′,5′), 129.1 (C-5‴), 128.5 (C-8‴), 128.1 (C-1‴), 127.0 (C-3‴), 126.8 (C-1″), 126.2 (C-7‴), 126.2 (C-8‴a), 126.0 (C-9‴a), 125.3 (C-2‴), 123.5 (C-2″), 123.2 (q, *J* = 272.7 Hz, CF_3_), 119.3 (br s, C-2′,6′), 119.1 (C-5), 116.6 (br s, C-4′), 114.1 (C-6″), 114.1 (C-3″), 67.7 (C-3), 55.8 (OCH_3_), and 42.4 (C-10) ppm. ^15^N (61 MHz, DMSO-d_6_): δ −251.1 (N-1) ppm. HRMS: *m*/*z* [M + H]^+^ for C_35_H_23_F_6_N_3_O_5_S calc. 712.13350; exp. 712.13370.

### 3.8. General Synthetic Procedures for Hydrochlorides **7**, **8**, **12**, and **13**

The stirring suspension of derivatives **7a**–**g**, **8a**–**g**, **12a**–**g**, and **13a**–**g** (30 mg) in dry ethanol (3 mL) was bubbled with hydrogen chloride gas, which was formed by the dropwise addition of concentrated sulfuric acid to a saturated aqueous solution of sodium chloride. The course of salt formation was monitored by TLC (*n*Hex:EtOAc, *v*/*v* 1:1). The resulting hydrochloride precipitate was filtered and washed with a small amount of dry ethanol.

*2-(4-{[(5Z)-3-[(Acridin-9-yl)methyl]-2,4-dioxo-1,3-thiazolidin-5-ylidene]methyl}phenoxy)-N-phenylacetamide dihydrochloride* (**7a·2HCl**). Yellow solid. Yield: 29 mg (85%). IR ν_max_ 3081, 1730, 1668, 1589, 1552, 1509, 1241, 1180, 751 cm^−1^. ^1^H NMR (600 MHz, DMSO-d_6_): δ 10.22 (s, 1H, H-1), 8.67 (d, *J* = 8.9 Hz, 2H, H-1‴,8‴), 8.46 (d, *J* = 8.8 Hz, 2H, H-4‴,5‴), 8.18 (t, *J* = 7.7 Hz, 2H, H-3‴,6‴), 7.91 (m, 2H, H-2‴,7‴), 7.90 (s, 1H, H-4), 7.61 (d, *J* = 7.4 Hz, 2H, H-2′,6′), 7.57 (d, *J* = 9.0 Hz, 2H, H-2″,6″), 7.31 (t, *J* = 7.9 Hz, 2H, H-3′,5′), 7.14 (d, *J* = 9.0 Hz, 2H, H-3″,5″), 7.07 (t, *J* = 7.4 Hz, 1H, H-4′), 6.06 (s, 2H, H-10), and 4.81 (s, 2H, H-3) ppm. ^13^C NMR (151 MHz, DMSO-d_6_): δ 167.5 (C-8), 165.9 (C-2), 165.6 (C-6), 160.0 (C-4″), 138.3 (C-1′), 134.8 (C-3‴,6‴), 134.1 (C-4), 132.4 (C-2″,6″), 128.8 (C-3′,5′), 127.7 (C-2‴,7‴), 125.7 (C-1″), 125.7 (C-1‴,8‴), 125.4 (C-8‴a,9‴a), 123.7 (C-4′), 119.6 (C-2′,6′), 117.3 (C-5), 115.7 (C-3″,5″), 67.0 (C-3), and 38.6 (C-10) ppm. For C_32_H_25_Cl_2_N_3_O_4_S (618.529 g · mol^−1^) calc.: C 62.14; H 4.07; N 6.79; and S 5.18%, exp.: C 62.03; H 4.02; N 6.67; and S 5.16%.

*2-(4-{[(5Z)-3-[(Acridin-9-yl)methyl]-2,4-dioxo-1,3-thiazolidin-5-ylidene]methyl}phenoxy)-N-(3,5-dimethoxyphenyl)acetamide dihydrochloride* (**7b·2HCl**). Yellow solid. Yield: 29 mg (86%). IR ν_max_ 3061, 1736, 1680, 1588, 1562, 1509, 1249, 1182, 1150, 1058, 748 cm^−1^. ^1^H NMR (600 MHz, DMSO-d_6_): δ 10.16 (s, 1H, H-1), 8.64 (d, *J* = 8.9 Hz, 2H, H-1‴,8‴), 8.42 (d, *J* = 8.8 Hz, 2H, H-4‴,5‴), 8.15 (t, *J* = 7.7 Hz, 2H, H-3‴,6‴), 7.90 (s, 1H, H-4), 7.90 (m, 2H, H-2‴,7‴), 7.57 (d, *J* = 9.0 Hz, 2H, H-2″,6″), 7.12 (d, *J* = 9.0 Hz, 2H, H-3″,5″), 6.88 (d, *J* = 2.3 Hz, 2H, H-2′,6′), 6.23 (t, *J* = 2.3 Hz, 1H, H-4′), 6.04 (s, 2H, H-10), 4.79 (s, 2H, H-3), and 3.70 (s, 6H, 2 × OCH_3_) ppm. ^13^C NMR (151 MHz, DMSO-d_6_): δ 167.5 (C-8), 166.0 (C-2), 165.6 (C-6), 160.5 (C-3′,5′), 160.0 (C-4″), 142.6 (C-9‴), 140.0 (C-1′), 134.3 (C-3‴,6‴), 134.1 (C-4), 132.4 (C-2″,6″), 127.6 (C-2‴,7‴), 125.7 (C-1‴,8‴), 125.6 (C-8‴a,9‴a), 125.4 (C-1″), 117.3 (C-5), 115.7 (C-3″,5″), 97.9 (C-2′,6′), 95.6 (C-4′), 67.0 (C-3), 55.1 (OCH_3_), and 38.5 (C-10) ppm. For C_34_H_29_Cl_2_N_3_O_6_S (690.587 g · mol^−1^) calc.: C 60.18; H 4.23; N 6.08; and S 4.64%, exp.: C 60.01; H 4.23; N 6.02; and S 4.63%.

*2-(4-{[(5Z)-3-[(Acridin-9-yl)methyl]-2,4-dioxo-1,3-thiazolidin-5-ylidene]methyl}phenoxy)-N-(3,4,5-trimethoxyphenyl)acetamide dihydrochloride* (**7c·2HCl**). Yellow solid. Yield: 29 mg (88%). IR ν_max_ 3286, 2928, 2835, 1728, 1704, 1668, 1590, 1558, 1505, 1227, 1181, 1124, 752 cm^−1^. ^1^H NMR (600 MHz, DMSO-d_6_): δ 10.13 (s, 1H, H-1), 8.64 (d, *J* = 8.9 Hz, 2H, H-1‴,8‴), 8.41 (d, *J* = 8.7 Hz, 2H, H-4‴,5‴), 8.14 (t, *J* = 7.8 Hz, 2H, H-3‴,6‴), 7.91 (s, 1H, H-4), 7.88 (t, *J* = 8.1 Hz, 2H, H-2‴,7‴), 7.57 (d, *J* = 9.0 Hz, 2H, H-2″,6″), 7.13 (d, *J* = 9.0 Hz, 2H, H-3″,5″), 7.03 (s, 2H, H-2′,6′), 6.04 (s, 2H, H-10), 4.78 (s, 2H, H-3), 3.75 (s, 6H, 2 × OCH_3_), and 3.61 (s, 3H, OCH_3_) ppm. ^13^C NMR (151 MHz, DMSO-d_6_): δ 167.5 (C-8), 165.6 (C-6), 165.8 (C-2), 160.0 (C-4″), 152.7 (C-3′,5′), 149.3 (C-4‴a,10‴a), 142.5 (C-9‴), 134.5 (C-1′), 134.1 (C-4), 134.1 (C-3‴,6‴), 133.7 (C-4′), 132.4 (C-2″,6″), 127.5 (C-2‴,7‴), 125.7 (C-1″), 125.6 (C-1‴,8‴), 125.4 (C-8‴a,9‴a), 117.3 (C-5), 115.7 (C-3″,5″), 97.3 (C-2′,6′), 67.0 (C-3), 60.1 (OCH_3_), 55.7 (OCH_3_), and 38.3 (C-10) ppm. For C_35_H_31_Cl_2_N_3_O_7_S (708.607 g · mol^−1^) calc.: C 59.33; H 4.41; N 5.93; and S 4.52%, exp.: C 59.30; H 4.41; N 6.00; and S 4.51%.

*2-(4-{[(5Z)-3-[(Acridin-9-yl)methyl]-2,4-dioxo-1,3-thiazolidin-5-ylidene]methyl}phenoxy)-N-(4-nitrophenyl)acetamide dihydrochloride* (**7d·2HCl**). Yellow solid. Yield: 31 mg (91%). IR ν_max_ 3085, 1733, 1711, 1686, 1592, 1566, 1502, 1330, 1250, 1176, 1068, 749 cm^−1^. ^1^H NMR (600 MHz, DMSO-d_6_): δ 10.84 (s, 1H, H-1), 8.59 (d, *J* = 8.9 Hz, 2H, H-1‴,8‴), 8.32 (d, *J* = 8.7 Hz, 2H, H-4‴,5‴), 8.23 (d, *J* = 9.2 Hz, 2H, H-3′,5′), 8.06 (t, *J* = 7.6 Hz, 2H, H-3‴,6‴), 7.90 (s, 1H, H-4), 7.88 (d, *J* = 9.2 Hz, 2H, H-2′,6′), 7.82 (t, *J* = 7.8 Hz, 2H, H-2‴,7‴), 7.57 (d, *J* = 9.0 Hz, 2H, H-2″,6″), 7.14 (d, *J* = 9.0 Hz, 2H, H-3″,5″), 6.01 (s, 2H, H-10), and 4.90 (s, 2H, H-3) ppm. ^13^C NMR (151 MHz, DMSO-d_6_): δ 167.5 (C-8), 167.1 (C-2), 165.6 (C-6), 159.8 (C-4″), 144.5 (C-1′), 142.5 (C-4′), 134.0 (C-4), 132.4 (C-2″,6″), 127.2 (C-2‴,7‴), 125.9 (C-1″), 125.3 (C-8‴a,9‴a), 125.3 (C-1‴,8‴), 125.0 (C-3′,5′), 119.3 (C-2′,6′), 117.4 (C-5), 115.7 (C-3″,5″), 66.9 (C-3), and 38.4 (C-10) ppm. For C_32_H_24_Cl_2_N_4_O_6_S (663.526 g · mol^−1^) calc.: C 57.93; H 3.65; N 8.44; and S 4.83%, exp.: C 58.01; H 3.69; N 8.45; and S 4.90%.

*2-(4-{[(5Z)-3-[(Acridin-9-yl)methyl]-2,4-dioxo-1,3-thiazolidin-5-ylidene]methyl}phenoxy)-N-[2-(trifluoromethyl)phenyl]acetamide dihydrochloride* (**7e·2HCl**). Yellow solid. Yield: 28 mg (83%). IR ν_max_ 3423, 3014, 2912, 1734, 1708, 1681, 1590, 1536, 1509, 1325, 1261, 1063, 1181, 466, 750 cm^−1^. ^1^H NMR (600 MHz, DMSO-d_6_): δ 9.80 (s, 1H, H-1), 8.65 (d, *J* = 8.9 Hz, 2H, H-1‴,8‴), 8.43 (d, *J* = 8.7 Hz, 2H, H-4‴,5‴), 8.15 (t, *J* = 7.7 Hz, 2H, H-3‴,6‴), 7.92 (s, 1H, H-4), 7.90 (t, *J* = 7.9 Hz, 2H, H-2‴,7‴), 7.75 (d, *J* = 7.9 Hz, 1H, H-3′), 7.70 (t, *J* = 7.7 Hz, 1H, H-5′), 7.61 (d, *J* = 8.0 Hz, 1H, H-6′), 7.58 (d, *J* = 8.9 Hz, 2H, H-2″,6″), 7.47 (t, *J* = 7.7 Hz, 1H, H-4′), 7.14 (d, *J* = 8.9 Hz, 2H, H-3″,5″), 6.05 (s, 2H, H-10), and 4.85 (s, 2H, H-3) ppm. ^13^C NMR (151 MHz, DMSO-d_6_): δ 167.5 (C-8),167.1 (C-2), 165.6 (C-6), 159.6 (C-4″), 142.9 (C-9‴), 134.6 (C-1′), 134.3 (C-3‴,6‴), 134.1 (C-4), 133.2 (C-5′), 132.3 (C-2″,6″), 129.3 (C-6′), 127.6 (C-2‴,7‴), 127.0 (C-4′), 126.4 (q, *J* = 4.7 Hz, C-3′), 125.9 (C-1″), 125.6 (C-1‴,8‴), 125.4 (C-8‴a,9‴a), 124.5 (C-2′), 123.5 (q, *J* = 272.3 Hz, CF_3_), 117.5 (C-5), 115.7 (C-3″,5″), 66.8 (C-3), and 38.5 (C-10) ppm. For C_33_H_24_Cl_2_F_3_N_3_O_4_S (686.526 g · mol^−1^) calc.: C 57.73; H 3.52; N 6.12; and S 4.67%, exp.: C 57.75; H 3.51; N 6.12; and S 4.68%.

*2-(4-{[(5Z)-3-[(Acridin-9-yl)methyl]-2,4-dioxo-1,3-thiazolidin-5-ylidene]methyl}phenoxy)-N-[3-(trifluoromethyl)phenyl]acetamide dihydrochloride* (**7f·2HCl**). Yellow solid. Yield: 22 mg (66%). IR ν_max_ 3093, 1732, 1703, 1672, 1591, 1567, 1509, 1332, 1248, 1181, 1067, 753 cm^−1^. ^1^H NMR (600 MHz, DMSO-d_6_): δ 10.57 (s, 1H, H-1), 8.63 (d, *J* = 8.9 Hz, 2H, H-1‴,8‴), 8.39 (d, *J* = 8.7 Hz, 2H, H-4‴,5‴), 8.13 (t, *J* = 7.7 Hz, 2H, H-3‴,6‴), 8.10 (s, 1H, H-2′), 7.90 (s, 1H, H-4), 7.86 (m, 3H, H-6′, H-2‴,7‴), 7.57 (d, *J* = 8.5 Hz, 2H, H-2″,6″), 7.56 (m, 1H, H-5′), 7.43 (d, *J* = 7.8 Hz, 1H, H-4′), 7.15 (d, *J* = 9.0 Hz, 2H, H-3″,5″), 6.03 (s, 2H, H-10), and 4.85 (s, 2H, H-3) ppm. ^13^C NMR (151 MHz, DMSO-d_6_): δ 167.5 (C-8), 166.7 (C-2), 165.6 (C-6), 159.9 (C-4″), 143.2 (C-9‴), 139.1 (C-1′), 134.0 (C-4), 133.5 (C-3‴,6‴), 132.4 (C-2″,6″), 129.9 (C-5′), 129.4 (q, *J* = 31.5 Hz, C-3′), 127.5 (C-2‴,7‴), 125.8 (C-1″), 125.5 (C-1‴,8‴), 125.3 (C-8‴a,9‴a), 124.1 (q, *J* = 272.0 Hz, CF_3_), 123.2 (C-6′), 120.1 (br s, C-4′), 117.4 (C-5), 115.7 (C-3″,5″), 115.7 (br s, C-2′), 66.9 (C-3), and 38.5 (C-10) ppm. For C_33_H_24_Cl_2_F_3_N_3_O_4_S (686.526 g · mol^−1^) vypočítané: C 57.73; H 3.52; N 6.12; and S 4.67%, exp.: C 57.69; H 3.52; N 6.12; and S 4.66%.

*2-(4-{[(5Z)-3-[(Acridin-9-yl)methyl]-2,4-dioxo-1,3-thiazolidin-5-ylidene]methyl}phenoxy)-N-[3,5-bis(trifluoromethyl)phenyl]acetamide dihydrochloride* (**7g·2HCl**). Yellow solid. Yield: 27 mg (81%). IR ν_max_ 3104, 1731, 1717, 1673, 1581, 1510, 1382, 1275, 1178, 1134, 1073, 753 cm^−1^. ^1^H NMR (600 MHz, DMSO-d_6_): δ 10.94 (s, 1H, H-1), 8.64 (d, *J* = 8.9 Hz, 2H, H-1‴,8‴), 8.40 (d, *J* = 8.8 Hz, 2H, H-4‴,5‴), 8.35 (s, 2H, H-2′,6′), 8.14 (t, *J* = 7.7 Hz, 2H, H-3‴,6‴), 7.91 (s, 1H, H-4), 7.88 (t, *J* = 8.0 Hz, 2H, H-2‴,7‴), 7.80 (s, 1H, H-4′), 7.58 (d, *J* = 9.0 Hz, 2H, H-2″,6″), 7.17 (d, *J* = 9.0 Hz, 2H, H-3″,5″), 6.04 (s, 2H, H-10), and 4.89 (s, 2H, H-3) ppm. ^13^C NMR (151 MHz, DMSO-d_6_): δ 167.5 (C-8), 167.3 (C-2), 165.6 (C-6), 159.7 (C-4″), 142.8 (C-9‴), 140.3 (C-1′), 134.0 (C-3‴,6‴), 134.0 (C-4), 132.4 (C-2″,6″), 130.8 (q, *J* = 32.0 Hz, C-3′,5′), 127.5 (C-2‴,7‴), 125.9 (C-1″), 125.6 (C-1‴,8‴), 125.4 (C-8‴a,9‴a), 123.2 (q, *J* = 272.0 Hz, CF_3_), 119.4 (br s, C-2′,6′), 117.4 (C-5), 116.6 (br s, C-4′), 115.7 (C-3″,5″), 66.8 (C-3), and 38.5 (C-10) ppm. For C_34_H_23_Cl_2_F_6_N_3_O_4_S (754.523 g · mol^−1^) calc.: C 54.12; H 3.07; N 5.57; and S 4.25%, exp.: C 54.14; H 3.09; N 5.55; and S 4.25%.

*2-(4-{[(5Z)-3-[(Acridin-4-yl)methyl]-2,4-dioxo-1,3-thiazolidin-5-ylidene]methyl}phenoxy)-N-phenylacetamide hydrochloride* (**8a·HCl**). Yellow solid. Yield: 22 mg (70%). IR ν_max_ 3257, 3036, 1737, 1678, 1594, 1537, 1509, 1249, 1180, 1146, 1066, 749, 717 cm^−1^. ^1^H NMR (600 MHz, DMSO-d_6_): δ 10.22 (s, 1H, H-1), 9.20 (s, 1H, H-9‴), 8.22 (d, *J* = 8.5 Hz, 1H, H-8‴), 8.19 (d, *J* = 8.7 Hz, 1H, H-5‴), 8.14 (dd, *J* = 7.2, 2.7 Hz, 1H, H-1‴), 7.97 (s, 1H, H-4), 7.91 (ddd, *J* = 8.5, 6.6, 1.4 Hz, 1H, H-6‴), 7.68 (m, 1H, H-7‴), 7.67 (d, *J* = 8.7 Hz, 2H, H-2″,6″), 7.64 (d, *J* = 7.5 Hz, 2H, H-2′,6′), 7.57 (m, 2H, H-2‴, H-3‴), 7.33 (t, *J* = 8.0 Hz, 2H, H-3′,5′), 7.19 (d, *J* = 8.8 Hz, 2H, H-3″,5″), 7.09 (t, *J* = 7.4 Hz, 1H, H-4′), 5.61 (s, 2H, H-10), and 4.83 (s, 2H, H-3) ppm. ^13^C NMR (151 MHz, DMSO-d_6_): δ 167.5 (C-8), 166.0 (C-2), 165.9 (C-6), 159.8 (C-4″), 147.4 (C-10‴a), 145.8 (C-4‴a), 138.4 (C-1′), 137.2 (C-9‴), 133.0 (C-4), 132.2 (C-2″,6″), 131.9 (C-4‴), 131.1 (C-6‴), 128.8 (C-3′,5′), 128.8 (C-5‴), 128.5 (C-8‴), 128.2 (C-1‴), 127.3 (C-3‴), 126.2 (C-7‴), 126.1 (C-1″), 126.1 (C-8‴a), 126.0 (C-9‴a), 125.4 (C-2‴), 123.7 (C-4′), 119.6 (C-2′,6′), 118.6 (C-5), 115.7 (C-3″,5″), 67.0 (C-3), and 42.3 (C-10) ppm. For C_32_H_24_ClN_3_O_4_S (582.071 g · mol^−1^) calc.: C 66.03; H 4.16; N 7.22; and S 5.51%, exp.: C 65.57; H 4.12; N 7.16; and S 5.48%.

*2-(4-{[(5Z)-3-[(Acridin-4-yl)methyl]-2,4-dioxo-1,3-thiazolidin-5-ylidene]methyl}phenoxy)-N-(3,5-dimethoxyphenyl)acetamide hydrochloride* (**8b·HCl**). Yellow solid. Yield: 20 mg (64%). IR ν_max_ 2930, 1729, 1678, 1594, 1547, 1509, 1256, 1180, 1148, 1060, 745 cm^−1^. ^1^H NMR (600 MHz, DMSO-d_6_): δ 10.10 (s, 1H, H-1), 9.17 (s, 1H, H-9‴), 8.20 (d, *J* = 8.4 Hz, 1H, H-8‴), 8.17 (dd, *J* = 8.8, 1.0 Hz, 1H, H-5‴), 8.13 (dd, *J* = 7.8, 1.8 Hz, 1H, H-1‴), 7.97 (s, 1H, H-4), 7.89 (ddd, *J* = 8.7, 6.6, 1.5 Hz, 1H, H-6‴), 7.66 (m, 3H, H-2″,6″, H-7‴), 7.63 (d, *J* = 2.3 Hz, 2H, H-2′,6′), 7.55 (m, 2H, H-2‴, H-3‴), 7.18 (d, *J* = 8.8 Hz, 2H, H-3″,5″), 6.25 (t, *J* = 2.3 Hz, 1H, H-4′), 5.60 (s, 2H, H-10), 4.80 (s, 2H, H-3), and 3.71 (s, 6H, 2 × OCH_3_) ppm. ^13^C NMR (151 MHz, DMSO-d_6_): δ 167.5 (C-8), 166.1 (C-2), 165.9 (C-6), 160.5 (C-3′,5′), 159.7 (C-4″), 147.7 (C-10‴a), 146.1 (C-4‴a), 140.0 (C-1′), 136.8 (C-9‴), 133.0 (C-4), 132.2 (C-2″,6″), 132.1 (C-4‴), 130.9 (C-6‴), 129.1 (C-5‴), 128.5 (C-8‴), 128.1 (C-1‴), 127.0 (C-3‴), 126.2 (C-7‴), 126.2 (C-8‴a), 126.1 (C-1″), 126.0 (C-9‴a), 125.3 (C-2‴), 118.6 (C-5), 115.7 (C-3″5″), 97.9 (C-2′,6′), 95.7 (C-4′), 67.0 (C-3), 55.1 (OCH_3_), and 42.3 (C-10) ppm. For C_34_H_28_ClN_3_O_6_S (642.123 g · mol^−1^) calc.: C 63.60; H 4.40; N 6.54; and S 4.99%, exp.: C 62.99; H 4.39; N 6.54; and S 4.97%.

*2-(4-{[(5Z)-3-[(Acridin-4-yl)methyl]-2,4-dioxo-1,3-thiazolidin-5-ylidene]methyl}phenoxy)-N-(3,4,5-trimethoxyphenyl)acetamide hydrochloride* (**8c·HCl**). Yellow solid. Yield: 22 mg (69%). IR ν_max_ 3277, 2939, 2823, 1737, 1674, 1596, 1503, 1227, 1176, 1129, 1064, 733 cm^−1^. ^1^H NMR (600 MHz, DMSO-d_6_): δ 10.08 (s, 1H, H-1), 9.17 (s, 1H, H-9‴), 8.20 (dd, *J* = 8.4, 1.0 Hz, 1H, H-8‴), 8.17 (dd, *J* = 8.8, 1.0 Hz, 1H, H-5‴), 8.13 (dd, *J* = 7.9, 2.0 Hz, 1H, H-1‴), 7.97 (s, 1H, H-4), 7.89 (ddd, *J* = 8.7, 6.6, 1.5 Hz, 1H, H-6‴), 7.66 (m, 3H, H-2″,6″, H-7‴), 7.55 (m, 2H, H-2‴, H-3‴), 7.19 (d, *J* = 8.8 Hz, 2H, H-3″,5″), 7.04 (s, 2H, H-2′,6′), 5.60 (s, 2H, H-10), 4.80 (s, 2H, H-3), 3.74 (s, 6H, 2 × OCH_3_), and 3.62 (s, 3H, OCH_3_) ppm. ^13^C NMR (151 MHz, DMSO-d_6_): δ 167.5 (C-8), 165.9 (C-2), 165.9 (C-6), 159.7 (C-4″), 152.7 (C-3′,5′), 147.7 (C-10‴a), 146.1 (C-4‴a), 136.8 (C-9‴), 134.5 (C-1′), 133.7 (C-4′), 133.0 (C-4), 132.2 (C-2″,6″), 132.1 (C-4‴), 130.9 (C-6‴), 129.1 (C-5‴), 128.5 (C-8‴), 128.1 (C-1‴), 127.0 (C-3‴), 126.2 (C-7‴), 126.2 (C-8‴a), 126.1 (C-1″), 126.0 (C-9‴a), 125.4 (C-2‴), 118.6 (C-5), 115.7 (C-3″,5″), 97.4 (C-2′,6′), 67.0 (C-3), 60.1 (OCH_3_), 55.7 (OCH_3_), and 42.4 (C-10) ppm. For C_35_H_30_ClN_3_O_7_S (672.149 g · mol^−1^) calc.: C 62.54; H 4.50; N 6.25; and S 4.77%, exp.: C 62.54; H 4.50; N 6.23; and S 4.76%.

*2-(4-{[(5Z)-3-[(Acridin-4-yl)methyl]-2,4-dioxo-1,3-thiazolidin-5-ylidene]methyl}phenoxy)-N-(4-nitrophenyl)acetamide hydrochloride* (**8d·HCl**). Yellow solid. Yield: 26 mg (83%). IR ν_max_ 1737, 1682, 1594, 1543, 1501, 1369, 1331, 1250, 1179, 1170, 712 cm^−1^. ^1^H NMR (600 MHz, DMSO-d_6_): δ 10.78 (s, 1H, H-1), 9.17 (s, 1H, H-9‴), 8.25 (d, *J* = 9.3 Hz, 2H, H-3′,5′), 8.21 (d, *J* = 8.4 Hz, 1H, H-8‴), 8.17 (d, *J* = 8.6 Hz, 1H, H-5‴), 8.13 (d, *J* = 7.9 Hz, 1H, H-1‴), 7.97 (s, 1H, H-4), 7.89 (m, 1H, H-6′), 7.89 (d, *J* = 9.3 Hz, 2H, H-2′,6′), 7.67 (m, 3H, H-2″,6″, H-7‴), 7.56 (m, 2H, H-2‴, H-3‴), 7.20 (d, *J* = 8.8 Hz, 2H, H-3″,5″), 5.60 (s, 2H, H-10), and 4.80 (s, 2H, H-3) ppm. ^13^C NMR (151 MHz, DMSO-d_6_): δ 167.5 (C-8), 167.1 (C-2), 165.9 (C-6), 159.6 (C-4″), 147.7 (C-10‴a), 146.1 (C-4‴a), 144.5 (C-1′), 142.6 (C-4′), 136.8 (C-9‴), 133.0 (C-4), 132.2 (C-2″,6″), 132.1 (C-4‴), 130.9 (C-6‴), 129.1 (C-5‴), 128.5 (C-8‴), 128.1 (C-1‴), 127.0 (C-3‴), 126.2 (C-7‴), 126.2 (C-1″), 126.0 (C-8‴a), 126.0 (C-9‴a), 125.3 (C-2‴), 125.0 (C-3′,5′), 119.3 (C-2′,6′), 118.7 (C-5), 115.7 (C-3″,5″), 67.0 (C-3), and 42.3 (C-10) ppm. For C_32_H_23_ClN_4_O_6_S (627.068 g · mol^−1^) calc.: C 61.29; H 3.70; N 8.93; and S 5.11%, exp.: C 60.70; H 3.68; N 8.93; and S 5.10%.

*2-(4-{[(5Z)-3-[(Acridin-4-yl)methyl]-2,4-dioxo-1,3-thiazolidin-5-ylidene]methyl}phenoxy)-N-[2-(trifluoromethyl)phenyl]acetamid hydrochloride* (**8e·HCl**). Yellow solid. Yield: 26 mg (82%). IR ν_max_ 3363, 1735, 1681, 1588, 1505, 1360, 1260, 1166, 1147, 1107, 1058, 756 cm^−1^. ^1^H NMR (600 MHz, DMSO-d_6_): δ 9.80 (s, 1H, H-1), 9.17 (s, 1H, H-9‴), 8.21 (d, *J* = 8.4 Hz, 1H, H-8‴), 8.17 (d, *J* = 8.8 Hz, 1H, H-5‴), 8.13 (dd, *J* = 7.4, 2.5 Hz, 1H, H-1‴), 7.99 (s, 1H, H-4), 7.89 (ddd, *J* = 8.6, 6.6, 1.5 Hz, 1H, H-6‴), 7.77 (d, *J* = 7.8 Hz, 1H, H-3′), 7.72 (t, *J* = 7.5 Hz, 1H, H-5′), 7.68 (d, *J* = 8.8 Hz, 2H, H-2″,6″), 7.65 (m, 4H, H-6′, H-3‴, H-2‴,7‴), 7.49 (t, *J* = 7.6 Hz, 1H, H-4′), 7.19 (d, *J* = 8.8 Hz, 2H, H-3″,5″), 5.61 (s, 2H, H-10), and 4.87 (s, 2H, H-3) ppm. ^13^C NMR (151 MHz, DMSO-d_6_): δ 167.5 (C-8), 167.1 (C-2), 165.9 (C-6), 159.4 (C-4″), 147.7 (C-10‴a), 146.1 (C-4‴a), 136.8 (C-9‴), 134.6 (C-1′), 133.2 (C-5′), 133.0 (C-4), 132.2 (C-4‴), 132.2 (C-2″,6″), 130.9 (C-6‴), 129.3 (C-6′), 129.1 (C-5‴), 128.5 (C-8‴), 128.1 (C-1‴), 127.0 (C-3‴), 126.9 (C-4′), 126.4 (C-3′), 126.4 (C-1″), 126.3 (C-8‴a), 126.2 (C-7‴), 126.0 (C-9‴a), 125.4 (C-2‴), 124.2 (q, *J* = 30.0 Hz, C-2′), 123.6 (q, *J* = 273.4 Hz, CF_3_), 118.8 (C-5), 115.7 (C-3″,5″), 66.8 (C-3), and 42.3 (C-10) ppm. For C_33_H_23_ClF_3_N_3_O_4_S (650.068 g · mol^−1^) calc.: C 60.97; H 3.57; N 6.46; and S 4.93%, exp.: C 60.46; H 3.54; N 6.43; and S 4.90%.

*2-(4-{[(5Z)-3-[(Acridin-4-yl)methyl]-2,4-dioxo-1,3-thiazolidin-5-ylidene]methyl}phenoxy)-N-[3-(trifluoromethyl)phenyl]acetamide hydrochloride* (**8f·HCl**). Yellow solid. Yield: 24 mg (76%). IR ν_max_ 3311, 3034, 1743, 1674, 1598, 1535, 1509, 1335, 1247, 1164, 1140, 1116, 1072, 741 cm^−1^. ^1^H NMR (600 MHz, DMSO-d_6_): δ 10.50 (s, 1H, H-1), 9.17 (s, 1H, H-9‴), 8.20 (d, *J* = 8.4 Hz, 1H, H-8‴), 8.17 (dd, *J* = 8.8, 1.0 Hz, 1H, H-5‴), 8.13 (m, 2H, H-2′, H-1‴), 7.97 (s, 1H, H-4), 7.89 (ddd, *J* = 8.7, 6.6, 1.5 Hz, 1H, H-6‴), 7.86 (d, *J* = 2.1 Hz, 1H, H-6′), 7.67 (d, *J* = 8.9 Hz, 2H, H-2″,6″), 7.66 (ddd, *J* = 8.3, 6.6, 1.2 Hz, 1H, H-7‴), 7.59 (t, *J* = 8.0 Hz, 1H, H-5′), 7.55 (m, 2H, H-2‴,3‴), 7.45 (d, *J* = 7.9 Hz, 1H, H-4′), 7.20 (d, *J* = 8.9 Hz, 2H, H-3″,5″), 5.60 (s, 2H, H-10), and 4.87 (s, 2H, H-3) ppm. ^13^C NMR (151 MHz, DMSO-d_6_): δ 167.5 (C-8), 166.7 (C-2), 165.9 (C-6), 159.6 (C-4″), 147.7 (C-10‴a), 146.1 (C-4‴a), 139.1 (C-1′), 136.8 (C-9‴), 133.0 (C-4), 132.3 (C-2″,6″), 132.2 (C-4‴), 130.9 (C-6‴), 130.1 (C-5′), 129.5 (q, *J* = 31.4 Hz, C-3′), 129.1 (C-5‴), 128.5 (C-8‴), 128.1 (C-1‴), 127.0 (C-3‴), 126.2 (C-7‴), 126.2 (C-8‴a), 126.2 (C-1″), 126.0 (C-9‴a), 125.4 (C-2‴), 124.1 (q, *J* = 272.5 Hz, CF_3_), 123.3 (C-6′), 120.1 (C-4′), 118.7 (C-5), 115.7 (C-2′), 115.7 (C-3″,5″), 67.0 (C-3), and 42.3 (C-10) ppm. For C_33_H_23_ClF_3_N_3_O_4_S (650.068 g · mol^−1^) calc.: C 60.97; H 3.57; N 6.46; and S 4.93%, exp.: C 60.82; H 3.55; N 6.45; and S 4.89%.

*2-(4-{[(5Z)-3-[(Acridin-4-yl)methyl]-2,4-dioxo-1,3-thiazolidin-5-ylidene]methyl}phenoxy)-N-[3-(trifluoromethyl)phenyl]acetamide hydrochloride* (**8g·HCl**). Yellow solid. Yield: 25 mg (79%). IR ν_max_ 2962, 1748, 1703, 1687, 1583, 1510, 1385, 1366, 1278, 1180, 1119, 1070, 756 cm^−1^. ^1^H NMR (600 MHz, DMSO-d_6_): δ 10.78 (s, 1H, H-1), 9.17 (s, 1H, H-9‴), 8.36 (s, 2H, H-2′,6′), 8.20 (d, *J* = 8.5 Hz, 1H, H-8‴), 8.17 (dd, *J* = 8.8, 1.1 Hz, 1H, H-5‴), 8.13 (dd, *J* = 7.8, 2.1 Hz, 1H, H-1‴), 7.98 (s, 1H, H-4), 7.89 (ddd, *J* = 8.5, 6.6, 1.4 Hz, 1H, H-6‴), 7.82 (s, 1H, H-4′), 7.68 (d, *J* = 8.8 Hz, 2H, H-2″,6″), 7.65 (ddd, *J* = 8.5, 6.6, 1.2 Hz, 1H, H-7‴), 7.55 (m, 2H, H-2‴, H-3‴), 7.22 (d, *J* = 8.9 Hz, 2H, H-3″,5″), 5.60 (s, 2H, H-10), and 4.90 (s, 2H, H-3) ppm. ^13^C NMR (151 MHz, DMSO-d_6_): δ 167.4 (C-8), 167.4 (C-2), 165.9 (C-6), 159.5 (C-4″), 147.6 (C-10‴a), 146.1 (C-4‴a), 140.2 (C-1′), 136.8 (C-9‴), 132.9 (C-4), 132.2 (C-4‴), 132.2 (C-2″,6″), 130.9 (C-6‴), 130.8 (q, *J* = 32.9 Hz, C-3′,5′), 129.1 (C-5‴), 128.5 (C-8‴), 128.1 (C-1‴), 127.0 (C-3‴), 126.3 (C-1″), 126.2 (C-7‴), 126.2 (C-8‴a), 126.0 (C-9‴a), 125.3 (C-2‴), 123.2 (q, *J* = 272.9 Hz, CF_3_), 119.5 (br s, C-2′,6′), 118.8 (C-5), 116.7 (br s, C-4′), 115.8 (C-3″,5″), 66.9 (C-3), and 42.3 (C-10) ppm. For C_34_H_22_ClF_6_N_3_O_4_S (720.321 g · mol^−1^) calc.: C 56.69; H 3.08; N 5.83; and S 4.45%, exp.: C 56.96; H 3.07; N 5.81; and S 4.42%.

*2-(4-{[(5Z)-3-[(Acridin-9-yl)methyl]-2,4-dioxo-1,3-thiazolidin-5-ylidene]methyl}-2-methoxyphenoxy)-N-phenylacetamide dihydrochloride* (**12a·2HCl**). Yellow solid. Yield: 26 mg (78%). IR ν_max_ 3395, 1737, 1682, 1589, 1548, 1511, 1259, 1152, 754 cm^−1^. ^1^H NMR (600 MHz, DMSO-d_6_): δ 10.21 (s, 1H, H-1), 8.63 (d, *J* = 8.9 Hz, 2H, H-1‴,8‴), 8.40 (d, *J* = 8.7 Hz, 2H, H-4‴,5‴), 8.13 (t, *J* = 7.7 Hz, 2H, H-3‴,6‴), 7.91 (s, 1H, H-4), 7.88 (t, *J* = 7.7 Hz, 2H, H-2‴,7‴), 7.59 (d, *J* = 8.5 Hz, 2H, H-2′,6′), 7.31 (dd, *J* = 8.5, 7.3 Hz, 2H, H-3′,5′), 7.22 (d, *J* = 2.1 Hz, 1H, H-6″), 7.15 (dd, *J* = 8.5, 2.1 Hz, 1H, H-2″), 7.06 (m, 3H, H-4′, H-3″,5″), 6.04 (s, 2H, H-10), 4.80 (s, 2H, H-3), and 3.83 (s, 3H, OCH_3_) ppm. ^13^C NMR (151 MHz, DMSO-d_6_): δ 167.5 (C-8), 165.9 (C-2), 165.5 (C-6), 149.9 (C-4″), 149.1 (C-5″), 142.8 (C-9‴), 138.4 (C-1′), 134.4 (C-4), 133.9 (C-3‴,6‴), 128.8 (C-3′,5′), 127.5 (C-2‴,7‴), 126.2 (C-1″), 125.5 (C-1‴,8‴), 125.3 (C-8‴a,9‴a), 123.8 (C-2″), 123.6 (C-4′), 119.4 (C-2′,6′), 117.6 (C-5), 114.0 (C-6″), 113.7 (C-3″), 67.6 (C-3), 55.7 (OCH_3_), and 38.5 (C-10) ppm. For C_33_H_27_Cl_2_N_3_O_5_S (648.555 g · mol^−1^) calc.: C 61.11; H 4.20; N 6.48; S 4.94%, exp.: C 60.38; H 4.16; N 6.37; and S 4.92%.

*2-(4-{[(5Z)-3-[(Acridin-9-yl)methyl]-2,4-dioxo-1,3-thiazolidin-5-ylidene]methyl}-2-methoxyphenoxy)-N-(3,5-dimethoxyphenyl)acetamide dihydrochloride* (**12b·2HCl**). Yellow solid. Yield: 26 mg (78%). IR ν_max_ 3384, 2945, 1737, 1682, 1610, 1552, 1518, 1265, 1149, 753 cm^−1^. ^1^H NMR (600 MHz, DMSO-d_6_): δ 10.17 (s, 1H, H-1), 8.63 (d, *J* = 8.9 Hz, 2H, H-1‴,8‴), 8.41 (d, *J* = 8.8 Hz, 2H, H-4‴,5‴), 8.14 (t, *J* = 7.4 Hz, 2H, H-3‴,6‴), 7.91 (s, 1H, H-4), 7.88 (t, *J* = 7.7 Hz, 2H, H-2‴,7‴), 7.22 (d, *J* = 2.2 Hz, 1H, H-6″), 7.15 (dd, *J* = 8.6, 2.2 Hz, 1H, H-2″), 7.04 (d, *J* = 8.6 Hz, 1H, H-3″), 6.85 (d, *J* = 2.2 Hz, 2H, H-2′,6′), 6.23 (t, *J* = 2.2 Hz, 1H, H-4′), 6.04 (s, 2H, H-10), 4.78 (s, 2H, H-3), 3.83 (s, 3H, OCH_3_), and 3.69 (s, 6H, 2 × OCH_3_) ppm. ^13^C NMR (151 MHz, DMSO-d_6_): δ 167.5 (C-8), 166.0 (C-2), 165.5 (C-6), 160.5 (C-3′,5′), 149.8 (C-4″), 149.1 (C-5″), 142.9 (C-9‴), 140.0 (C-1′), 134.4 (C-4), 134.0 (C-3‴,6‴), 127.5 (C-2‴,7‴), 126.2 (C-1″), 125.6 (C-1‴,8‴), 125.3 (C-8‴a,9‴a), 123.8 (C-2″), 117.6 (C-5), 114.0 (C-6″), 113.7 (C-3″), 97.6 (C-2′,6′), 95.6 (C-4′), 67.5 (C-3), 55.7 (OCH_3_), 55.1 (OCH_3_), and 38.5 (C-10) ppm. For C_35_H_31_Cl_2_N_3_O_7_S (708.607 g · mol^−1^) calc.: C 59.33; H 4.41; N 5.93; S 4.52%, exp.: C 59.59; H 4.39; N 5.91; and S 4.57%.

*2-(4-{[(5Z)-3-[(Acridin-9-yl)methyl]-2,4-dioxo-1,3-thiazolidin-5-ylidene]methyl}-2-methoxyphenoxy)-N-(3,4,5-trimethoxyphenyl)acetamide dihydrochloride* (**12c·2HCl**). Yellow solid. Yield: 26 mg (78%). IR ν_max_ 2940, 1733, 1681, 1589, 1505, 1261, 1223, 1182, 1125, 756 cm^−1^. ^1^H NMR (600 MHz, DMSO-d_6_): δ 10.15 (s, 1H, H-1), 8.63 (d, *J* = 8.9 Hz, 2H, H-1‴,8‴), 8.40 (d, *J* = 8.7 Hz, 2H, H-4‴,5‴), 8.13 (t, *J* = 7.8 Hz, 2H, H-3‴,6‴), 7.91 (s, 1H, H-4), 7.88 (t, *J* = 7.8 Hz, 2H, H-2‴,7‴), 7.22 (d, *J* = 2.2 Hz, 1H, H-6″), 7.15 (dd, *J* = 8.6, 2.2 Hz, 1H, H-1″), 7.05 (d, *J* = 8.6 Hz, 1H, H-3″), 6.99 (s, 2H, H-2′,6′), 6.04 (s, 2H, H-10), 4.78 (s, 2H, H-3), 3.83 (s, 3H, OCH_3_), 3.72 (s, 6H, 2 × OCH_3_), and 3.60 (s, 3H, OCH_3_) ppm. ^13^C NMR (151 MHz, DMSO-d_6_): δ 167.5 (C-8), 165.8 (C-2), 165.5 (C-6), 152.7 (C-3′,5′), 149.9 (C-4″), 149.2 (C-5″), 143.0 (C-9‴), 134.5 (C-1′), 134.4 (C-4), 134.0 (C-3‴,6‴), 133.6 (C-4′), 127.5 (C-2‴,7‴), 126.2 (C-1″), 125.5 (C-1‴,8‴), 125.3 (C-8‴a,9‴a), 123.8 (C-2″), 117.6 (C-5), 113.9 (C-6″), 113.8 (C-3″), 97.1 (C-2′,6′), 67.6 (C-3), 60.1 (OCH_3_), 55.7 (OCH_3_), 55.7 (OCH_3_), and 38.5 (C-10) ppm. For C_36_H_33_Cl_2_N_3_O_8_S (738.633 g · mol^−1^) calc.: C 58.54; H 4.50; N 5.69; and S 4.34%, exp.: C 57.99; H 4.49; N 5.69; and S 4.31%.

*2-(4-{[(5Z)-3-[(Acridin-9-yl)methyl]-2,4-dioxo-1,3-thiazolidin-5-ylidene]methyl}-2-methoxyphenoxy)-N-(4-nitrophenyl)acetamide dihydrochloride* (**12d·2HCl**). Yellow solid. Yield: 26 mg (78%). IR ν_max_ 2916, 1732, 1680, 1586, 1562, 1510, 1326, 1255, 1175, 1150, 751 cm^−1^. ^1^H NMR (600 MHz, DMSO-d_6_): δ 10.93 (s, 1H, H-1), 8.65 (d, *J* = 8.9 Hz, 2H, H-1‴,8‴), 8.42 (d, *J* = 8.7 Hz, 2H, H-4‴,5‴), 8.23 (d, *J* = 9.2 Hz, 2H, H-3′,5′), 8.15 (t, *J* = 7.8 Hz, 2H, H-3‴,6‴), 7.91 (s, 1H, H-4), 7.89 (t, *J* = 7.8 Hz, 2H, H-2‴,7‴), 7.23 (d, *J* = 2.2 Hz, 1H, H-6″), 7.15 (dd, *J* = 8.6, 2.2 Hz, 1H, H-2″), 7.07 (d, *J* = 8.6 Hz, 1H, H-3″), 6.86 (d, *J* = 9.2 Hz, 2H, H-2′,6′), 6.05 (s, 2H, H-10), 4.90 (s, 2H, H-3), and 3.83 (s, 3H, OCH_3_) ppm. ^13^C NMR (151 MHz, DMSO-d_6_): δ 167.5 (C-8), 167.1 (C-2), 165.5 (C-6), 149.7 (C-4″), 149.1 (C-5″), 144.6 (C-1′), 142.5 (C-4′), 134.4 (C-4), 134.0 (C-3‴,6‴), 127.6 (C-2‴,7‴), 126.3 (C-1″), 125.6 (C-1‴,8‴), 125.4 (C-8‴a,9‴a), 125.0 (C-3′,5′), 123.8 (C-2″), 119.1 (C-2′,6′), 117.7 (C-5), 114.0 (C-6″), 113.8 (C-3″), 67.4 (C-3), 55.7 (OCH_3_), and 38.5 (C-10) ppm. For C_33_H_26_Cl_2_N_4_O_7_S (693.552 g · mol^−1^) calc.: C 57.15; H 3.78; N 8.08; and S 4.62%, exp.: C 57.12; H 3.76; N 8.00; and S 4.59%.

*2-(4-{[(5Z)-3-[(Acridin-9-yl)methyl]-2,4-dioxo-1,3-thiazolidin-5-ylidene]methyl}-2-methoxyphenoxy)-N-[2-(trifluoromethyl)phenyl]acetamide dihydrochloride* (**12e·2HCl**). Yellow solid. Yield: 26 mg (78%). IR ν_max_ 3414, 3015, 1750, 1699, 1681, 1593, 1539, 1514, 1334, 1298, 1175, 1147, 756 cm^−1^. ^1^H NMR (600 MHz, DMSO-d_6_): δ 9.58 (s, 1H, H-1), 8.63 (d, *J* = 8.9 Hz, 2H, H-1‴,8‴), 8.40 (d, *J* = 8.8 Hz, 2H, H-4‴,5‴), 8.13 (t, *J* = 7.8 Hz, 2H, H-3‴,6‴), 7.92 (s, 1H, H-4), 7.88 (t, *J* = 7.8 Hz, 2H, H-2‴,7‴), 7.80 (d, *J* = 8.1 Hz, 1H, H-6′), 7.75 (dd, *J* = 8.0, 1.5 Hz, 1H, H-3′), 7.70 (td, *J* = 7.8, 1.5 Hz, 1H, H-5′), 7.44 (t, *J* = 7.7 Hz, 1H, H-4′), 7.24 (d, *J* = 2.2 Hz, 1H, H-6″), 7.17 (dd, *J* = 8.6, 2.2 Hz, 1H, H-2″), 7.11 (d, *J* = 8.5 Hz, 1H, H-3″), 6.04 (s, 2H, H-10), 4.85 (s, 2H, H-3), and 3.83 (s, 3H, OCH_3_) ppm. ^13^C NMR (151 MHz, DMSO-d_6_): δ 167.5 (C-8), 166.9 (C-2), 165.5 (C-6), 149.2 (C-5″), 149.2 (C-4″), 134.5 (C-1′), 134.4 (C-4), 134.0 (C-3‴,6‴), 133.0 (C-5′), 127.6 (br s, C-6′), 127.5 (C-2‴,7‴), 126.5 (C-1″), 126.3 (C-3′), 126.3 (C-4′), 125.5 (C-1‴,8‴), 125.3 (C-8‴a,9‴a), 123.7 (C-2″), 123.6 (q, *J* = 273.3 Hz, CF_3_), 122.7 (C-2′), 117.9 (C-5), 113.9 (C-3″), 113.8 (C-6″), 67.3 (C-3), 55.7 (OCH_3_), and 38.5 (C-10) ppm. For C_34_H_26_Cl_2_F_3_N_3_O_5_S (716.552 g · mol^−1^) calc.: C 56.99; H 3.66; N 5.86; and S 4.47%, exp.: C 56.89; H 3.66; N 5.90; and S 4.49%.

*2-(4-{[(5Z)-3-[(Acridin-9-yl)methyl]-2,4-dioxo-1,3-thiazolidin-5-ylidene]methyl}-2-methoxyphenoxy)-N-[3-(trifluoromethyl)phenyl]acetamide dihydrochloride* (**12f·2HCl**). Yellow solid. Yield: 27 mg (81%). IR ν_max_ 3387, 3015, 1736, 1682, 1589, 1512, 1329, 1261, 1127, 753 cm^−1^. ^1^H NMR (600 MHz, DMSO-d_6_): δ 10.61 (s, 1H, H-1), 8.62 (d, *J* = 8.9 Hz, 2H, H-1‴,8‴), 8.37 (d, *J* = 8.9 Hz, 2H, H-4‴,5‴), 8.10 (t, *J* = 7.7 Hz, 2H, H-3‴,6‴), 8.10 (s, 1H, H-2′), 7.91 (s, 1H, H-4), 7.86 (t, *J* = 7.7 Hz, 2H, H-2‴,7‴), 7.81 (dd, *J* = 8.1, 2.1 Hz, 1H, H-6′), 7.56 (t, *J* = 8.0 Hz, 1H, H-5′), 7.42 (d, *J* = 7.9 Hz, 1H, H-4′), 7.23 (d, *J* = 2.2 Hz, 1H, H-6″), 7.15 (dd, *J* = 8.6, 2.2 Hz, 1H, H-2″), 7.07 (d, *J* = 8.5 Hz, 1H, H-3″), 6.03 (s, 2H, H-10), 4.85 (s, 2H, H-3), and 3.83 (s, 3H, OCH_3_) ppm. ^13^C NMR (151 MHz, DMSO-d_6_): δ 167.5 (C-8), 166.7 (C-2), 165.6 (C-6), 149.8 (C-4″), 149.2 (C-5″), 139.2 (C-1′), 134.4 (C-4), 134.0 (C-3‴,6‴), 130.1 (C-5′), 129.5 (q, *J* = 32.0, C-3′), 127.4 (C-2‴,7‴), 126.3 (C-1″), 125.5 (C-1‴,8‴), 125.3 (C-8‴a,9‴a), 124.0 (q, *J* = 272.0 Hz, CF_3_), 123.8 (C-2″), 123.0 (C-6′), 120.0 (C-4′), 117.7 (C-5), 115.4 (br s, C-2′), 114.0 (C-6″), 113.8 (C-3″), 67.5 (C-3), 55.7 (OCH_3_), and 38.5 (C-10) ppm. For C_34_H_26_Cl_2_F_3_N_3_O_5_S (716.552 g · mol^−1^) calc.: C 56.99; H 3.66; N 5.86; and S 4.47%, exp.: C 57.45; H 3.69; N 5.85; and S 4.46%.

*2-(4-{[(5Z)-3-[(Acridin-9-yl)methyl]-2,4-dioxo-1,3-thiazolidin-5-ylidene]methyl}-2-methoxyphenoxy)-N-[3,5-bis(trifluoromethyl)phenyl]acetamide dihydrochloride* (**12g·2HCl**). Yellow solid. Yield: 27 mg (82%). IR ν_max_ 3387, 1739, 1709, 1686, 1592, 1543, 1512, 1380, 1285, 1172, 1128, 752 cm^−1^. ^1^H NMR (600 MHz, DMSO-d_6_): δ 11.01 (s, 1H, H-1), 8.63 (d, *J* = 8.9 Hz, 2H, H-1‴,8‴), 8.39 (d, *J* = 8.8 Hz, 2H, H-4‴,5‴), 8.31 (s, 2H, H-2′,6′), 8.13 (t, *J* = 7.8 Hz, 2H, H-3‴,6‴), 7.91 (s, 1H, H-4), 7.87 (t, *J* = 7.8 Hz, 2H, H-2‴,7‴), 7.79 (s, 1H, H-4′), 7.23 (d, *J* = 2.2 Hz, 1H, H-6″), 7.14 (dd, *J* = 8.6, 2.2 Hz, 1H, H-2″), 7.09 (d, *J* = 8.5 Hz, 1H, H-3″), 6.04 (s, 2H, H-10), 4.89 (s, 2H, H-3), and 3.83 (s, 3H, OCH_3_) ppm. ^13^C NMR (151 MHz, DMSO-d_6_): δ 167.5 (C-2), 167.4 (C-8), 165.5 (C-6), 149.6 (C-4″), 149.2 (C-5″), 147.8 (C-4‴a,10‴a), 143.1 (C-9‴), 140.3 (C-1′), 134.4 (C-4), 134.0 (C-3‴,6‴), 130.8 (q, *J* = 32.9 Hz, C-3′,5′), 127.5 (C-2‴,7‴), 126.4 (C-1″), 125.5 (C-1‴,8‴), 125.3 (C-8‴a,9‴a), 123.7 (C-2″), 123.2 (q, *J* = 273.0 Hz, CF_3_), 119.2 (br s, C-2′,6′), 117.8 (C-5), 116.5 (C-4′), 114.1 (C-6″), 114.0 (C-3″), 67.4 (C-3), 55.7 (OCH_3_), and 38.5 (C-10) ppm. For C_35_H_25_Cl_2_F_6_N_3_O_5_S (784.549 g · mol^−1^) calc.: C 53.58; H 3.21; N 5.36; and S 4.09%, exp.: C 53.63; H 3.18; N 5.36; and S 4.09%.

*2-(4-{[(5Z)-3-[(Acridin-4-yl)methyl]-2,4-dioxo-1,3-thiazolidin-5-ylidene]methyl}-2-methoxyphenoxy)-N-phenylacetamide hydrochloride* (**13a·HCl**). Yellow solid. Yield: 26 mg (80%). IR ν_max_ 3389, 1732, 1686, 1602, 1543, 1513, 1269, 1144, 741 cm^−1^. ^1^H NMR (600 MHz, DMSO-d_6_): δ 10.15 (s, 1H, H-1), 9.17 (s, 1H, H-9‴), 8.20 (d, *J* = 9.0 Hz, 1H, H-8‴), 8.17 (d, *J* = 8.7 Hz, 1H, H-5‴), 8.13 (dd, *J* = 8.1, 1.9 Hz, 1H, H-1‴), 7.97 (s, 1H, H-4), 7.89 (ddd, *J* = 8.8, 6.6, 1.3 Hz, 1H, H-6‴), 7.66 (ddd, *J* = 8.2, 6.7, 1.1 Hz, 1H, H-7‴), 7.62 (d, *J* = 8.6 Hz, 2H, H-2′,6′), 7.55 (m, 2H, H-2‴, H-3‴), 7.33 (t, *J* = 8.2 Hz, 2H, H-3′,5′), 7.33 (br s, 1H, H-6″), 7.25 (dd, *J* = 8.6, 2.1 Hz, 1H, H-2″), 7.12 (d, *J* = 8.5 Hz, 1H, H-3″), 7.08 (t, *J* = 7.4 Hz, 1H, H-4′), 5.60 (s, 2H, H-10), 4.82 (S, 2H, H-3), and 3.88 (s, 3H, OCH_3_) ppm. ^13^C NMR (151 MHz, DMSO-d_6_): δ 167.5 (C-8), 166.0 (C-2), 165.8 (C-6), 149.6 (C-4″), 149.2 (C-5″), 147.6 (C-10‴a), 146.0 (C-4‴a), 138.4 (C-1′), 136.8 (C-9‴), 133.3 (C-4), 132.1 (C-4‴), 130.9 (C-6‴), 129.1 (C-5‴), 128.8 (C-3′,5′), 128.5 (C-8‴), 128.1 (C-1‴), 127.0 (C-3‴), 126.6 (C-1″), 126.2 (C-7‴), 126.2 (C-8‴a), 126.0 (C-9‴a), 125.3 (C-2‴), 123.7 (C-4′), 123.5 (C-2″), 119.4 (C-2′,6′), 118.9 (C-5), 114.0 (C-6″), 113.8 (C-3″), 67.7 (C-3), 55.7 (OCH_3_), and 42.3 (C-10) ppm. For C_33_H_26_ClN_3_O_5_S (612.097 g · mol^−1^) calc.: C 64.75; H 4.28; N 6.87; and S 5.24%, exp.: C 64.38; H 4.24; N 6.83; and S 5.19%.

*2-(4-{[(5Z)-3-[(Acridin-4-yl)methyl]-2,4-dioxo-1,3-thiazolidin-5-ylidene]methyl}-2-methoxyphenoxy)-N-(3,5-dimethoxyphenyl)acetamide hydrochloride* (**13b·HCl**). Yellow solid. Yield: 25 mg (79%). IR ν_max_ 3351, 2936, 1730, 1703, 1667, 1604, 1555, 1510, 1271, 1202, 1150, 1061, 743 cm^−1^. ^1^H NMR (600 MHz, DMSO-d_6_): δ 10.13 (s, 1H, H-1), 9.17 (s, 1H, H-9‴), 8.21 (d, *J* = 8.3 Hz, 1H, H-8‴), 8.17 (d, *J* = 8.8 Hz, 1H, H-5‴), 8.13 (dd, *J* = 8.3, 1.8 Hz, 1H, H-1‴), 7.97 (s, 1H, H-4), 7.90 (ddd, *J* = 8.4, 6.6, 1.4 Hz, 1H, H-6‴), 7.66 (ddd, *J* = 8.0, 6.6, 1.2 Hz, 1H, H-7‴), 7.56 (m, 2H, H-2‴, H-3‴), 7.33 (d, *J* = 2.1 Hz, 1H, H-6″), 7.25 (dd, *J* = 8.6, 2.1 Hz, 1H, H-2″), 7.10 (d, *J* = 8.5 Hz, 1H, H-3″), 6.87 (d, *J* = 2.2 Hz, 2H, H-2′,6′), 6.25 (t, *J* = 2.2 Hz, 1H, H-4′), 5.60 (s, 2H, H-10), 4.80 (s, 2H, H-3), 3.88 (s, 3H, OCH_3_), and 3.71 (s, 6H, 2 × OCH_3_) ppm. ^13^C NMR (151 MHz, DMSO-d_6_): δ 167.5 (C-8), 166.1 (C-2), 165.8 (C-6), 160.5 (C-3′,5′), 149.6 (C-4″), 149.2 (C-5″), 147.6 (C-10‴a), 146.0 (C-4‴a), 140.0 (C-1′), 136.8 (C-9‴), 133.3 (C-4), 132.1 (C-4‴), 130.9 (C-6‴), 129.0 (C-5‴), 128.5 (C-8‴), 128.1 (C-1‴), 127.0 (C-3‴), 126.6 (C-1″), 126.2 (C-7‴), 126.2 (C-8‴a), 126.0 (C-9‴a), 125.3 (C-2‴), 123.5 (C-2″), 118.9 (C-5), 114.0 (C-6″), 113.8 (C-3″), 97.7 (C-2′,6′), 95.6 (C-4′), 67.7 (C-3), 55.7 (OCH_3_), 55.1 (OCH_3_), and 42.3 (C-10) ppm. For C_35_H_30_ClN_3_O_7_S (672.149 g · mol^−1^) calc.: C 62.54; H 4.50; N 6.25; and S 4.77%, exp.: C 62.85; H 4.53; N 6.30; and S 4.73%.

*2-(4-{[(5Z)-3-[(Acridin-4-yl)methyl]-2,4-dioxo-1,3-thiazolidin-5-ylidene]methyl}-2-methoxyphenoxy)-N-(3,4,5-trimethoxyphenyl)acetamide hydrochloride* (**13c·HCl**). Yellow solid. Yield: 27 mg (85%). IR ν_max_ 3361, 2929, 1736, 1677, 1606, 1538, 1504, 1279, 1232, 1148, 1129, 1048, 739 cm^−1^. ^1^H NMR (600 MHz, DMSO-d_6_): δ 10.10 (s, 1H, H-1), 9.17 (s, 1H, H-9‴), 8.20 (d, *J* = 8.2 Hz, 1H, H-8‴), 8.17 (d, *J* = 8.8 Hz, 1H, H-5‴), 8.13 (dd, *J* = 8.1, 1.8 Hz, 1H, H-1‴), 7.97 (s, 1H, H-4), 7.89 (ddd, *J* = 8.4, 6.6, 1.4 Hz, 1H, H-6‴), 7.66 (ddd, *J* = 8.0, 6.6, 1.1 Hz, 1H, H-7‴), 7.56 (m, 2H, H-2‴,3‴), 7.33 (d, *J* = 2.1 Hz, 1H, H-6″), 7.25 (dd, *J* = 8.5, 2.1 Hz, 1H, H-2″), 7.11 (d, *J* = 8.5 Hz, 1H, H-3″), 7.02 (s, 2H, H-2′,6′), 5.60 (s, 2H, H-10), 4.80 (s, 2H, H-3), 3.88 (s, 3H, OCH_3_), 3.74 (s, 6H, 2 × OCH_3_), and 3.62 (s, 3H, OCH_3_) ppm. ^13^C NMR (151 MHz, DMSO-d_6_): δ 167.5 (C-8), 165.9 (C-2), 165.8 (C-6), 152.7 (C-3′,5′), 149.6 (C-4″), 149.2 (C-5″), 147.6 (C-10‴a), 146.0 (C-4‴a), 136.8 (C-9‴), 134.5 (C-1′), 133.7 (C-4′), 133.3 (C-4), 132.1 (C-4‴), 130.9 (C-6‴), 129.1 (C-5‴), 128.5 (C-8‴), 128.1 (C-1‴), 127.0 (C-3‴), 126.6 (C-1″), 126.2 (C-7‴), 126.2 (C-8‴a), 126.0 (C-9‴a), 125.3 (C-2‴), 123.5 (C-2″), 118.9 (C-5), 114.0 (C-6″), 113.8 (C-3″), 97.1 (C-2′,6′), 67.7 (C-3), 60.1 (OCH_3_), 55.7 (OCH_3_), 55.7 (OCH_3_), and 42.3 (C-10) ppm. For C_36_H_32_ClN_3_O_8_S (702.175 g · mol^−1^) calc.: C 61.58; H 4.59; N 5.98; and S 4.57%, exp.: C 61.86; H 4.60; N 6.01; and S 4.55%.

*2-(4-{[(5Z)-3-[(Acridin-4-yl)methyl]-2,4-dioxo-1,3-thiazolidin-5-ylidene]methyl}-2-methoxyphenoxy)-N-(4-nitrophenyl)acetamide hydrochloride* (**13d·HCl**). Yellow solid. Yield: 24 mg (76%). IR ν_max_ 3367, 1738, 1712, 1679, 1595, 1545, 1508, 1338, 1265, 1144, 1058, 741 cm^−1^. ^1^H NMR (600 MHz, DMSO-d_6_): δ 10.85 (s, 1H, H-1), 9.18 (s, 1H, H-9‴), 8.25 (d, *J* = 9.3 Hz, 2H, H-3′,5′), 8.21 (d, *J* = 8.4 Hz, 1H, H-8‴), 8.17 (d, *J* = 8.8 Hz, 1H, H-5‴), 8.13 (dd, *J* = 8.2, 2.0 Hz, 1H, H-1‴), 7.97 (s, 1H, H-4), 7.90 (ddd, *J* = 8.4, 6.6, 1.4 Hz, 1H, H-6‴), 7.88 (d, *J* = 9.2 Hz, 2H, H-2′,6′), 7.66 (ddd, *J* = 8.2, 6.7, 1.2 Hz, 1H, H-7‴), 7.56 (m, 2H, H-2‴,3‴), 7.33 (d, *J* = 2.1 Hz, 1H, H-6″), 7.24 (dd, *J* = 8.5, 2.1 Hz, 1H, H-2″), 7.12 (d, *J* = 8.5 Hz, 1H, H-3″), 5.60 (s, 2H, H-10), 4.91 (s, 2H, H-3), and 3.88 (s, 3H, OCH_3_) ppm. ^13^C NMR (151 MHz, DMSO-d_6_): δ 167.5 (C-8), 167.1 (C-2), 165.8 (C-6), 149.5 (C-4″), 149.2 (C-5″), 147.7 (C-10‴a), 146.0 (C-4‴a), 144.6 (C-1′), 142.5 (C-4′), 136.9 (C-9‴), 133.3 (C-4), 132.1 (C-4‴), 131.0 (C-6‴), 129.0 (C-5‴), 128.5 (C-8‴), 128.2 (C-1‴), 127.1 (C-3‴), 126.7 (C-1″), 126.2 (C-7‴), 126.2 (C-8‴a), 126.0 (C-9‴a), 125.4 (C-2‴), 125.0 (C-3′,5′), 123.5 (C-2″), 119.1 (C-2′,6′), 119.0 (C-5), 114.1 (C-6″), 113.9 (C-3″), 67.6 (C-3), 55.7 (OCH_3_), and 42.3 (C-10) ppm. For C_33_H_25_ClN_4_O_7_S (657.094 g · mol^−1^) calc.: C 60.32; H 3.84; N 8.53; and S 4.88%, exp.: C 60.86; H 3.90; N 8.51; and S 4.88%.

*2-(4-{[(5Z)-3-[(Acridin-4-yl)methyl]-2,4-dioxo-1,3-thiazolidin-5-ylidene]methyl}-2-methoxyphenoxy)-N-[2-(trifluoromethyl)phenyl]acetamide hydrochloride* (**13e·HCl**). Yellow solid. Yield: 26 mg (82%). IR ν_max_ 3401, 1733, 1677, 1590, 1543, 1511, 1365, 1268, 1142, 1101, 1030, 738 cm^−1^. ^1^H NMR (600 MHz, DMSO-d_6_): δ 9.59 (s, 1H, H-1), 9.18 (s, 1H, H-9‴), 8.21 (d, *J* = 8.4 Hz, 1H, H-8‴), 8.18 (d, *J* = 8.8 Hz, 1H, H-5‴), 8.14 (dd, *J* = 8.0, 2.5 Hz, 1H, H-1‴), 7.99 (s, 1H, H-4), 7.90 (ddd, *J* = 8.8, 6.6, 1.4 Hz, 1H, H-6‴), 7.84 (d, *J* = 8.1 Hz, 1H, H-6′), 7.77 (d, *J* = 7.3 Hz, 1H, H-3′), 7.72 (t, *J* = 7.5 Hz, 1H, H-5′), 7.67 (ddd, *J* = 8.2, 6.6, 1.1 Hz, 1H, H-7‴), 7.57 (m, 2H, H-2‴, H-3‴), 7.46 (t, *J* = 7.7 Hz, 1H, H-4′), 7.33 (d, *J* = 2.2 Hz, 1H, H-6″), 7.24 (dd, *J* = 8.5, 2.1 Hz, 1H, H-2″), 7.12 (d, *J* = 8.5 Hz, 1H, H-3″), 5.61 (s, 2H, H-10), 4.87 (s, 2H, H-3), and 3.88 (s, 3H, OCH_3_) ppm. ^13^C NMR (151 MHz, DMSO-d_6_): δ 167.4 (C-8), 166.9 (C-2), 165.8 (C-6), 149.2 (C-5″), 148.9 (C-4″), 147.6 (C-10‴a), 146.0 (C-4‴a), 136.9 (C-9‴), 134.5 (C-1′), 133.3 (C-4), 133.3 (C-5′), 132.1 (C-4‴), 131.0 (C-6‴), 129.0 (C-5‴), 128.5 (C-8‴), 128.2 (C-1‴), 127.6 (br s, C-6′), 127.1 (C-3‴), 126.9 (C-1″), 126.3 (C-3′), 126.3 (C-4′), 126.2 (C-7‴), 126.2 (C-8‴a), 126.0 (C-9‴a), 125.4 (C-2‴), 123.7 (q, *J* = 273.2 Hz, CF_3_), 123.4 (C-2″), 122.7 (C-2′), 119.2 (C-5), 114.0 (C-3″), 113.9 (C-6″), 67.4 (C-3), 55.8 (OCH_3_), and 42.3 (C-10) ppm. For C_34_H_25_ClF_3_N_3_O_5_S (680.094 g · mol^−1^) calc.: C 60.05; H 3.71; N 6.18; and S 4.71%, exp.: C 59.99; H 3.69; N 6.20; and S 4.71%.

*2-(4-{[(5Z)-3-[(Acridin-4-yl)methyl]-2,4-dioxo-1,3-thiazolidin-5-ylidene]methyl}-2-methoxyphenoxy)-N-[3-(trifluoromethyl)phenyl]acetamide hydrochloride* (**13f·HCl**). Yellow solid. Yield: 20 mg (62%). IR ν_max_ 3349, 2917, 2849, 1728, 1709, 1665, 1607, 1562, 1510, 1334, 1269, 1152, 1118, 1072, 744 cm^−1^. ^1^H NMR (600 MHz, DMSO-d_6_): δ 10.53 (s, 1H, H-1), 9.17 (s, 1H, H-9‴), 8.21 (d, *J* = 8.3 Hz, 1H, H-8‴), 8.17 (dd, *J* = 8.8, 1.0 Hz, 1H, H-5‴), 8.13 (dd, *J* = 8.2, 1.8 Hz, 1H, H-1‴), 8.11 (t, *J* = 8.0 Hz, 1H, H-2′), 7.97 (s, 1H, H-4), 7.89 (ddd, *J* = 8.8, 6.6, 1.5 Hz, 1H, H-6‴), 7.83 (d, *J* = 8.6 Hz, 1H, H-6′), 7.66 (ddd, *J* = 8.2, 6.7, 1.1 Hz, 1H, H-7‴), 7.56 (m, 4H, H-3′,5′, H-2‴, H-3‴), 7.44 (d, *J* = 7.8 Hz, 1H, H-4′), 7.34 (d, *J* = 2.1 Hz, H-6″), 7.25 (dd, *J* = 8.4, 2.1 Hz, 1H, H-2″), 7.13 (d, *J* = 8.5 Hz, 1H, H-3″), 5.60 (s, 2H, H-10), 4.86 (s, 2H, H-3), and 3.88 (s, 3H, OCH_3_) ppm. ^13^C NMR (151 MHz, DMSO-d_6_): δ 167.5 (C-8), 166.8 (C-2), 165.8 (C-6), 149.5 (C-4″), 149.2 (C-5″), 147.6 (C-10‴a), 146.0 (C-4‴a), 139.2 (C-1′), 136.8 (C-9‴), 133.3 (C-4), 132.1 (C-4‴), 130.9 (C-6‴), 130.1 (C-5′), 129.5 (q, *J* = 31.5 Hz, C-3′), 129.1 (C-5‴), 128.5 (C-8‴), 128.1 (C-1‴), 127.0 (C-3‴), 126.7 (C-1″), 126.2 (C-7‴), 126.2 (C-8‴a), 126.0 (C-9‴a), 125.3 (C-2‴), 124.1 (q, *J* = 272.3 Hz, CF_3_), 123.5 (C-2″), 123.0 (C-6′), 120.0 (br s, C-4′), 119.0 (C-5), 115.5 (br s, C-2′), 114.1 (C-6″), 113.9 (C-3″), 67.6 (C-3), 55.7 (OCH_3_), and 42.3 (C-10) ppm. For C_34_H_25_ClF_3_N_3_O_5_S (680.094 g · mol^−1^) calc.: C 60.05; H 3.71; N 6.18; and S 4.71%, exp.: C 60.09; H 3.70; N 6.20; and S 4.71%.

*2-(4-{[(5Z)-3-[(Acridin-4-yl)methyl]-2,4-dioxo-1,3-thiazolidin-5-ylidene]methyl}-2-methoxyphenoxy)-N-[3,5-bis(trifluoromethyl)phenyl]acetamide hydrochloride* (**13g·HCl**). Yellow solid. Yield: 11 mg (35%). IR ν_max_ 3348, 1731, 1703, 1673, 1607, 1543, 1506, 1377, 1275, 1171, 1127, 1056, 752 cm^−1^. ^1^H NMR (600 MHz, DMSO-d_6_): δ 10.85 (s, 1H, H-1), 9.17 (s, 1H, H-9‴), 8.32 (br s, 2H, H-2′,6′), 8.20 (d, *J* = 8.4 Hz, 1H, H-8‴), 8.17 (dd, *J* = 8.8, 1.0 Hz, 1H, H-5‴), 8.13 (dd, *J* = 8.0, 1.9 Hz, 1H, H-1‴), 7.97 (s, 1H, H-4), 7.89 (ddd, *J* = 8.5, 6.6, 1.4 Hz, 1H, H-6‴), 7.81 (s, 1H, H-4′), 7.66 (ddd, *J* = 8.1, 6.6, 1.2 Hz, 1H, H-7‴), 7.55 (m, 2H, H-2‴, H-3‴), 7.34 (d, *J* = 2.1 Hz, 1H, H-6″), 7.24 (dd, *J* = 8.5, 2.1 Hz, 1H, H-2″), 7.15 (d, *J* = 8.5 Hz, 1H, H-3″), 5.60 (s, 2H, H-10), 4.90 (s, 2H, H-3), and 3.88 (s, 3H, OCH_3_) ppm. ^13^C NMR (151 MHz, DMSO-d_6_): δ 167.5 (C-2), 167.4 (C-8), 165.8 (C-6), 149.4 (C-4″), 149.3 (C-5″), 147.6 (C-10‴a), 146.0 (C-4‴a), 140.3 (C-1′), 136.8 (C-9‴), 133.3 (C-4), 132.1 (C-4‴), 130.9 (C-6‴), 130.8 (q, *J* = 32.0 Hz, C-3′,5′), 129.1 (C-5‴), 128.5 (C-8‴), 128.1 (C-1‴), 127.0 (C-3‴), 126.8 (C-1″), 126.2 (C-7‴), 126.2 (C-8‴a), 126.0 (C-9‴a), 125.3 (C-2‴), 123.4 (C-2″), 123.2 (q, *J* = 272.8 Hz, CF_3_), 119.3 (br s, C-2′,6′), 119.1 (C-5), 116.6 (br s, C-4′), 114.1 (C-6″), 114.1 (C-3″), 67.7 (C-3), 55.8 (OCH_3_), and 42.3 (C-10) ppm. For C_35_H_24_F_6_N_3_O_5_S (748.091 g · mol^−1^) calc.: C 56.19; H 3.23; N 5.62; and S 4.29%, exp.: C 56.06; H 3.22; N 5.61; and S 4.29%.

### 3.9. NMR Spectroscopy

The NMR spectra were recorded on Varian Mercury (Palo Alto, CA, USA, 400.11 MHz for ^1^H) and Varian VNMRS spectrometers (Palo Alto, CA, USA; 599.87 MHz for ^1^H, 150.84 MHz for ^13^C, and 60.79 MHz for ^15^N) with a 5 mm inverse-detection H-X probe equipped with a z-gradient coil at 299.15 K. All the pulse programs were obtained from the Varian sequence library. The chemical shifts (*δ* in ppm) are given relative to the reference standard TMS (0.0 ppm for ^1^H and ^13^C) or internal solvent and the partially deuterated residual DMSO-d_6_ 39.5 ppm for ^13^C and DMSO-d_5_ 2.5 ppm for ^1^H. External nitromethane (0.0 ppm) was used for ^15^N references. The NMR spectra were processed and analyzed in MestReNova v. 15.0.1 (Mestrelab Research, Santiago de Compostela, Spain).

### 3.10. IR Spectroscopy 

The infrared spectra of the prepared compounds were recorded with an Avatar FTIR 6700 (Fourier transform infrared spectroscopy) spectrometer in the range of 400–4000 cm^−1^ with 64 repetitions for a single spectrum using the ATR (attenuated total reflectance) technique. All the obtained data were analyzed using Omnic 8.2.0.387 (2010) software, and the structures of all the new compounds were confirmed through the analysis of the FTIR spectra by functional group identification. 

### 3.11. HRMS 

The solid samples were dissolved in methanol and then diluted to a final concentration of 1 to 5 µg/mL using methanol containing 0.5% formic acid and 5 mM ammonium formate. The samples were injected using a TriVersa NanoMate^®^ nanoelectrospray robot (Advion, Ithaca, NY, USA). The volume of sample aspired into the tip for a single injection was 20 µL, and the maximum spraying rate was approximately 220 nL/min. In the positive mode, the gas pressure (N_2_ extruding the sample from the tip) was set to 0.3 psi, and the applied voltage was 1.4 kV. The samples were injected into an Orbitrap Fusion Lumos mass spectrometer (Thermo Fisher Scientific, Waltham, MA, USA). The ion transfer tube temperature was maintained at 275 °C. A full scan (MS^1^) was performed using an Orbitrap detector with a resolution of 120,000, a maximum injection time of 200 ms, and 2 microscans. Subsequent scans (MS^2^ to MS^4^) were also performed with an Orbitrap detector after fragmentation using CID (collision-induced dissociation), with relative energies ranging from 10 to 100, and HCD (higher-energy collisional dissociation), with relative energies ranging from 10 to 200. The collision pressure was 8 × 10^−3^ Torr. The isolation width for all levels of fragmentation was set to 1 unit (minimum). The automatic gain control was typically set between 2 × 10^4^ and 5 × 10^5^, depending on sample concentration. The obtained data were of high quality as they included high-resolution MS/MS and multi-stage MSn spectra acquired at various collision energies using different fragmentation techniques. The measured data were manually processed using Mass Frontier™ 8.0 software (Thermo Scientific™ Bratislava, Slovakia) within the Curator module. This module employs advanced algorithms to detect incompatibility between the declared structure precursor and the product MS^n^ fragmentation spectra. These compounds were added to the high-quality mzCloud™ spectral library (https://www.mzcloud.org). The mzCloud™ ID are 13149 (**7a**), 13150 (**7b**), 13151 (**7c**), 13152 (**7d**), 13158 (**7e**), 13159 (**7f**), 13160 (**7g**), 13190 (**8a**), 13191 (**8b**), 13168 (**8c**), 13169 (**8d**), 13192 (**8e**), 13193 (**8f**), 13170 (**8g**), 13161 (**12a**), 13180 (**12b**), 13167 (**12c**), 13181 (**12d**), 13182 (**12e**), 13183 (**12f**), 13184 (**12g**), 13194 (**13a**), 13195 (**13b**), 13201 (**13c**), 13202 (**13d**), 13203 (**13e**), 13204 (**13f**), and 13205 (**13g**).

### 3.12. Elemental Analysis

The elemental analysis of C, H, and N was performed using a CHNOS Elemental Analyzer vario MICRO from Elementar Analysensysteme GmbH (Langenselbold, Germany).

### 3.13. Biological Activity 

#### 3.13.1. Cell Lines and Culture Conditions

The human cancer cell lines employed in this study were sourced from reputable institutions, including the American Type Culture Collection (ATCC; Manassas, VA, USA) and the European Collection of Authenticated Cell Cultures (ECACC, Salisbury, UK). 

The human cancer cell lines HeLa (cervical adenocarcinoma), HCT116 (colorectal adenocarcinoma), A2780 (human ovarian adenocarcinoma), A2780cis (human ovarian adenocarcinoma cisplatin-resistant), Jurkat (acute T-lymphoblastic leukemia), and Hep G2 (hepatocellular carcinoma) were cultured in an RPMI 1640 growth medium (Biosera, Kansas City, MO, USA). The cancer cell lines MDA-MB-231 (triple-negative breast adenocarcinoma), A2058 (human melanoma), U87 (human glioblastoma), PATU 8902 (pancreas adenocarcinoma), and A549 (human lung adenocarcinoma) were cultured in a DMEM medium (Biosera, Kansas City, MO, USA). The media were supplemented with 10% fetal bovine serum (FBS) (Invitrogen, Carlsbad, CA, USA) and 1× HyClone™ Antibiotic/Antimycotic Solution (GE Healthcare, Piscataway, NJ, USA).

The non-cancerous cell line MCF-10A (human epithelial breast cells) was cultured in a DMEM F12 medium supplemented with 10% FBS (Invitrogen, Carlsbad, CA, USA), 1× HyClone™ Antibiotic/Antimycotic Solution (GE Healthcare, Piscataway, NJ, USA), insulin, hEGF, and hydrocortisone. The BJ-5ta cells (human dermal fibroblasts) were cultured in a DMEM-M199 4:1 medium mixture, supplemented with 10% FBS and hygromycin B (0.01 mg/mL). The cells were maintained under standard conditions with an atmosphere containing 5% CO_2_ at 37 °C. Cell viability, estimated by trypan blue exclusion, was consistently greater than 95% before each experiment.

#### 3.13.2. MTT Assay 

To evaluate the IC_50_ (half-maximal inhibitory concentration) and confirm the antiproliferative activity of the tested substances, we employed the colorimetric MTT assay (3-(4,5-dimethylthiazol-2-yl)-2,5-diphenyltetrazolium bromide) (Sigma-Aldrich Chemie, Steinheim, Germany). The cell lines were seeded in 96-well plates at a density of 5 × 10^3^ to 10 × 10^3^ cells per well. After 24 h, the tested substances, dissolved in the cultivation medium, were added to the cells at final concentrations of 5, 10, 20, and 50 µmol/L. The cells were then incubated for 48 h under standard cultivation conditions. Following the incubation period, MTT was added to each well. The MTT was metabolized by the cells into insoluble formazan crystals, which were subsequently dissolved by adding 100 µL of 10% SDS. After another 24 h, the absorbance was measured at a wavelength of 540 nm using the automated Cytation™ 3 Cell Imaging Multi-Mode Reader (Biotek, Winooski, VT, USA). The experiments were performed in at least three independent repetitions. IC_50_ values were calculated using a nonlinear regression method [56].

### 3.14. Fluorescence Quenching Studies

#### 3.14.1. Material

The bovine serum albumin (BSA) used in this study was obtained from Sigma-Aldrich (St. Louis, MI, USA).

#### 3.14.2. Fluorescence Spectroscopy

The fluorescence spectra were recorded using a Varian Cary Eclipse spectrofluorometer (Palo Alto, CA, USA) in a 10 mM phosphate-buffered saline solution (pH = 7.4) at 24 °C. The spectra were measured with an excitation wavelength of 280 nm using a slit width of 10 nm for both the excitation and emission beams over a range of 300–450 nm. Spectrofluorometric BSA titrations were performed with increasing concentrations of acridine derivatives.

## 4. Conclusions

In this investigation, we demonstrated the potential of novel derivatives by incorporating three essential components (an acridine scaffold, thiazolidine-2,4-dione with benzylidene linkage, and an aryl acetamido moiety) as an antitumor agent. The strategic integration of these three distinct structural motifs paves the way for the development of potent and selective anticancer agents. 

We efficiently synthesized the newly designed molecules via a convergent multi-step process given that the linear approach was unsuccessful. The successful formation of final compounds **7**, **8**, **12**, and **13**, along with their hydrochloride salt forms, corroborated the precision of our synthetic strategy. The comprehensive characterization of the synthesized derivatives was achieved through advanced spectroscopic techniques, including 1D, 2D NMR, FTIR, HRMS, and elemental analysis, thus confirming their structures.

Our evaluation of the synthesized derivatives against various cancer cell lines revealed compelling antitumor potential. Specifically, derivatives bearing substituents, such as 3,4,5-trimethoxy, 4-nitro, and 2-trifluoromethyl groups and the acridin-9-yl fragment, exhibited lower IC_50_ values, indicating higher potency. Additionally, these derivatives displayed low cytotoxicity against the non-cancerous cell lines MCF-10A and Bj-5ta. Notably, the derivatives **7c** (IC_50_ = 6.80 ± 2.40 μM), **12d** (IC_50_ = 9.40 ± 0.30 μM), and **7d**·**2HCl** (IC_50_ = 4.55 ± 0.35 μM) demonstrated exceptional selectivity and potency against HeLa cell lines. Furthermore, the derivatives **12c**·**2HCl** (IC_50_ = 5.40 ± 2.40 μM), **13d** (IC_50_ = 4.90 ± 2.90 μM), and **7d**·**2HCl** (IC_50_ = 8.60 ± 2.90 μM) exhibited significant efficacy against HCT116 cancer cell lines. It is important to note that three of the tested compounds, namely, **7e**·**2HCl**, **7f**, and **7f**·**2HCl**, showed activity against pancreatic PATU cells. This type of cancer exhibits very high mortality due to asymptomatic early stages, the occurrence of metastases, and frequent resistance to chemotherapy.

The results indicate a strong interaction between the selected acridine derivatives and BSA. Fluorescence spectroscopy suggests that albumin could serve as an effective carrier for transporting these derivatives in the bloodstream.

In conclusion, the novel derivatives synthesized in this study exhibit significant promise as anticancer agents, combining structural innovation with potent biological activity. Future work will focus on further optimizing these compounds for enhanced efficacy and reduced toxicity, as well as exploring their mechanisms of action in greater detail. The promising results obtained thus far underscore the potential of these derivatives to contribute to the development of new effective cancer therapies.

## Data Availability

The data presented in this study and associated additional data are available upon request.

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
