# Peer review of "Design, Synthesis, and Characterization of Novel Thiazolidine-2,4-Dione-Acridine Hybrids as Antitumor Agents"

_molecules, 2024, doi:10.3390/molecules29143387_

Round 1

Reviewer 1 Report

Comments and Suggestions for Authors

These are written in my report. There is no need for repetition.

Comments on the Quality of English Language

Needs improvement

Author Response

Dear Reviewer 1,

I hope this letter finds you in good health and spirits. I am writing to provide an update on our manuscript entitled “Design, Synthesis, and Characterization of Novel Thiazolidine-2,4-Dione-Acridine Hybrids as Antitumor Agents” to the special issue Advances in Synthesis and Biological Activity of Novel Derivatives Based on Five-Membered Heterocyclic Scaffolds and Their Intermediates—Second Edition for consideration for publication in Molecules.

We are pleased to inform you that we have carefully addressed all the suggestions and comments.

We are sincerely grateful for your thorough comments, valuable insights, and the time you dedicated to reviewing our manuscript. Your feedback is instrumental in improving the quality and clarity of our work.

Thank you once again for your time and effort in evaluating our research.

Sincerely,

Assoc. prof. Mária Vilková

Our responses:

The title "Design, Synthesis, and Characterization of Novel Thiazolidine-2,4-Dione and Acridine Derivatives as Antitumor Agents"

- This title is misleading. It doesn’t represent the content of your work. The following title is a much more accurate and informative. "Design, Synthesis, and Characterization of Novel Thiazolidine-2,4-dione-Acridine Hybrids as Antitumor Agents"

This title conveys that your research focuses on compounds that combine both thiazolidine-2,4-dione and acridine moieties within a single molecule.

The title of our manuscript was changed according to your suggestion.

Line 19: accurately should be omitted.

               The sentence was corrected.

Line 29: compared to their free base forms. This should read…. compared with their free base forms.

Please note that compared to indicates similarities (He is highly rich compared to Bill Gates), while compared with indicates differences (He is very poor compared with Bill Gates).

- This remark is to be considered in all the text where it appears.

The sentences were corrected.

Line 93: Considering the above structural features… Do you mean those of Figure 1? If yes, please write Figure 1.

No, we mean structural features mentioned in Introduction. We changed the sentence: Considering the structural features [10,17,30] mentioned in Introduction

Line 101: Scheme 1 and Scheme 2. Should read Schemes 1 and 2.

The sentence was corrected.

Line 104: reactions of 2-chloro-N-phenylacetamides 2a–g and 4-hydroxybenzaldehyde were unsuccsessful….

Should read reactions between 2-chloro-N-phenylacetamides 2a–g and 4-hydroxybenzaldehyde

The sentence was corrected.

line 106: better-leaving group……should read…….. better leaving group

The sentence was corrected.

Line 112: and thiazolidine-2,4-dione…….should read ……….with thiazolidine-2,4-dione

The sentence was corrected.

Line 114: the reactions of 4e–g as well as 5a–g with thiazolidine-2,4-dione (9) were ineffective.

- This should read: the reactions of 4e–g and 5a–g with thiazolidine-2,4-dione (9) were unsuccessful.

- Ineffective …..should read…… unsuccessful

- The authors did not comment on these unsuccessful reactions!.

The yields of these reactions were low. We added the yields in Table in Scheme 1.

Line 115: Then, the introduction of acridine moiety [34] via N-substitution on the thiazolidine-2,4-dione was accomplished.

- This reference has nothing to do with present reaction. This sentence should be omitted.

The reference was deleted.

Lines 116-120: Reactions of derivatives 6a–d with 9-(bromomethyl)acridine were carried out……… . This led to the formation of 2-(4-{[(5Z)-3-[(acridin-9-yl)methyl]-2,4-dioxo-1,3-thiazolidin-5-ylidene]methyl} phenoxy)-N-phenylacetamides 7a–d (Scheme 1).

Should read

The reaction of 6a–d with 9-(bromomethyl)acridine was carried out………………..to give 2-(4-{[(5Z)-3-[(acridin-9-yl)methyl]-2,4-dioxo-1,3-thiazolidin-5-ylidene]methyl}phenoxy)-N-phenylacetamides 7a–d in 54% to 56% yield (Scheme 1).

The sentence was changed.

Line 120: The pure products 7a–d were obtained by crystallization in DMSO/MeOH with yields ranging from 54% to 56%. This sentence should be omitted.

The sentence was deleted.

Line 121: In the same way, the reactions of 6a–d and 4-(bromomethyl)acridine were carried out.

Should read

However, when the derivatives 6a–d were similarly allowed to react with 4-(bromomethyl)acridine, the derivative 6a failed to react under these conditions, while the other derivatives 6b–d afforded the corresponding expected products 8b–d, but in very low yields (16–38) (Scheme 1).

The sentence was changed.

Page 144: Scheme 1 should read Scheme 2.

The number was corrected.

Page 153:

- from …..should read …… of

- One one proton singlet….. should read…..a one proton singlet

The sentence was corrected.

Page 156: The singlets from two methylene groups…… Should read

- The singlets of the two methylene groups …..

The sentence was corrected.

Page 277: In vitro……should read in italic In vitro

The title was corrected.

 Pages 133-137: Under this approach, the initial step involved the reaction of thiazolidine-2,4-dione (9) with KOH in ethanol at room temperature for 2 hours. Next, the isolated and dried potassium salt of thiazolidine-2,4-dione reacted with 9-(bromomethyl)acridine or 4-(bromomethyl)acridine resulting in derivatives 10 and 11, respectively.

- Please improve the writing.

The text was improved.

Line 140: yields higher than 60% except derivative 12d.

- Please give the percentage yield of 12d.

The percentage yield was added.

141- by introducing gaseous hydrogen chloride

- by bubbling dry HCl gas

The sentence was corrected.

Line151: two two-proton singlets

- 2 two-proton singles or if you wish, two singlets each integrating 2 protons

The sentence was corrected.

Line 152: three one-proton singlets

- 3 one-proton singlets or if you wish, three singlets each integrating one proton

The sentence was corrected.

Line 153: and one six-proton singlet

- and a six-proton singlet or if you wish, a singlet integrating six protons

The sentence was corrected.

Line 437: progression…..should read ……progress

The sentence was corrected.

Lines 376-371: the discussion given about SAR in these lines is contradictory and not convincing.

In the same context please note that Figure 9 has a big problem. The authors indicated that 2-(trifluoromethyl)phenyl-, 4-nitrophenyl-, and 3,4,5-trimethoxyphenyl- groups are electron withdrawing groups. This is in error. the third group 3,4,5-trimethoxyphenyl- is an electron donating one!

The Figure 9 was corrected to support the SAR discussion.

Please check also line 1524: derivatives bearing electron-withdrawing substituents, such as 3,4,5-trimethoxy, (the methoxyl group is an electron donating group)

The sentence was corrected

Line 1533: This type of cancer has a very high mortality

- has is not the proper word!

The sentence was corrected

Line 1416: Items 3.7, 3.8, 3.9, and 3.10 may be involved (without much details, only the essentials) under item 3.1 general line 236.

We would like to keep these texts as they are, as this is how we normally present them in the journal Molecules.

Finally,

- what is the positive control you have used in your anticancer evaluation of your compounds?

Positive control was cisplatin. The values of IC50 were added into Tables.

- For the sake of simplicity, you could give new numbers to the HCl salts. Please look at Table 4 and the numbers of the HCl salts!

The Table 4 was corrected.

Reviewer 2 Report

Comments and Suggestions for Authors

The authors presented the design, synthesis, and characterization of thiazolidine-2,4-dione and acridine derivatives and their study as antitumor agents. The work is interesting and well presented. The compounds presented novel while the method reported is standard. 

The products are fully characterized. The 2d-NMR study of the products is well performed.

The manuscript is clear, relevant for the field and presented in a well-structured manner. The cited references are recent publications and relevant.

The manuscript is scientifically sound and the experimental design is appropriate to test the hypothesis.

The manuscript’s results appear to be reproducible based on the details given in the methods section.

The figures/tables/images/schemes are appropriate, easy to interpret and understand.

The conclusions are consistent with the evidence and arguments presented.

Further comments:

-Scheme 1 and 2. Regarding this work please also provide the yield range of each reaction step.

-Line 144. This is scheme 2 not scheme 1. Please correct.

-Experimental: If any of the precursor compounds 2-6 are reported in the literature, the authors must compare the reported data with the literature to prove that the have the same compound and they should provide the relevant reference. Only two sets of data are necessary for literature compounds (such as 1H-NMR and mp). 

-Experimental: The authors don't report IR data for all compounds prepared. Please correct this.

-Experimental: "General synthetic procedure for compounds 7, 8, 12, 13". Give the exact reaction times for each compound.

-The supporting information file is of good quality.

Author Response

Dear Reviewer 2,

I hope this letter finds you in good health and spirits. I am writing to provide an update on our manuscript entitled “Design, Synthesis, and Characterization of Novel Thiazolidine-2,4-Dione-Acridine Hybrids as Antitumor Agents” to the special issue Advances in Synthesis and Biological Activity of Novel Derivatives Based on Five-Membered Heterocyclic Scaffolds and Their Intermediates—Second Edition for consideration for publication in Molecules.

We are pleased to inform you that we have carefully addressed all the suggestions and comments.

We are sincerely grateful for your thorough comments, valuable insights, and the time you dedicated to reviewing our manuscript.

Thank you once again for your time and effort in evaluating our research.

Sincerely,

Assoc. prof. Mária Vilková

Our responses:

-Scheme 1 and 2. Regarding this work please also provide the yield range of each reaction step.

The yield range for each reaction step were added.

-Line 144. This is scheme 2 not scheme 1. Please correct.

The number of Scheme 2 was corrected.

-Experimental: If any of the precursor compounds 2-6 are reported in the literature, the authors must compare the reported data with the literature to prove that the have the same compound and they should provide the relevant reference. Only two sets of data are necessary for literature compounds (such as 1H-NMR and mp). 

The data of previously reported compounds were included.

-Experimental: The authors don't report IR data for all compounds prepared. Please correct this.

The IR data were included.

-Experimental: "General synthetic procedure for compounds 7, 8, 12, 13". Give the exact reaction times for each compound.

The exact reaction times were added.

Reviewer 3 Report

Comments and Suggestions for Authors

The authors of the manuscript "Design, synthesis and characterization of novel thiazolidine-2,4-dione and acridine derivatives as antitumor agents" presented the synthesis, structural characterization and evaluated the cytotoxic potential against several cancer cell lines of novel hybrids of acridine-thiazolidine-2,4-diones. The newly synthesized compounds are very well characterized spectroscopically and the structure is well proven.

The introduction is adequate to the research topic, the research is well planned and described.

The iThenticate report shows 32% similarity to other manuscripts. But the marked fragments are mainly spectroscopic data, the authors have done a lot of synthetic work and characterized all the compounds, so this is not surprising.

In my opinion, it is a good and interesting manuscript.

Apart from a few editing shortcomings, I only see two problems:

- lack of reference compound in studies on cell lines,

- no MS spectra in the supplement.

From an editing point of view, the authors should enter the substance numbers in bold everywhere (not in bold, among others, on lines 447, 526, 560, 578). On line 1421, change DMSO-d5 to DMSO-d6. Some fragments are written in a different font (e.g. lines 1455-1359 and entire references).

I also think that no of compound.HCl should be changed to no of compound·HCl (or no of compound×HCl) throughout the manuscript.

Author Response

Dear Reviewer 3,

I hope this letter finds you in good health and spirits. I am writing to provide an update on our manuscript entitled “Design, Synthesis, and Characterization of Novel Thiazolidine-2,4-Dione-Acridine Hybrids as Antitumor Agents” to the special issue Advances in Synthesis and Biological Activity of Novel Derivatives Based on Five-Membered Heterocyclic Scaffolds and Their Intermediates—Second Edition for consideration for publication in Molecules.

We are pleased to inform you that we have carefully addressed all the suggestions and comments.

We are sincerely grateful for your thorough comments, valuable insights, and the time you dedicated to reviewing our manuscript.

Thank you once again for your time and effort in evaluating our research.

Sincerely,

Assoc. prof. Mária Vilková

Our responses:

Apart from a few editing shortcomings, I only see two problems:

- lack of reference compound in studies on cell lines,

The reference compound and IC50 values were added.

- no MS spectra in the supplement.

MS spectra were included.

From an editing point of view, the authors should enter the substance numbers in bold everywhere (not in bold, among others, on lines 447, 526, 560, 578).

We corrected all substance's numbers.

On line 1421, change DMSO-d5 to DMSO-d6.

Changing DMSO-d5 to DMSO-d6 on line 1421 is not appropriate because in 1H NMR spectroscopy, the residual proton signal from DMSO-d5 is detectable, while DMSO-d6, being fully deuterated, does not produce a signal in the 1H NMR spectrum. Therefore, DMSO-d5 is the correct designation.

Some fragments are written in a different font (e.g. lines 1455-1359 and entire references).

The font was corrected.

I also think that no of compound.HCl should be changed to no of compound·HCl (or no of compound×HCl) throughout the manuscript.

The symbols in numbers of hydrochlorides have been corrected.

Round 2

Reviewer 1 Report

Comments and Suggestions for Authors

Further remarks

Please note that the reference numbers should always precede the punctuation marks or the commas such as:

  • Incorrect: ….promising direction in drug development.[16,17]
  • Correct: …..promising direction in drug development [16,17].

Also  

          ….. including antitumor, [18–20] antibacterial, [21,22] (incorrect)

         ….. including antitumor [18–20], antibacterial [21,22], (correct)

This should be applied to all the text

Line 93

mentioned in Introduction   should read    mentioned in the introduction

Author Response

Dear Reviewer,

we are pleased to inform you that we have carefully corrected all your remarks mentioned bellow. We are sincerely grateful for your thorough comments and the time you dedicated to reviewing our manuscript.

Please note that the reference numbers should always precede the punctuation marks or the commas such as:

  • Incorrect: ….promising direction in drug development.[16,17]
  • Correct: …..promising direction in drug development [16,17].

Also  

          ….. including antitumor, [18–20] antibacterial, [21,22] (incorrect)

         ….. including antitumor [18–20], antibacterial [21,22], (correct)

This should be applied to all the text

Line 93

mentioned in Introduction   should read    mentioned in the introduction